# Robust Federated Finetuning of LLMs via Alternating Optimization of LoRA

## Abstract

Parameter-Efficient Fine-Tuning (PEFT) methods like Low-Rank Adaptation (LoRA) optimize federated training by reducing computational and communication costs. We propose RoLoRA, a federated framework using alternating optimization to fine-tune LoRA adapters. Our approach emphasizes the importance of learning up and down projection matrices to enhance expressiveness and robustness. We use both theoretical analysis and extensive experiments to demonstrate the advantages of RoLoRA over prior approaches that either generate imperfect model updates or limit expressiveness of the model. We present theoretical analysis on a simplified linear model to demonstrate the importance of learning both down-projection and up-projection matrices in LoRA. We provide extensive experimental evaluations on a toy neural network on MNIST as well as large language models including RoBERTa-Large, Llama-2-7B on diverse tasks to demonstrate the advantages of RoLoRA over other methods.

## 1. Introduction

The remarkable performance of large language models (LLMs) stems from their ability to learn at scale. With their broad adaptability and extensive scope, LLMs depend on vast and diverse datasets to effectively generalize across a wide range of tasks and domains. Federated learning (McMahan et al., 2017) offers a promising solution for leveraging data from multiple sources, which could be particularly advantageous for LLMs.

Recently, Parameter-Efficient Fine-Tuning (PEFT) has emerged as an innovative training strategy that updates only a small subset of model parameters, substantially reducing computational and memory demands. A notable method in this category is LoRA (Hu et al., 2021), which utilizes

low-rank matrices to approximate weight changes during fine-tuning. These matrices are integrated with pre-trained weights for inference, facilitating reduced data transfer in scenarios such as federated learning, where update size directly impacts communication efficiency. Many works integrate LoRA into federated setting (Zhang et al., 2023b; Babakniya et al., 2023; Kuang et al., 2023; Chen et al., 2024; Sun et al., 2024). FedPETuning (Zhang et al., 2023b) compares various PEFT methods in a federated setting. SLoRA (Babakniya et al., 2023) presents a hybrid approach that combines sparse fine-tuning with LoRA to address data heterogeneity in federated settings. Furthermore, FS-LLM (Kuang et al., 2023) presents a framework for fine-tuning LLMs in federated environments. However, these studies typically apply the FedAVG algorithm directly to LoRA modules, resulting in in-exact model updates, as we will discuss later in the paper.

To address the issue of in-exact model updates, a few recent works have proposed modifications to the down-projection and up-projection components in LoRA. In FlexLoRA (Bai et al., 2024), the authors propose updating these projections with matrix multiplication followed by truncated SVD. A related method is also considered in (Wang et al., 2024). Another approach, by Sun et al., introduces a federated finetuning framework named FFA-LoRA (Sun et al., 2024), which builds on LoRA by freezing the down-projection matrices across all clients and updating only the up-projection matrices. They apply differential privacy (Dwork et al., 2006) to provide privacy guarantees for clients' data. With a sufficient number of finetuning parameters, FFA-LoRA, using a larger learning rate, can achieve performance comparable to FedAVG of LoRA while reducing communication costs by half. However, we observe that with fewer fine-tuning parameters, FFA-LoRA is less robust than FedAVG of LoRA, primarily due to its reduced expressiveness from freezing down-projections. In this work, we explore the necessity of learning down-projection matrices and propose a federated fine-tuning framework with computational and communication advantages.

We connect the objective of learning down-projection matrices in a federated setting to multitask linear representation learning (MLRL), an approach in which a shared low-rank representation is jointly learned across multiple tasks. While, to the best of our knowledge, the alternating opti-

[1]Anonymous Institution, Anonymous City, Anonymous Region, Anonymous Country. Correspondence to: Anonymous Author <anon.email@domain.com>.

Preliminary work. Under review by the International Conference on Machine Learning (ICML). Do not distribute.

*Figure 1.* RoLoRA framework overview.

mization of down- and up-projection matrices has not been explored within the context of LoRA, prior works on MLRL (Collins et al., 2021; Thekumparampil et al., 2021) have demonstrated the importance of alternately updating low-rank representations and task-specific heads, demonstrating the necessity of learning a shared representation. Inspired by MLRL, we tackle this challenge by employing alternating optimization for LoRA adapters. We theoretically establish that alternating updates to the two components of LoRA, while maintaining a common global model, enable effective optimization of down-projections and ensure convergence to the global minimizer in a tractable setting.

### 1.1. Main Contributions

- **RoLoRA framework.** We propose RoLoRA, a robust federated fine-tuning framework based on the alternating optimization of LoRA as shown in Figure 1. RoLoRA fully leverages the expressiveness of LoRA adapters while keeping the computational and communication advantages.
- **Theoretical Insights.** We show that in a tractable setting involving a local linear model, RoLoRA converges exponentially to the global minimizer when clients solve linear regression problems, using rank-1 LoRA adapters. In this case, RoLoRA is reduced to an alternating minimization-descent approach, outperforming FFA-LoRA, whose fixed down-projection limits performance. This highlights the importance of training the down-projection in LoRA for improved federated learning performance.
- **Empirical results.** Through evaluations on a two-layer neural network with MNIST and on large language models (RoBERTa-Large, Llama-2-7B) across various tasks (GLUE, HumanEval, MMLU, Commonsense reasoning tasks), we demonstrate that RoLoRA maintains robustness against reductions in fine-tuning parameters and increases in client numbers compared to prior approaches.

### 1.2. Notations

We adopts the notation that lower-case letters represent scalar variables, lower-case bold-face letters denote column vectors, and upper-case bold-face letters denote matrices.

The $d \times d$ identity matrix is represented by $\mathbf{I}_d$. Depending on the context, $\|.\|$ denotes the $l_2$ norm of a vector or the Frobenius norm of a matrix, $\|.\|_{op}$ denotes the operator norm of a matrix, $|.|$ denotes the absolute value of a scalar, $\top$ denotes matrix or vector transpose. For a number $N$, $[N] = \{1, \ldots, N\}$.

## 2. Related Works

**Parameter Efficient Fine Tuning (PEFT): LoRA and Its Enhancements** Parameter efficient finetuning (PEFT) allows for updates to a smaller subset of parameters, significantly reducing the computational and memory requirements. One of the most well-known methods is LoRA(Hu et al., 2021). LoRA uses low-rank matrices to approximate changes in weights during fine-tuning, allowing them to be integrated with pre-trained weights before inference. In (Zhang et al., 2023a), the authors propose a memory-efficient fine-tuning method, LoRA-FA, which keeps the projection-down weight fixed and updates the projection-up weight during fine-tuning. In (Zhu et al., 2024), the authors highlight the asymmetry between the projection-up and projection-down matrices and focus solely on comparing the effects of freezing either the projection-up or projection-down matrices. (Hao et al., 2024) introduces the idea of resampling the projection-down matrices, aligning with our observation that freezing projection-down matrices negatively impacts a model's expressiveness. Furthermore, (Hayou et al., 2024) explore the distinct roles of projection-up and projection-down matrices, enhancing performance by assigning different learning rates to each.

**PEFT in Federated Setting** PEFT adjusts only a few lightweight or a small portion of the total parameters for specific tasks, keeping most foundational model parameters unchanged. This feature can help reduce data transfer in federated learning, where communication depends on the size of updates. Zhang et al. (Zhang et al., 2023b) compares multiple PEFT methods in federated setting, including Adapter(Houlsby et al., 2019), LoRA(Hu et al., 2021), Prompt tuning(Liu et al., 2022) and Bit-Fit(Zaken et al., 2022). SLoRA(Babakniya et al., 2023), which combines sparse finetuning and LoRA, is proposed to address the data heterogeneity in federated setting. As discussed before,

(Sun et al., 2024) design a federated finetuning framework FFA-LoRA by freezing projection-down matrices for all the clients and only updating projection-up matrices. FLoRA (Wang et al., 2024) considers clients with heterogeneous-rank LoRA adapters and proposes a federated fine-tuning approach.

## 3. Preliminaries

### 3.1. Low-Rank Adaptation: LoRA

Low-Rank Adaptation (LoRA) (Hu et al., 2021) fine-tunes large language models efficiently by maintaining the original model weights fixed and adding small, trainable matrices in each layer. These matrices perform low-rank decompositions of updates, reducing the number of trainable parameters. This approach is based on the finding that updates to model weights during task-specific tuning are usually of low rank, which allows for fewer parameters to be adjusted. For example, for a pre-trained weight matrix $\mathbf{W}_0 \in \mathbb{R}^{d \times d}$, the update is a low-rank product $\mathbf{AB}$, where the down-projection $\mathbf{A} \in \mathbb{R}^{d \times r}$ and the up-projection $\mathbf{B} \in \mathbb{R}^{r \times d}$, with $r \ll d$. Only $\mathbf{A}$ and $\mathbf{B}$ are trainable, allowing $\mathbf{W} = \mathbf{W}_0 + \alpha \mathbf{AB}$, with $\alpha$ adjusting the update's impact. Applying LoRA in a federated setting is a practical choice. By using LoRA adapters, clients can fine-tune foundation models efficiently with limited resources. Since only these specific matrices need to be transmitted to a central server, this approach significantly reduces communication costs. This makes LoRA an advantageous solution for enhancing model performance in collaborative scenario comparing to full parameter finetuning in the federated setting.

### 3.2. FedAVG of LoRA Introduces Interference

Integrating LoRA within a federated setting presents challenges. In such a setup, each of the $N$ clients is provided with the pretrained model weights $\mathbf{W}_0$, which remain fixed during finetuning. Clients are required only to send the updated matrices $\mathbf{B}_i$ and $\mathbf{A}_i$ to a central server for aggregation. While most current studies, such as SLoRA (Babakniya et al., 2023) and FedPETuning (Zhang et al., 2023b), commonly apply FedAVG directly to these matrices as shown in (2), this approach might not be optimal. The precise update for each client's model, $\Delta \mathbf{W}_i$, should be calculated as the product of the low-rank matrices $\mathbf{A}_i$ and $\mathbf{B}_i$. Consequently, aggregation on the individual matrices leads to inaccurate model aggregation.

$$\frac{1}{N} \sum_{i=1}^{N} \Delta \mathbf{W}_i = \frac{1}{N}(\mathbf{A}_1 \mathbf{B}_1 + \mathbf{A}_2 \mathbf{B}_2 + ... + \mathbf{A}_N \mathbf{B}_N) \quad (1)$$

$$\neq \frac{1}{N}(\mathbf{A_1} + \mathbf{A_2} + ... + \mathbf{A_N})\frac{1}{N}(\mathbf{B_1} + \mathbf{B_2} + ... + \mathbf{B_N}) \quad (2)$$

There are a few options to avoid it.

**Updating B and A by matrix multiplication and truncated-SVD.** One approach (Wang et al., 2024; Bai et al., 2024) involves first computing the product of local matrices $\mathbf{B}_i$ and $\mathbf{A}_i$ to accurately recover $\Delta \mathbf{W}_i$. Then, the global $\mathbf{B}$ and $\mathbf{A}$ of next iteration are obtained by performing truncated SVD on the averaged set of $\Delta \mathbf{W}_i$. However, this method introduces computational overhead due to the matrix multiplication and SVD operations.

**Freezing A (B) during finetuning.** Another method is to make clients freeze $\mathbf{B}$ or $\mathbf{A}$ as in Sun et al. (Sun et al., 2024), leading to precise computation of $\Delta \mathbf{W}$. However, this method limits the expressiveness of the adapter.

With these considerations, we propose a federated finetuning framework, named RoLoRA, based on alternating optimization of LoRA.

## 4. RoLoRA Framework

In this section, we describe the framework design of RoLoRA and discuss its practical advantages.

**Alternating Optimization and Corresponding Aggregation** Motivated by the observations discussed in Section 3.2, we propose applying alternating optimization to the local LoRA adapters of each client in a setting with $N$ clients. Unlike the approach in FFA-LoRA, where $\mathbf{A}$ is consistently frozen, we suggest an alternating update strategy. There are alternating odd and even communication rounds designated for updating, aggregating $\mathbf{A}$ and $\mathbf{B}$, respectively.

$$\text{In the odd comm. round:} \quad \frac{1}{N} \sum_{i=1}^{N} \Delta \mathbf{W}_i^{2t+1}$$

$$= \frac{1}{N}(\mathbf{A}_1^t \mathbf{B}_1^{t+1} + \mathbf{A}_2^t \mathbf{B}_2^{t+1} + ... + \mathbf{A}_N^t \mathbf{B}_N^{t+1}) \quad (3)$$

$$= \frac{1}{N}\mathbf{A}^t(\mathbf{B}_1^{t+1} + \mathbf{B}_2^{t+1} + ... + \mathbf{B}_N^{t+1})$$

$$\text{In the even comm. round:} \quad \frac{1}{N} \sum_{i=1}^{N} \Delta \mathbf{W}_i^{2t+2}$$

$$= \frac{1}{N}(\mathbf{A}_1^{t+1} \mathbf{B}_1^{t+1} + \mathbf{A}_2^{t+1} \mathbf{B}_2^{t+1} + ... + \mathbf{A}_N^{t+1} \mathbf{B}_N^{t+1}) \quad (4)$$

$$= \frac{1}{N}(\mathbf{A}_1^{t+1} + \mathbf{A}_2^{t+1} + ... + \mathbf{A}_N^{t+1})\mathbf{B}^{t+1}$$

As in Algorithm 1, all clients freeze $\mathbf{A}^t$ and update $\mathbf{B}^t$ in the odd communication round. The central server then aggregates these updates to compute $\mathbf{B}^{t+1} = \frac{1}{N} \sum_{i=1}^{N} \mathbf{B}_i^{t+1}$ and distributes $\mathbf{B}^{t+1}$ back to the clients. In the subsequent communication round, clients freeze $\mathbf{B}^{t+1}$ and update $\mathbf{A}^t$. The server aggregates these to obtain $\mathbf{A}^{t+1} = \frac{1}{N} \sum_{i=1}^{N} \mathbf{A}_i^{t+1}$ and returns $\mathbf{A}^{t+1}$ to the clients. It is important to note that in round $2t+1$, the frozen $\mathbf{A}_i^t$ are identical across all clients, as they are synchronized with $\mathbf{A}^t$ from the central server at the beginning of the round. This strategy ensures that the update and aggregation method introduces no interference, as demonstrated in (3) and (4).

---

**Algorithm 1** RoLoRA iterations

---

1: **Input:** number of iterations $T$, number of clients $N$
2: **for** $t = 1$ **to** $T$ **do**
3:     **for** $i = 1$ **to** $N$ **do**
4:         Fix $\mathbf{A}^t$, $\mathbf{B}_i^{t+1} = $ GD-update$(\mathbf{A}^t, \mathbf{B}^t)$
5:         Transmits $\mathbf{B}_i^{t+1}$ to server
6:     **end for**
7:     Server aggregates $\mathbf{B}^{t+1} = \frac{1}{N}\sum_{i=1}^N \mathbf{B}_i^{t+1}$, broadcasts $\mathbf{B}^{t+1}$
8:     **for** $i = 1$ **to** $N$ **do**
9:         Fix $\mathbf{B}^{t+1}$, $\mathbf{A}_i^{t+1} = $ GD-update$(\mathbf{A}^t, \mathbf{B}^{t+1})$
10:       Transmits $\mathbf{A}_i^{t+1}$ to server
11:     **end for**
12:     Server aggregates $\mathbf{A}^{t+1} = \frac{1}{N}\sum_{i=1}^N \mathbf{A}_i^{t+1}$, broadcasts $\mathbf{A}^{t+1}$
13: **end for**

---

**Computation and Communication Cost** The parameter-freezing nature of RoLoRA enhances computational and communication efficiency. In each communication round, the number of trainable parameters in the model is effectively halved compared to FedAVG of LoRA. The only additional cost for RoLoRA compared to FFA-LoRA is the alternating freezing of the corresponding parameters. We remark this additional cost is negligible because it is applied to the clients' models and can be executed concurrently during the server's aggregation.

## 5. Analysis

In this section, we provide an intuitive analysis of the necessity of training down-projection of LoRA module in a federated setting. We first theoretically compare RoLoRA and FFA-LoRA on a linear model. Then we empirically verify the effectiveness of the theoretical analysis on a two-layer neural network.

### 5.1. Federated LoRA on a Toy Model

Consider a federated setting with $N$ clients, each with the following local linear model

$$f_i(\mathbf{X}_i) = \mathbf{X}_i \mathbf{a}\mathbf{b}^\top \quad (5)$$

where $\mathbf{Y}_i \in \mathbb{R}^{m \times d}$, $\mathbf{X}_i \in \mathbb{R}^{m \times d}$ with the sample size $m$, $\mathbf{a} \in \mathbb{R}^d$ (a unit vector) and $\mathbf{b} \in \mathbb{R}^d$ are the LoRA weights corresponding to rank $r = 1$. In this setting, we model the local data of $i$-th client such that

$$\mathbf{Y}_i = \mathbf{X}_i \mathbf{a}^* \mathbf{b}^{*^\top} \quad (6)$$

for some ground truth LoRA weights $\mathbf{a}^* \in \mathbb{R}^d$ (a unit vector) and $\mathbf{b}^* \in \mathbb{R}^d$. We consider the following objective

$$\min_{\mathbf{a} \in \mathbb{R}^d, \mathbf{b} \in \mathbb{R}^d} \frac{1}{N} \sum_{i=1}^N l_i(\mathbf{a}, \mathbf{b}) \quad (7)$$

where the local loss is $l_i(\mathbf{a}, \mathbf{b}) = \frac{1}{m}\|\mathbf{X}_i \mathbf{a}^* \mathbf{b}^{*^\top} - \mathbf{X}_i \mathbf{a}\mathbf{b}^\top\|^2$. Each $\mathbf{X}_i$ is assumed to be a Gaussian random matrix, where each entry is independently and identically distributed according to a standard Gaussian distribution.

We remind the reader that Section 1.2 provides a summary of mathematical notations and also point to Table 3 in Appendix A.1 for a summary of the symbols used throughout the theoretical analysis.

**Results.** In this section, we assume homogeneous clients where there is a single target model as in (6). In the special case with the model as in (5) and the objective in (7), we modify RoLoRA from Algorithm 1 to Algorithm 2, employing alternating minimization for $\mathbf{b}$ (line 5) and gradient descent for $\mathbf{a}$ (line 9). Details are described in Algorithm 2. We note that the analysis of the alternating minimization-gradient descent algorithm is inspired by (Collins et al., 2021; Seyedehsara et al., 2022; Vaswani, 2024) for a different setting of MLRL. We aim to show that the training

---

**Algorithm 2** RoLoRA for linear regressor, Alt-min-GD iterations

---

1: **Input:** GD Step size $\eta$, number of iterations $T$, number of clients $N$
2: **for** $t = 1$ **to** $T$ **do**
3:     Let $\mathbf{a} \leftarrow \mathbf{a}^{t-1}, \mathbf{b} \leftarrow \mathbf{b}^{t-1}$.
4:     **for** $i = 1$ **to** $N$ **do**
5:         set $\tilde{\mathbf{b}}_i \leftarrow \operatorname{argmin}_{\mathbf{b}} l_i(\mathbf{a}, \mathbf{b})$
6:     **end for**
7:     $\bar{\mathbf{b}} = \frac{1}{N}\sum_{i=1}^N \tilde{\mathbf{b}}_i$
8:     **for** $i = 1$ **to** $N$ **do**
9:         Compute $\nabla_{\mathbf{a}} l_i(\mathbf{a}, \bar{\mathbf{b}})$
10:     **end for**
11:     $\hat{\mathbf{a}}^+ \leftarrow \mathbf{a} - \frac{\eta}{N}\sum_{i=1}^N \nabla_{\mathbf{a}} l_i(\mathbf{a}, \bar{\mathbf{b}})$, $\hat{\mathbf{a}} \leftarrow \frac{\hat{\mathbf{a}}^+}{\|\hat{\mathbf{a}}^+\|}$
12:     $\mathbf{a}^t \leftarrow \hat{\mathbf{a}}, \mathbf{b}^t \leftarrow \bar{\mathbf{b}}$
13: **end for**

---

procedure described in Algorithm 2 learns the target model $(\mathbf{a}^*, \mathbf{b}^*)$. First, we make typical assumptions on the ground truth $\mathbf{b}^*$.

**Assumption 5.1.** There exists $L_{max} < \infty$ (known a priori), s.t. $\|\mathbf{b}^*\| \leq L_{max}$.

Next, to obtain the convergence results, we define the angle distance between two unit vectors.

**Definition 5.2.** (Angle Distance) For two unit vectors $\mathbf{a}, \mathbf{a}^* \in \mathbb{R}^d$, the angle distance between $\mathbf{a}$ and $\mathbf{a}^*$ is defined as

$$|\sin\theta(\mathbf{a}, \mathbf{a}^*)| = \|(\mathbf{I}_d - \mathbf{a}\mathbf{a}^\top)\mathbf{a}^*\| \quad (8)$$

where $\mathbf{I}_d - \mathbf{a}\mathbf{a}^\top$ is the projection operator to the direction orthogonal to $\mathbf{a}$.

Let $\delta^t = \|(\mathbf{I}_d - \mathbf{a}^*\mathbf{a}^{*\top})\mathbf{a}^t\| = \|(\mathbf{I}_d - \mathbf{a}^t\mathbf{a}^{t\top})\mathbf{a}^*\|$ denote the angle distance between $\mathbf{a}^*$ and $\mathbf{a}^t$ of $t$-th iteration. We initialize $\mathbf{a}^0$ such that $|\sin\theta(\mathbf{a}^*, \mathbf{a}^0)| = \delta_0$, where $0 < \delta_0 < 1$, and $b^0$ is zero. All clients obtain the same initialization for parameters. We show that the algorithm learns the target model by showing the angle distance between $\mathbf{a}$ and $\mathbf{a}^*$ is decreasing in each iteration. Now we are ready to state our main results.

**Lemma 5.3.** *Let $\delta^t = \|(\mathbf{I}_d - \mathbf{a}^*\mathbf{a}^{*\top})\mathbf{a}^t\|$ be the angle distance between $\mathbf{a}^*$ and $\mathbf{a}^t$ of $t$-th iteration. Assume that Assumption 5.1 holds and $\delta^t \leq \delta^{t-1} \leq \cdots \leq \delta^0$. Let $m$ be the number of samples for each updating step, let auxiliary error thresholds $\epsilon' = \frac{\epsilon_2}{(1-\epsilon_0)(1-\epsilon_1)}, \tilde{\epsilon} = \frac{\epsilon_3}{1-\epsilon_0}$ for $\epsilon_0, \epsilon_1, \epsilon_2, \epsilon_3 \in (0, 1)$, if $m = \Omega(q)$ for $q = \max\left(\frac{\log(N)}{[\min(\epsilon_1, \epsilon_2)]^2}, \frac{d\log(\frac{2}{\epsilon_0})}{\epsilon_2^2}\right)$, and auxiliary error thresholds are small such that $\epsilon', \tilde{\epsilon} < \frac{1-(\delta^0)^2}{16}$, for any $t$ and $\eta \leq \frac{1}{L_{max}^2}$, then we have,*

$$\delta^{t+1} \leq \delta^t \sqrt{1 - \eta(1 - \delta^{0^2})\|\mathbf{b}^*\|^2} \qquad (9)$$

*with probability at least $1 - 2q^{-10}$.*

Theorem 5.4 follows by recursively applying Lemma 5.3 and taking a union bound over all $t \in [T]$.

**Theorem 5.4.** *(Convergence of RoLoRA for linear regressor in homogeneous setting) Suppose we are in the setting described in Section 5.1 and apply Algorithm 2 for optimization. Given a random initial $\mathbf{a}^0$, an initial angle distance $\delta_0 \in (0, 1)$, we set step size $\eta \leq \frac{1}{L_{max}^2}$ and the number of iterations $T \geq \frac{2}{c(1-(\delta^0)^2)} \log(\frac{\delta^0}{\epsilon})$, for $c \in (0, 1)$. Under these conditions, if with sufficient number of samples $m = \Omega(q)$ and small auxiliary error thresholds $\epsilon' = \frac{\epsilon_2}{(1-\epsilon_0)(1-\epsilon_1)}, \tilde{\epsilon} = \frac{\epsilon_3}{1-\epsilon_0}$, such that $\epsilon', \tilde{\epsilon} < \frac{1-(\delta^0)^2}{16}$, we achieve that with probability at least $1 - 2Tq^{-10}$ for $q = \max\left(\frac{\log(N)}{[\min(\epsilon_1, \epsilon_2)]^2}, \frac{d\log(\frac{2}{\epsilon_0})}{\epsilon_2^2}\right)$,*

$$\sin\theta(\mathbf{a}^T, \mathbf{a}^*) \leq \epsilon$$

*which we refer to as $\epsilon$-accurate recovery. In addition,*

$$\|\mathbf{a}^T(\mathbf{b}^{T+1})^\top - \mathbf{a}^*(\mathbf{b}^*)^\top\| \leq (1 + \epsilon')\epsilon\|\mathbf{a}^*\mathbf{b}^{*\top}\|.$$

Theorem 5.4 and Lemma 5.3 show that with a random initialization for the unit vector $\mathbf{a}$ ($\delta^0 \in (0, 1)$), RoLoRA makes the global model converge to the target model exponentially fast with large $q$. The requirement for sample complexity is well-supported, as demonstrated in (Collins et al., 2022; Du et al., 2021).

While the proof of the above results are relegated to the Appendix, we provide a brief outline of the proof. In Appendices A.3, we first analyze the minimization step

for updating $\mathbf{b}_i^t$ (Lemma A.9), then establish a bound on the deviation of the gradient from its expectation with respect to $\mathbf{a}$ (Lemma A.10), and finally derive a bound for $|\sin\theta(\mathbf{a}^{t+1}, \mathbf{a}^*)|$ based on the gradient descent update rule for $\mathbf{a}$ (Lemma 5.3). The proof of Theorem 5.4 is in Section A.4.

**Intuition on Freezing-A Scheme (FFA-LoRA) can Saturate.** We begin by applying the FFA-LoRA scheme to a centralized setting, aiming to solve the following optimization problem:

$$\min_{\mathbf{b}\in\mathbb{R}^d} \|\mathbf{X}\mathbf{a}^*\mathbf{b}^{*\top} - \mathbf{X}\mathbf{a}^0\mathbf{b}^\top\|^2 \qquad (10)$$

where $\mathbf{a}^* \in \mathbb{R}^d$ and $\mathbf{b}^* \in \mathbb{R}^d$ represent the ground truth parameters, and $\mathbf{a}^0 \in \mathbb{R}^d$ is the random initialization. The objective can be transformed to $\sum_{p=i}^d (\mathbf{a}^*b_p^* - \mathbf{a}^0b_p)^\top\mathbf{X}^\top\mathbf{X}(\mathbf{a}^*b_p^* - \mathbf{a}^0b_p)$, with $b_p$ as the $p$-th entry of $\mathbf{b}$, $b_p^*$ as the $p$-th entry of $\mathbf{b}^*$. In FFA-LoRA scheme, $\mathbf{a}^0$ remains fixed during training. If $\mathbf{a}^0$ is not initialized to be parallel to $\mathbf{a}^*$, the objective can never be reduced to zero. This is because optimizing $\mathbf{b}$ only scales the vector $\mathbf{a}^0b_p$ along the direction of $\mathbf{a}^0$, without altering the angular distance between $\mathbf{a}^0$ and $\mathbf{a}^*$.

Suppose we are in the federated setting described in Section 5.1, we apply FFA-LoRA, to optimize the objective in (7). In FFA-LoRA scheme, we fix $\mathbf{a}$ of all clients to a random unit vector $\mathbf{a}^0$, where the initial angle distance $\delta^0 = |\sin\theta(\mathbf{a}^*, \mathbf{a}^0)|, \delta^0 \in (0, 1)$. And we only update $\mathbf{b}_i$ by minimizing $l_i$ and aggregate them.

**Proposition 5.5.** *(FFA-LoRA lower bound) Suppose we are in the setting described in Section 5.1. For any set of ground truth parameters $(\mathbf{a}^*, \mathbf{b}^*)$, the initialization $\mathbf{a}^0$, initial angle distance $\delta^0 \in (0, 1)$, we apply FFA-LoRA scheme to obtain a shared global model $(\mathbf{a}^0, \mathbf{b}^{FFA})$, yielding an expected global loss of*

$$\mathbb{E}[\frac{1}{Nm}\sum_{i=1}^N \|\mathbf{X}_i\mathbf{a}^*\mathbf{b}^{*\top} - \mathbf{X}_i\mathbf{a}^0(\mathbf{b}^{FFA})^\top\|^2]$$
$$= (1 + \tilde{c})\|\mathbf{b}^*\|^2(\delta^0)^2 \qquad (11)$$

*where the expectation is over all the randomness in the $\mathbf{X}_i$, and $\tilde{c} = O(\frac{1}{Nm})$.*

See Appendix A.4.1 for the proof. Proposition 5.5 shows that for any choice of $\delta^0 \in (0, 1)$, the global objective reached by FFA-LoRA is shown as in (11). The global objective of FFA-LoRA is dominated by $\|\mathbf{b}^*\|^2(\delta^0)^2$ which is due to the angular distance between $\mathbf{a}^0$ and $\mathbf{a}^*$.

By Theorem 5.4, we demonstrate that RoLoRA achieves $\epsilon$-accurate recovery of the global minimizer. Specifically, the expected global loss of RoLoRA can be upper bounded

by $(1 + \tilde{c})\|\mathbf{b}^*\|^2\epsilon^2$. Under the same initialization and ground truth parameters for both FFA-LoRA and RoLoRA, RoLoRA's ability to update $\mathbf{a}$ reduces the global loss caused by the angle distance between $\mathbf{a}$ and $\mathbf{a}^*$ from $\|\mathbf{b}^*\|^2(\delta^0)^2$ to $\|\mathbf{b}^*\|^2\epsilon^2$. By increasing the number of iterations, $\epsilon$ can be made arbitrarily small.

In Appendix A.5, we analyze the convergence of RoLoRA with single rank-1 LoRA structure in a federated setting with *heterogeneous* clients. By showing the decreasing of the angle distance between the ground truth $\mathbf{a}^*$ and the shared down-projection $\mathbf{a}$, we demonstrate that RoLoRA allows the global model to converge to global minimum while the global loss of FFA-LoRA can be dominated by the term caused by the angle distance between the random initialization $\mathbf{a}^0$ and $\mathbf{a}^*$.

Through this analysis of the LoRA structure with rank-1, we highlight the necessity of updating the down-projections.

### 5.2. Verifying Results On a Two-Layer NN

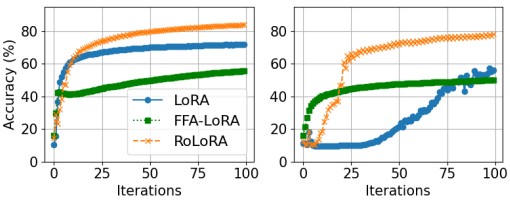

*Figure 2.* (Left) Comparison of three methods on a toy model with 5 clients. (Right) Comparison of three methods on a toy model with 10 clients.

The previous analysis considers a simple linear model for each client. To assess the validity in a non-linear model, we consider a two-layer neural network model on each client given by

$$f_i(x_i) = \mathsf{ReLU}(x_i\mathbf{A}\mathbf{B})\mathbf{W}_{out} \tag{12}$$

where $\mathbf{W}_{out} \in \mathbb{R}^{d \times c}$, $\mathbf{A} \in \mathbb{R}^{d \times r}$ and $\mathbf{B} \in \mathbb{R}^{r \times d}$ are weights. We train the model on MNIST (Deng, 2012) with 60,000 training images. We consider two different ways to distribute training images to clients. The first is to distribute the images to 5 clients and each client gets access to training images of two specific labels, while the second is to distribute the images to 10 clients and each client only has training images of one specific label. There is no overlap in the training samples each client can access. Only weights matrices $\mathbf{B}$ and $\mathbf{A}$ are tunable, while $\mathbf{W}_{out}$ are fixed. We apply the typical initialization, where $\mathbf{A}$ is initialized to a random Gaussian matrix, and $\mathbf{B}$ is initialized to zero. We use $c = 10, d = 784, r = 16$ and make each client train 5 epochs locally with batch-size 64 and aggregate clients' update following three methods: FedAVG of LoRA, referred as LoRA; FFA-LoRA (Sun et al., 2024), which freezes $\mathbf{A}$ during training, and RoLoRA, which alternately update $\mathbf{B}$

and $\mathbf{A}$. We experiment with multiple learning rates, display the best performance of each method in Figure 2.

As shown in Figure 2, we evaluate the performance of the model in each iteration on the test set with 10,000 images. We observe that the accuracy of FFA-LoRA plateaus around 55% in both settings, which aligns with our theoretical analysis. The decline in LoRA's performance with an increasing number of clients is most likely due to less accurate model aggregation, as demonstrated in (1) and (2). Notably, RoLoRA demonstrates greater robustness in these settings.

## 6. Experiments on LLMs

In this section, we evaluate the performance of RoLoRA in various federated settings. Considering all clients will participate in each round, we will explore the following methods based on FedAVG.

- **LoRA** means LoRA adapter and its finetuning algorithm are directly applied to local finetuning of clients in the federated system. Specifically, in iteration $t$, the server receives $\mathbf{A}_i^t$ and $\mathbf{B}_i^t$ from client $i$ and aggregates by $\mathbf{A}^t = \mathsf{Avg}(\mathbf{A}_i^t)$ and $\mathbf{B}^t = \mathsf{Avg}(\mathbf{B}_i^t)$.
- **LoRA-FFA** (Sun et al., 2024) is a baseline that enable the clients to finetune $\mathbf{B}$ and keep $\mathbf{A}$ frozen locally. Thus, in iteration $t$, the server aggregates by $\mathbf{B}^t = \mathsf{Avg}(\mathbf{B}_i^t)$.
- **RoLoRA** enables clients to alternate updating $\mathbf{A}$ and $\mathbf{B}$ as described in Section 4.

FlexLoRA (Bai et al., 2024) fine-tunes large models in a federated setting by aggregating the matrix products of LoRA components and compressing them using truncated-SVD. However, due to its significant memory and computation overheads it is not directly comparable with other schemes. Nonetheless, we include its performance in Table 6 in Appendix.

**Implementation & Configurations.** We implement all the methods based on FederatedScope-LLM (Kuang et al., 2023). We use NVIDIA GeForce RTX 4090 or NVIDIA A40 for all the experiments. To make a fair comparison, for each dataset, we obtain the best performance on test set and report the average over multiple seeds. Specifically, the learning rate is chosen from the set $\{5e-4, 1e-3, 2e-3, 5e-3, 1e-2, 2e-2, 5e-2, 1e-1\}$. Other hyper-parameters for experiments are specified in Table 4 in Appendix B.1. Please note that in all tasks, we compare the performance of the three methods *under the same number of communication rounds*.

### 6.1. Language Understanding Tasks

**Model and Datasets.** We take the pre-trained RoBERTa-Large (355M) (Liu et al., 2019) models from the Hugging-

Face Transformers library, and evaluate the performance of three federated finetuning methods on 5 datasets (SST-2, QNLI, MNLI, QQP, RTE) from the GLUE (Wang et al., 2019). GLUE benchmark is a comprehensive set of tasks for evaluating the performance of language models on a variety of natural language understanding tasks. Due to the limitation of the unpublished test set in GLUE, we follow the previous studies (Zhang et al., 2023b) and use the original validation set as the new test set and split a part of the training set as the validation set.

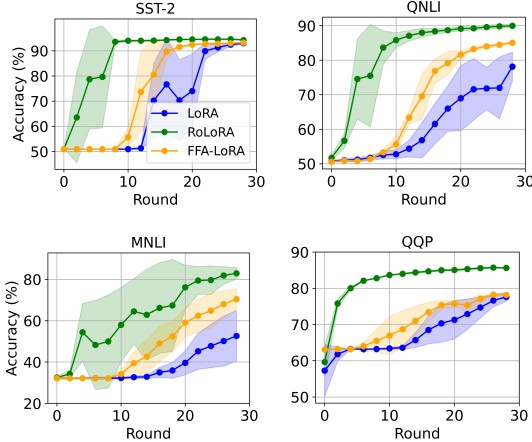

*Figure 3.* Accuracies over rounds with RoBERTa-Large models on SST-2, QNLI, MNLI, and QQP. The total number of clients is 50. We use rank 4.

**Effect of Number of Clients** In this section, we study the effect of the number of clients. The configurations are presented in Table 5 in Appendix. In Table 1, we increased the number of clients from 3 to 20, and then to 50, ensuring that there is no overlap in the training samples each client can access. Consequently, each client receives a smaller fraction of the total dataset. We observe that as the number of clients increases, while maintaining the same number of fine-tuning samples, the performance of the LoRA method significantly deteriorates for most datasets. In contrast, RoLoRA maintains its accuracy levels. The performance of FFA-LoRA also declines, attributed to the limited expressiveness of the random initialization of **A** for clients' heterogeneous data. Notably, RoLoRA achieves this accuracy while incurring only half the communication costs associated with LoRA. Figure 3 illustrates the dynamics during finetuning for three methods, highlighting that the convergence speed of RoLoRA is substantially better than that of the other two methods.

**Effect of Number of Finetuning Parameters** In Figure 4, we compare three methods across five GLUE datasets. We apply LoRA module to every weight matrix of the selected layers, given different budgets of LoRA parameters. For each dataset, we experiment with three budgets $(\mathcal{P}_1, \mathcal{P}_2, \mathcal{P}_3)$

ranging from low to high. The corresponding layer sets that are attached with LoRA adapters, $\mathcal{P}_1, \mathcal{P}_2, \mathcal{P}_3$, are detailed in Table 7 in Appendix B.1. The figures indicates that with sufficient number of finetuning parameters, FFA-LoRA can achieve comparable best accuracy as LoRA and RoLoRA, aligning with the results in (Sun et al., 2024); as the number of LoRA parameters is reduced, the performance of the three methods deteriorates to varying degrees. However, RoLoRA, which achieves performance comparable to LoRA, demonstrates greater robustness compared to FFA-LoRA, especially under conditions of limited fine-tuning parameters. It is important to note that with the same fine-tuning parameters, the communication cost of RoLoRA and FFA-LoRA is always half of that of LoRA due to their parameter freezing nature. This implies that RoLoRA not only sustains its performance but also enhances communication efficiency. Additional results of varifying ranks are provided in Figure 6, 7, and 8 in Appendix B.2.2.

**Align Communication Cost for Three Methods** In Figure 5, we conduct a comparison of three methods under the constraint of identical communication costs under the assumption that the number of clients is small. To align the communication costs across these methods, two approaches are considered. The first approach involves doubling the rank of FFA-LoRA and RoLoRA, with results presented in Appendix B.2.2. The second approach requires doubling the number of layers equipped with LoRA modules. In Figure 5, the latter strategy is employed. Specifically, for both FFA-LoRA and RoLoRA, we adjust the communication costs by doubling the number of layers equipped with LoRA modules, thereby standardizing the size of the transmitted messages. The configurations are presented in Table 8 in Appendix. Figure 5 demonstrates that when operating within a constrained communication cost budget, the performance of RoLoRA surpasses that of the other two methods for most of the tasks.

### 6.2. Commonsense Reasoning Tasks

**Model, Datasets.** We evaluate RoLoRA against FFA-LoRA and LoRA on Llama-2-7B(Touvron et al., 2023) for commonsense reasoning tasks. The commonsense reasoning tasks include 8 sub-tasks, each provided with predefined training and testing datasets. Following the setting in (Hu et al., 2021), we merge the training datasets from all 8 sub-tasks to create a unified training dataset, which is then evenly distributed among the clients. Evaluations are conducted individually on the testing dataset for each sub-task.

**Results** In Table 2, we compare the results of three methods with Llama-2-7B models on 8 commonsense reasoning tasks. The configurations are presented in Appendix B.2.4. The performance is reported as the mean accuracy with standard deviations across 3 trials. RoLoRA consistently

| rank | Clients num | Method | SST-2 | QNLI | MNLI | QQP | RTE | Avg. |
|---|---|---|---|---|---|---|---|---|
| 4 | 3 | LoRA | $95.62_{\pm0.17}$ | $91.59_{\pm0.21}$ | $86.20_{\pm0.05}$ | $86.13_{\pm0.10}$ | $81.46_{\pm1.22}$ | 88.20 |
| | | FFA-LoRA | $95.18_{\pm0.09}$ | $91.35_{\pm0.32}$ | $84.58_{\pm0.21}$ | $85.50_{\pm0.25}$ | $81.10_{\pm0.33}$ | 87.48 |
| | | RoLoRA | $95.49_{\pm0.16}$ | $91.64_{\pm0.30}$ | $85.70_{\pm0.04}$ | $86.14_{\pm0.06}$ | $82.43_{\pm0.84}$ | 88.28 |
| 4 | 20 | LoRA | $94.3_{\pm0.27}$ | $86.67_{\pm2.02}$ | $78.55_{\pm7.31}$ | $83.1_{\pm0.04}$ | $51.87_{\pm3.24}$ | 78.90 |
| | | FFA-LoRA | $93.88_{\pm0.06}$ | $89.11_{\pm0.19}$ | $80.99_{\pm1.74}$ | $83.92_{\pm0.2}$ | $57.16_{\pm1.46}$ | 80.01 |
| | | RoLoRA | $94.88_{\pm0.18}$ | $90.35_{\pm0.37}$ | $85.28_{\pm1.04}$ | $85.83_{\pm0.1}$ | $78.82_{\pm1.7}$ | 87.03 |
| 4 | 50 | LoRA | $93.00_{\pm0.35}$ | $78.13_{\pm5.13}$ | $52.64_{\pm15.07}$ | $77.60_{\pm1.47}$ | $52.23_{\pm1.1}$ | 70.72 |
| | | FFA-LoRA | $93.23_{\pm0.12}$ | $85.05_{\pm0.34}$ | $69.97_{\pm5.57}$ | $78.44_{\pm0.41}$ | $55.72_{\pm1.99}$ | 76.48 |
| | | RoLoRA | $94.80_{\pm0.17}$ | $90.00_{\pm0.63}$ | $82.98_{\pm3.36}$ | $85.71_{\pm0.18}$ | $75.57_{\pm2.88}$ | 85.81 |
| 8 | 50 | LoRA | $93.00_{\pm0.23}$ | $79.87_{\pm1.52}$ | $56.96_{\pm2.02}$ | $77.45_{\pm1.97}$ | $53.79_{\pm6.57}$ | 64.03 |
| | | FFA-LoRA | $92.74_{\pm0.13}$ | $83.69_{\pm0.75}$ | $64.51_{\pm1.92}$ | $79.71_{\pm2.04}$ | $53.07_{\pm1.3}$ | 72.46 |
| | | RoLoRA | $94.53_{\pm0.17}$ | $90.1_{\pm0.45}$ | $85.17_{\pm0.41}$ | $85.25_{\pm0.13}$ | $76.3_{\pm4.9}$ | 86.27 |

*Table 1.* Results with RoBERTa-Large models with varying client numbers (3, 20, 50), maintaining a constant sample count during fine-tuning.

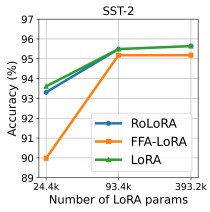 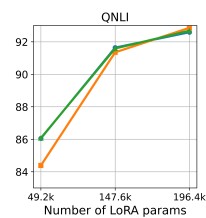 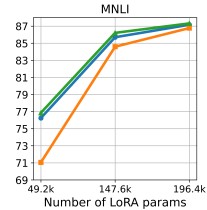 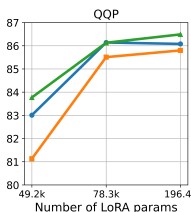 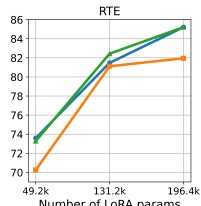

*Figure 4.* Results with RoBERTa-Large models on GLUE under different fine-tuning parameter budgets, involving three clients with rank 4.

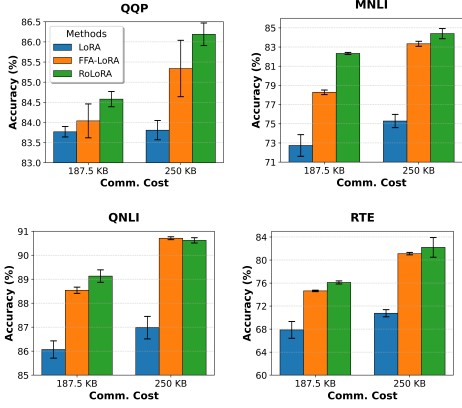

*Figure 5.* RoBERTa-Large accuracies on QQP, MNLI, QNLI, and RTE with specific uplink communication budget. It involves 3 clients using rank 4. Error bars reflect standard deviations.

achieves the highest accuracy across all tasks, demonstrating significant improvements over both LoRA and FFA-LoRA. We also highlights that FFA-LoRA exhibits large performance variances across trials, such as a standard deviation of 9.55 for PIQA and 8.44 for SIQA, respectively. This significant variability is likely due to the initialization quality of parameter $\mathbf{A}$, as different initializations could lead to varying optimization trajectories and final performance

outcomes as discussed in Section 5. Additional results on this task are presented in Table 10 in Appendix B.2.4.

| | BoolQ | PIQA | SIQA | HellaSwag |
|---|---|---|---|---|
| LoRA | $61.42_{\pm0.29}$ | $33.19_{\pm9.8}$ | $31.88_{\pm3.95}$ | $21.23_{\pm2.82}$ |
| FFA-LoRA | $53.43_{\pm4.3}$ | $35.49_{\pm9.55}$ | $10.63_{\pm8.44}$ | $11.81_{\pm4.53}$ |
| RoLoRA | $61.83_{\pm0.22}$ | $61.26_{\pm3.3}$ | $39.76_{\pm0.41}$ | $27.49_{\pm2.34}$ |
| | WinoGrande | ARC-e | ARC-c | OBQA |
| LoRA | $31.36_{\pm5.02}$ | $27.36_{\pm0.89}$ | $32.03_{\pm1.14}$ | $26.07_{\pm2.32}$ |
| FFA-LoRA | $1.61_{\pm2.14}$ | $6.88_{\pm0.42}$ | $7.93_{\pm0.89}$ | $15.0_{\pm5.41}$ |
| RoLoRA | $47.67_{\pm0.75}$ | $33.19_{\pm1.29}$ | $40.13_{\pm1.73}$ | $31.67_{\pm1.4}$ |

*Table 2.* Results with Llama-2-7B models on commonsense reasoning tasks. This involves 50 clients using rank 8.

**More results.** We include more experimental results on Llama-2-7B on HumanEval and MMLU tasks in Appendix B.2.5.

# 7. Conclusion

In this work, we introduce RoLoRA, a federated framework that leverages alternating optimization to finetune LoRA adapters. Our approach highlights the role of learning down-projection matrices to enhance both expressiveness and robustness. Through theoretical analysis on a simplified linear model, and comprehensive experiments on a toy neural network and large language models like RoBERTa-Large and Llama-2-7B, we show that RoLoRA outperforms existing methods that limit model updates or expressiveness.

## Impact Statement

This paper presents work whose goal is to advance the field of Machine Learning. There are many potential societal consequences of our work, none of which we feel must be specifically highlighted here.

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

## A. Theoretical Analysis

### A.1. Notation

Table 3 provides a summary of the symbols used throughout this theoretical analysis.

| Notation | Description |
|---|---|
| $\mathbf{a}^*, \mathbf{b}_i^*$ | Ground truth parameters of client $i$ |
| $\bar{\mathbf{b}}^*$ | Average of $\mathbf{b}_i^*$ |
| $\mathbf{a}^t, \mathbf{b}^t$ | Global model parameters of $t$-th iteration |
| $\delta_t$ | The angle distance between $\mathbf{a}^*$ and $\mathbf{a}^t$, $\lvert \sin\theta(\mathbf{a}^*, \mathbf{a}^t) \rvert$ |
| $\eta$ | Step size |
| $\mathbf{I_d}$ | $d \times d$ identity matrix |
| $\lVert \cdot \rVert$ | $l_2$ norm of a vector |
| $\lVert \cdot \rVert_{op}$ | Operator norm ($l_2$ norm) of a matrix |
| $\lvert \cdot \rvert$ | Absolute value of a scalar |
| $\lVert \cdot \rVert_{\psi_2}$ | Sub-Gaussian norm of a sub-Gaussian random variable |
| $\lVert \cdot \rVert_{\psi_1}$ | Sub-exponential norm of a sub-exponential random variable |
| $N$ | Total number of clients |
| $\hat{\mathbf{a}}^+$ | Updated $\mathbf{a}$ by gradient descent |
| $\hat{\mathbf{a}}$ | Normalized $\hat{\mathbf{a}}^+$ |
| $\bar{\mathbf{b}}$ | Average of $\mathbf{b}_i$ |

*Table 3.* Notations

### A.2. Auxiliary

**Definition A.1** (Sub-Gaussian Norm)**.** The sub-Gaussian norm of a random variable $\xi$, denoted as $\lVert\xi\rVert_{\psi_2}$, is defined as:

$$\lVert\xi\rVert_{\psi_2} = \inf\{t > 0 : \mathbb{E}[\exp(\xi^2/t^2)] \leq 2\}.$$

A random variable is said to be *sub-Gaussian* if its sub-Gaussian norm is finite. Gaussian random variables are sub-Gaussian. The sub-Gaussian norm of a standard Gaussian random variable $\xi \sim \mathcal{N}(0,1)$ is $\lVert\xi\rVert_{\psi_2} = \sqrt{8/3}$.

**Definition A.2** (Sub-Exponential Norm)**.** The sub-exponential norm of a random variable $\xi$, denoted as $\lVert\xi\rVert_{\psi_1}$, is defined as:

$$\lVert\xi\rVert_{\psi_1} = \inf\{t > 0 : \mathbb{E}[\exp(\lvert\xi\rvert/t)] \leq 2\}.$$

A random variable is said to be *sub-exponential* if its sub-exponential norm is finite.

**Lemma A.3** (The product of sub-Gaussians is sub-exponential)**.** *Let $\xi$ and $\upsilon$ be sub-Gaussian random variables. Then $\xi\upsilon$ is sub-exponential. Moreover,*

$$\lVert\xi\upsilon\rVert_{\psi_1} \leq \lVert\xi\rVert_{\psi_2} \cdot \lVert\upsilon\rVert_{\psi_2}$$

**Lemma A.4** (Sum of independent sub-Gaussians)**.** *Let $X_1, \cdots, X_N$ be independent mean-zero sub-Gaussian random variables. Then $\sum_{i=1}^N X_i$ is also sub-Gaussian with*

$$\left\lVert \sum_{i=1}^N X_i \right\rVert_{\psi_2}^2 \leq C \sum_{i=1}^N \lVert X_i \rVert_{\psi_2}^2,$$

*where $C$ is some absolute constant.*

*Proof.* See proof of Lemma 2.6.1 of (Vershynin, 2018). $\qquad\square$

**Corollary A.5.** *For random vector $\mathbf{x} \in \mathbb{R}^d$ with entries being independent standard Gaussian random variables, the inner product $\mathbf{a}^\top \mathbf{x}$ is sub-Gaussian for any fixed $\mathbf{a} \in \mathbb{R}^d$, and*

$$\left\lVert \mathbf{a}^\top \mathbf{x} \right\rVert_{\psi_2} \leq C' \lVert\mathbf{a}\rVert$$

*where $C'$ is some absolute constant.*

*Proof.* Note that $\mathbf{a}^\top \mathbf{x} = \sum_{i=1}^d a_i \xi_i$, where $\xi_i \sim \mathcal{N}(0,1)$ is the $i$-th entry of the random vector $\mathbf{x}$. Choose $C$ to be the absolute constant specified in Lemma A.4 for standard Gaussian random variables, and set $C' = \sqrt{8C/3}$. We have

$$\left\| \mathbf{a}^\top \mathbf{x} \right\|_{\psi_2}^2 \leq C \sum_{i=1}^N \|a_i \xi_i\|_{\psi_2}^2 \overset{(a)}{=} C \sum_{i=1}^N a_i^2 \|\xi_i\|_{\psi_2}^2 \overset{(b)}{=} \frac{8}{3} \cdot C \|\mathbf{a}\|^2 \;\Rightarrow\; \left\| \mathbf{a}^\top \mathbf{x} \right\|_{\psi_2} \leq \sqrt{\frac{8C}{3}} \|\mathbf{a}\| = C' \|\mathbf{a}\|.$$

Step (a) makes use of the homogeneity of the sub-Gaussian norm, and step (b) uses the fact that $\|\xi\|_{\psi_2} = \sqrt{8/3}$ for $\xi \sim \mathcal{N}(0,1)$. $\qquad\square$

**Definition A.6** ($\epsilon$-net). Consider a subset $\mathcal{A} \subseteq \mathbb{R}^d$ in the $d$-dimensional Euclidean space. Let $\epsilon > 0$. A subset $\mathcal{N} \subseteq \mathcal{A}$ is called an $\epsilon$-net of $\mathcal{A}$ if every point of $\mathcal{A}$ is within a distance $\epsilon$ of some point in $\mathcal{N}$, i.e.,

$$\forall \mathbf{x} \in \mathcal{A}, \; \exists \mathbf{x}' \in \mathcal{N}, \; \|\mathbf{x} - \mathbf{x}'\| \leq \epsilon.$$

**Lemma A.7** (Computing the operator norm on a net). *Let $\mathbf{a} \in \mathbb{R}^d$ and $\epsilon \in [0,1)$. Then, for any $\epsilon$-net $\mathcal{N}$ of the sphere $\mathcal{S}^{d-1}$, we have*

$$\|\mathbf{a}\| \leq \frac{1}{1-\epsilon} \sup_{\mathbf{x} \in \mathcal{N}} \langle \mathbf{a}, \mathbf{x} \rangle$$

*Proof.* See proof of Lemma 4.4.1 of (Vershynin, 2018). $\qquad\square$

**Theorem A.8** (Bernstein's inequality). *Let $X_1, \ldots, X_N$ be independent mean-zero sub-exponential random variables. Then, for every $t \geq 0$, we have*

$$\mathbb{P}\left( \left| \sum_{i=1}^N X_i \right| \geq t \right) \leq 2 \exp\left( -c \min\left( \frac{t^2}{\sum_{i=1}^N \|X_i\|_{\psi_1}^2}, \frac{t}{\max_i \|X_i\|_{\psi_1}} \right) \right),$$

*where $c > 0$ is an absolute constant.*

*Proof.* See proof of Theorem 2.8.1 of (Vershynin, 2018). $\qquad\square$

### A.3. Vector-vector case with homogeneous clients

Theorem 5.4 follows by recursively applying Lemma 5.3 for $T$ iterations. In Lemma A.9 and Lemma A.10, we start by obtaining the important bounds that will be reused. Using Lemma A.9 and Lemma A.10, based on the update rule of $\mathbf{a}$, we analyze the convergence behavior of $\mathbf{a}$ in Lemma 5.3.

**Lemma A.9.** *Let $\mathbf{a} = \mathbf{a}^t$. Let $\delta^t = \|(\mathbf{I}_d - \mathbf{a}^* \mathbf{a}^{*\top})\mathbf{a}\| = \|(\mathbf{I}_d - \mathbf{aa}^\top)\mathbf{a}^*\|$ denote the angle distance between $\mathbf{a}^*$ and $\mathbf{a}$. Let $\mathbf{g}^\top = \mathbf{a}^\top \mathbf{a}^* \mathbf{b}^{*\top}, \bar{\mathbf{b}} = \frac{1}{N} \sum_{i=1}^N \mathbf{b}_i$, If $m = \Omega(q)$, and $q = \max\left( \frac{\log(N)}{[\min(\epsilon_1, \epsilon_2)]^2}, \frac{d \log(\frac{2}{\epsilon_0})}{\epsilon_2^2} \right)$, then with probability $1 - q^{-10}$,*

$$\|\bar{\mathbf{b}} - \mathbf{g}\| \leq \epsilon' \delta^t \|\mathbf{b}^*\| \tag{13}$$

*where $\epsilon' = \frac{\epsilon_2}{(1-\epsilon_0)(1-\epsilon_1)}$, for $\epsilon_0, \epsilon_1, \epsilon_2 \in (0,1)$.*

*Proof.* We drop superscript $t$ for simplicity. Following Algorithm 2, we start by computing the update for $\tilde{\mathbf{b}}_i$. With $\mathbf{g}^\top = \mathbf{a}^\top \mathbf{a}^* \mathbf{b}^{*\top}$, we get:

$$\tilde{\mathbf{b}}_i^\top = \frac{\mathbf{a}^\top \mathbf{X}_i^\top \mathbf{X}_i \mathbf{a}^* \mathbf{b}^{*\top}}{\mathbf{a}^\top \mathbf{X}_i^\top \mathbf{X}_i \mathbf{a}} \tag{14}$$

$$= \frac{\mathbf{a}^\top \mathbf{X}_i^\top \mathbf{X}_i \mathbf{aa}^\top \mathbf{a}^* \mathbf{b}^{*\top} + \mathbf{a}^\top \mathbf{X}_i^\top \mathbf{X}_i (\mathbf{I}_d - \mathbf{aa}^\top) \mathbf{a}^* \mathbf{b}^{*\top}}{\mathbf{a}^\top \mathbf{X}_i^\top \mathbf{X}_i \mathbf{a}} \tag{15}$$

$$= \mathbf{g}^\top + \frac{\mathbf{a}^\top \mathbf{X}_i^\top \mathbf{X}_i (\mathbf{I}_d - \mathbf{aa}^\top) \mathbf{a}^* \mathbf{b}^{*\top}}{\mathbf{a}^\top \mathbf{X}_i^\top \mathbf{X}_i \mathbf{a}}. \tag{16}$$

Therefore,

$$\|\tilde{\mathbf{b}}_i - \mathbf{g}\| \leq |\mathbf{a}^\top \mathbf{X}_i^\top \mathbf{X}_i \mathbf{a}|^{-1} \cdot \|\mathbf{a}^\top \mathbf{X}_i^\top \mathbf{X}_i (\mathbf{I}_d - \mathbf{a}\mathbf{a}^\top) \mathbf{a}^* \mathbf{b}^{*\top}\| = \|\mathbf{X}_i \mathbf{a}\|^{-2} \cdot \|\mathbf{a}^\top \mathbf{X}_i^\top \mathbf{X}_i (\mathbf{I}_d - \mathbf{a}\mathbf{a}^\top) \mathbf{a}^* \mathbf{b}^{*\top}\|. \quad (17)$$

Note that since each entry of $\mathbf{X}_i$ is independent and identically distributed according to a standard Gaussian, and $\|\mathbf{a}\| = 1$, $\mathbf{X}_i \mathbf{a}$ is a random standard Gaussian vector. By Theorem 3.1.1 of (Vershynin, 2018), the following is true for any $\epsilon_1 \in (0, 1)$

$$\mathbb{P}\left\{\|\mathbf{X}_i \mathbf{a}\|^2 \leq (1 - \epsilon_1)m\right\} \leq \exp\left(-\frac{c_1 \epsilon_1^2 m}{K^4}\right) \quad (18)$$

where $K = \|\xi\|_{\psi_2} \geq 1$ for $\xi \sim \mathcal{N}(0, 1)$ and $c_1$ is some large absolute constant that makes (18) holds. Next we upper bound $\|\mathbf{a}^\top \mathbf{X}_i^\top \mathbf{X}_i (\mathbf{I}_d - \mathbf{a}\mathbf{a}^\top) \mathbf{a}^* \mathbf{b}^{*\top}\|$. Note that $\mathbb{E}[\mathbf{a}^\top \mathbf{X}_i^\top \mathbf{X}_i (\mathbf{I}_d - \mathbf{a}\mathbf{a}^\top) \mathbf{a}^* \mathbf{b}^{*\top}] = \mathbf{a}^\top \mathbb{E}[\mathbf{X}_i^\top \mathbf{X}_i] (\mathbf{I}_d - \mathbf{a}\mathbf{a}^\top) \mathbf{a}^* \mathbf{b}^{*\top} = m\mathbf{a}^\top (\mathbf{I}_d - \mathbf{a}\mathbf{a}^\top) \mathbf{a}^* \mathbf{b}^{*\top} = 0$. First we need to apply sub-exponential Berstein inequality to bound the deviation from this mean, and then apply epsilon net argument. Let $\mathcal{N}$ be any $\epsilon_0$-net of the unit sphere $\mathcal{S}^{d-1}$ in the $d$-dimensional real Euclidean space, then by Lemma A.7, we have

$$\|\mathbf{a}^\top \mathbf{X}_i^\top \mathbf{X}_i (\mathbf{I}_d - \mathbf{a}\mathbf{a}^\top) \mathbf{a}^* \mathbf{b}^{*\top}\| \leq \frac{1}{1 - \epsilon_0} \max_{\mathbf{w} \in \mathcal{N}} \mathbf{a}^\top \mathbf{X}_i^\top \mathbf{X}_i (\mathbf{I}_d - \mathbf{a}\mathbf{a}^\top) \mathbf{a}^* \mathbf{b}^{*\top} \mathbf{w} \quad (19)$$

$$\leq \frac{1}{1 - \epsilon_0} \max_{\mathbf{w} \in \mathcal{N}} |\mathbf{a}^\top \mathbf{X}_i^\top \mathbf{X}_i (\mathbf{I}_d - \mathbf{a}\mathbf{a}^\top) \mathbf{a}^* \mathbf{b}^{*\top} \mathbf{w}| \quad (20)$$

Meanwhile, denote the $j$-th row of $\mathbf{X}_i$ by $\mathbf{x}_{i,j}^\top$, for every $\mathbf{w} \in \mathcal{N}$, we have

$$\mathbf{a}^\top \mathbf{X}_i^\top \mathbf{X}_i (\mathbf{I}_d - \mathbf{a}\mathbf{a}^\top) \mathbf{a}^* \mathbf{b}^{*\top} \mathbf{w} = \sum_{j=1}^m (\mathbf{a}^\top \mathbf{x}_{i,j})(\mathbf{x}_{i,j}^\top (\mathbf{I}_d - \mathbf{a}\mathbf{a}^\top) \mathbf{a}^* \mathbf{b}^{*\top} \mathbf{w}) \quad (21)$$

On the right hand side of (21), $\mathbf{a}^\top \mathbf{x}_{i,j}$ and $\mathbf{x}_{i,j}^\top (\mathbf{I}_d - \mathbf{a}\mathbf{a}^\top) \mathbf{a}^* \mathbf{b}^{*\top} \mathbf{w}$ are sub-Gaussian random variables. Thus, the summands on the right-hand side of (21) are products of sub-Gaussian random variables, making them sub-exponential. Now by choosing $c_2 = (C')^2$ for the $C'$ in Corollary A.5, we have the following chain of inequalities for all $i, j$:

$$\|(\mathbf{a}^\top \mathbf{x}_{i,j})(\mathbf{x}_{i,j}^\top (\mathbf{I}_d - \mathbf{a}\mathbf{a}^\top) \mathbf{a}^* \mathbf{b}^{*\top} \mathbf{w})\|_{\psi_1} \leq \|\mathbf{a}^\top \mathbf{x}_{i,j}\|_{\psi_2} \cdot \|\mathbf{x}_{i,j}^\top (\mathbf{I}_d - \mathbf{a}\mathbf{a}^\top) \mathbf{a}^* \mathbf{b}^{*\top} \mathbf{w}\|_{\psi_2} \quad (22)$$

$$\leq c_2 \cdot \|\mathbf{a}\| \cdot \|(\mathbf{I}_d - \mathbf{a}\mathbf{a}^\top) \mathbf{a}^* \mathbf{b}^{*\top} \mathbf{w}\| \quad (23)$$

$$\leq c_2 \cdot \|\mathbf{a}\| \cdot \|(\mathbf{I}_d - \mathbf{a}\mathbf{a}^\top) \mathbf{a}^* \mathbf{b}^{*\top}\|_{op} \|\mathbf{w}\| \quad (24)$$

$$\leq c_2 \cdot \|(\mathbf{I}_d - \mathbf{a}\mathbf{a}^\top) \mathbf{a}^* \mathbf{b}^{*\top}\|_{op} \quad (25)$$

Equation (22) is due to Lemma A.3, (23) is due to Corollary A.5, (25) is by the fact that $\|\mathbf{a}\| = \|\mathbf{w}\| = 1$.

Furthermore, these summands are mutually independent and have zero mean. By applying sub-exponential Bernstein's inequality (Theorem A.8) with $t = \epsilon_2 m \|(\mathbf{I}_d - \mathbf{a}\mathbf{a}^\top) \mathbf{a}^* \mathbf{b}^{*\top}\|_{op}$, we get

$$\mathbb{P}\left\{\left|\mathbf{a}^\top \mathbf{X}_i^\top \mathbf{X}_i (\mathbf{I}_d - \mathbf{a}\mathbf{a}^\top) \mathbf{a}^* \mathbf{b}^{*\top} \mathbf{w}\right| \geq \epsilon_2 m \|(\mathbf{I}_d - \mathbf{a}\mathbf{a}^\top) \mathbf{a}^* \mathbf{b}^{*\top}\|_{op}\right\} \quad (26)$$

$$= \mathbb{P}\left\{\left|\sum_{j=1}^m (\mathbf{a}^\top \mathbf{x}_{i,j})(\mathbf{x}_{i,j}^\top (\mathbf{I}_d - \mathbf{a}\mathbf{a}^\top) \mathbf{a}^* \mathbf{b}^{*\top} \mathbf{w})\right| \geq \epsilon_2 m \|(\mathbf{I}_d - \mathbf{a}\mathbf{a}^\top) \mathbf{a}^* \mathbf{b}^{*\top}\|_{op}\right\} \quad (27)$$

$$\leq 2\exp\left(-c\min\left(\frac{\epsilon_2^2 m^2 \|(\mathbf{I}_d - \mathbf{a}\mathbf{a}^\top) \mathbf{a}^* \mathbf{b}^{*\top}\|_{op}^2}{\sum_{j=1}^m \|(\mathbf{a}^\top \mathbf{x}_{i,j})(\mathbf{x}_{i,j}^\top (\mathbf{I}_d - \mathbf{a}\mathbf{a}^\top) \mathbf{a}^* \mathbf{b}^{*\top} \mathbf{w})\|_{\psi_1}^2}, \frac{\epsilon_2 m \|(\mathbf{I}_d - \mathbf{a}\mathbf{a}^\top) \mathbf{a}^* \mathbf{b}^{*\top}\|_{op}}{\max_j \|(\mathbf{a}^\top \mathbf{x}_{i,j})(\mathbf{x}_{i,j}^\top (\mathbf{I}_d - \mathbf{a}\mathbf{a}^\top) \mathbf{a}^* \mathbf{b}^{*\top} \mathbf{w})\|_{\psi_1}}\right)\right) \quad (28)$$

$$= 2\exp\left(-c_3 \epsilon_2^2 m\right) \quad (29)$$

for any fixed $\mathbf{w} \in \mathcal{N}$, $\epsilon_2 \in (0, 1)$, and some absolute constant $c_3$. (29) follows because

$$\frac{\epsilon_2^2 m^2 \|(\mathbf{I}_d - \mathbf{a}\mathbf{a}^\top) \mathbf{a}^* \mathbf{b}^{*\top}\|_{op}^2}{\sum_{j=1}^m \|(\mathbf{a}^\top \mathbf{x}_{i,j})(\mathbf{x}_{i,j}^\top (\mathbf{I}_d - \mathbf{a}\mathbf{a}^\top) \mathbf{a}^* \mathbf{b}^{*\top} \mathbf{w})\|_{\psi_1}^2} \geq \frac{\epsilon_2^2 m}{c_2^2} \quad (30)$$

$$\frac{\epsilon_2 m \|(\mathbf{I}_d - \mathbf{a}\mathbf{a}^\top) \mathbf{a}^* \mathbf{b}^{*\top}\|_{op}}{\max_j \|(\mathbf{a}^\top \mathbf{x}_{i,j})(\mathbf{x}_{i,j}^\top (\mathbf{I}_d - \mathbf{a}\mathbf{a}^\top) \mathbf{a}^* \mathbf{b}^{*\top} \mathbf{w})\|_{\psi_1}} \geq \frac{\epsilon_2 m}{c_2} \quad (31)$$

And $\frac{\epsilon_2^2 m}{c_2^2} \leq \frac{\epsilon_2 m}{c_2}$. Now we apply union bound over all elements in $\mathcal{N}$. By Corollary 4.2.13 in (Vershynin, 2018), there exists an $\epsilon_0$-net $\mathcal{N}$ with $|\mathcal{N}| \leq (\frac{2}{\epsilon_0} + 1)^d$, therefore for this choice of $\mathcal{N}$,

$$\mathbb{P}\left\{\max_{\mathbf{w} \in \mathcal{N}} \left|\mathbf{a}^\top \mathbf{X}_i^\top \mathbf{X}_i (\mathbf{I}_d - \mathbf{a}\mathbf{a}^\top)\mathbf{a}^* \mathbf{b}^{*\top} \mathbf{w}\right| \geq \epsilon_2 m \|(\mathbf{I}_d - \mathbf{a}\mathbf{a}^\top)\mathbf{a}^* \mathbf{b}^{*\top}\|_{op}\right\} \tag{32}$$

$$\leq \sum_{\mathbf{w} \in \mathcal{N}} \mathbb{P}\left\{\left|\mathbf{a}^\top \mathbf{X}_i^\top \mathbf{X}_i (\mathbf{I}_d - \mathbf{a}\mathbf{a}^\top)\mathbf{a}^* \mathbf{b}^{*\top} \mathbf{w}\right| \geq \epsilon_2 m \|(\mathbf{I}_d - \mathbf{a}\mathbf{a}^\top)\mathbf{a}^* \mathbf{b}^{*\top}\|_{op}\right\} \tag{33}$$

$$\leq \left(\frac{2}{\epsilon_0} + 1\right)^d \cdot 2\exp\left(-c_3 \epsilon_2^2 m\right) \tag{34}$$

$$= 2\exp\left(d\log(1 + \tfrac{2}{\epsilon_0}) - c_3 \epsilon_2^2 m\right) \tag{35}$$

Combining (17),(18), (20), and (35), we get

$$\mathbb{P}\left\{\|\tilde{\mathbf{b}}_i - \mathbf{g}\| \leq \epsilon'\|(\mathbf{I}_d - \mathbf{a}\mathbf{a}^\top)\mathbf{a}^* \mathbf{b}^{*\top}\|_{op}\right\} \geq 1 - p_0 \tag{36}$$

where $\epsilon' = \frac{\epsilon_2}{(1-\epsilon_0)(1-\epsilon_1)}$ and $p_0 = 2\exp\left(d\log(1 + \frac{2}{\epsilon_0}) - c_3 \epsilon_2^2 m\right) + \exp\left(-\frac{c_1 \epsilon_1^2 m}{K^4}\right)$. Using a union bound over $i \in [N]$, we have

$$\mathbb{P}\left\{\bigcap_{i=1}^N \|\tilde{\mathbf{b}}_i - \mathbf{g}\| \leq \epsilon'\|(\mathbf{I}_d - \mathbf{a}\mathbf{a}^\top)\mathbf{a}^* \mathbf{b}^{*\top}\|_{op}\right\} \geq 1 - Np_0. \tag{37}$$

Next we bound $\|\bar{\mathbf{b}} - \mathbf{g}\|$ where $\bar{\mathbf{b}}$ is the average of $\{\mathbf{b}_i\}_{i=1}^N$.

$$\|\bar{\mathbf{b}} - \mathbf{g}\| = \|\frac{1}{N} \sum_{i=1}^N (\tilde{\mathbf{b}}_i - \mathbf{g})\| \tag{38}$$

$$\leq \frac{1}{N} \sum_{i=1}^N \|\tilde{\mathbf{b}}_i - \mathbf{g}\| \tag{39}$$

$$\leq \epsilon'\|(\mathbf{I}_d - \mathbf{a}\mathbf{a}^\top)\mathbf{a}^* \mathbf{b}^{*\top}\|_{op} \tag{40}$$

$$= \epsilon'\|(\mathbf{I}_d - \mathbf{a}\mathbf{a}^\top)\mathbf{a}^*\| \cdot \|\mathbf{b}^{*\top}\| \tag{41}$$

$$= \epsilon' \delta^t \|\mathbf{b}^*\| \tag{42}$$

with probability $1 - Np_0$. (39) follows by Jensen's inequality. (41) follows since $\|\mathbf{u}\mathbf{v}^\top\|_{op} = \|\mathbf{u}\| \cdot \|\mathbf{v}\|$. (42) follows since $\delta^t = \|(\mathbf{I}_d - \mathbf{a}\mathbf{a}^\top)\mathbf{a}^*\|$.

If $m = \Omega(q)$, where $q = \max(\frac{\log(N)}{[\min(\epsilon_1,\epsilon_2)]^2}, \frac{d\log(\frac{2}{\epsilon_0})}{\epsilon_2^2})$, then $1 - Np_0 > 1 - \exp(-Cq) > 1 - q^{-10}$ for large absolute constant $C$. Then with probability at least $1 - q^{-10}$,

$$\|\bar{\mathbf{b}} - \mathbf{g}\| \leq \epsilon' \delta^t \|\mathbf{b}^*\| \tag{43}$$

$\square$

**Lemma A.10.** *Let* $\mathbf{a} = \mathbf{a}^t$. *Let* $\delta^t = \|(\mathbf{I}_d - \mathbf{a}^* \mathbf{a}^{*\top})\mathbf{a}\| = \|(\mathbf{I}_d - \mathbf{a}\mathbf{a}^\top)\mathbf{a}^*\|$ *denote the angle distance between* $\mathbf{a}^*$ *and* $\mathbf{a}$. *Then for* $Nm = \Omega(\frac{d\log\frac{2}{\epsilon_0}}{\epsilon_3^2})$ *and* $q = \max(\frac{\log(N)}{[\min(\epsilon_1,\epsilon_2)]^2}, \frac{d\log(\frac{2}{\epsilon_0})}{\epsilon_2^2})$, *with probability at least* $1 - 2q^{-10}$,

$$\|\nabla_{\mathbf{a}} l(\mathbf{a}, \bar{\mathbf{b}}) - \mathbb{E}[\nabla_{\mathbf{a}} l(\mathbf{a}, \bar{\mathbf{b}})]\| \leq 2\tilde{\epsilon}((\epsilon')^2 + \epsilon')\delta^t \|\mathbf{b}^*\|^2 \tag{44}$$

*where* $\tilde{\epsilon} = \frac{\epsilon_3}{1-\epsilon_0}$, *and* $\epsilon' = \frac{\epsilon_2}{(1-\epsilon_0)(1-\epsilon_1)}$, *for* $\epsilon_0, \epsilon_1, \epsilon_2, \epsilon_3 \in (0,1)$.

*Proof.* Based on the loss function $l(\mathbf{a}, \mathbf{b}) = \frac{1}{N}\sum_{i=1}^{N} l_i(\mathbf{a}, \mathbf{b}) = \frac{1}{Nm}\sum_{i=1}^{N}\|\mathbf{X}_i\mathbf{a}^*\mathbf{b}^{*\top} - \mathbf{X}_i\mathbf{a}\mathbf{b}^\top\|^2$, we bound the expected gradient with respect to $\mathbf{a}$ and the deviation from it. The gradient with respect to $\mathbf{a}$ and its expectation are computed as:

$$\nabla_\mathbf{a} l(\mathbf{a}, \bar{\mathbf{b}}) = \frac{2}{Nm}\sum_{i=1}^{N}(\mathbf{X}_i^\top\mathbf{X}_i\mathbf{a}\bar{\mathbf{b}}^\top\bar{\mathbf{b}} - \mathbf{X}_i^\top\mathbf{Y}_i\bar{\mathbf{b}}) \tag{45}$$

$$= \frac{2}{Nm}\sum_{i=1}^{N}(\mathbf{X}_i^\top\mathbf{X}_i\mathbf{a}\bar{\mathbf{b}}^\top\bar{\mathbf{b}} - \mathbf{X}_i^\top\mathbf{X}_i\mathbf{a}^*\mathbf{b}^{*\top}\bar{\mathbf{b}}) \tag{46}$$

$$= \frac{2}{Nm}\sum_{i=1}^{N}\mathbf{X}_i^\top\mathbf{X}_i(\mathbf{a}\bar{\mathbf{b}}^\top - \mathbf{a}^*\mathbf{b}^{*\top})\bar{\mathbf{b}} \tag{47}$$

$$\mathbb{E}[\nabla_\mathbf{a} l(\mathbf{a}, \bar{\mathbf{b}})] = \frac{2}{Nm}\sum_{i=1}^{N}m(\mathbf{a}\bar{\mathbf{b}}^\top - \mathbf{a}^*\mathbf{b}^{*\top})\bar{\mathbf{b}} \tag{48}$$

$$= 2(\mathbf{a}\bar{\mathbf{b}}^\top - \mathbf{a}^*\mathbf{b}^{*\top})\bar{\mathbf{b}} \tag{49}$$

Next, we bound $\|\nabla_\mathbf{a} l(\mathbf{a}, \bar{\mathbf{b}}) - \mathbb{E}[\nabla_\mathbf{a} l(\mathbf{a}, \bar{\mathbf{b}})]\|$. Construct $\epsilon_0$-net $\mathcal{N}$ over $d$ dimensional unit spheres $\mathcal{S}^{d-1}$, by Lemma A.7, we have

$$\|\nabla_\mathbf{a} l(\mathbf{a}, \bar{\mathbf{b}}) - \mathbb{E}[\nabla_\mathbf{a} l(\mathbf{a}, \bar{\mathbf{b}})]\| \le \frac{1}{1-\epsilon_0}\max_{\mathbf{w}\in\mathcal{N}}\left|\mathbf{w}^\top\nabla_\mathbf{a} l(\mathbf{a}, \bar{\mathbf{b}}) - \mathbf{w}^\top\mathbb{E}[\nabla_\mathbf{a} l(\mathbf{a}, \bar{\mathbf{b}})]\right| \tag{50}$$

$$\le \frac{1}{1-\epsilon_0}\frac{2}{Nm}\max_{\mathbf{w}\in\mathcal{N}}\left|\sum_{i=1}^{N}\sum_{j=1}^{m}(\mathbf{x}_{i,j}^\top\mathbf{w})(\mathbf{x}_{i,j}(\mathbf{a}\bar{\mathbf{b}}^\top - \mathbf{a}^*\mathbf{b}^{*\top}))\bar{\mathbf{b}} - \mathbf{w}^\top(\mathbf{a}\bar{\mathbf{b}}^\top - \mathbf{a}^*\mathbf{b}^{*\top})\bar{\mathbf{b}}\right| \tag{51}$$

where $\mathbf{x}_{i,j}^\top$ is the $j$-th row of $\mathbf{X}_i$. Observe that $\mathbf{x}_{i,j}^\top\mathbf{w}$ and $\mathbf{x}_{i,j}(\mathbf{a}\bar{\mathbf{b}}^\top - \mathbf{a}^*\mathbf{b}^{*\top})\bar{\mathbf{b}}$ are sub-Gaussian variables. Thus the product of them are sub-exponentials. For the right hand side of (51), the summands are sub-exponential random variables with sub-exponential norm

$$\|(\mathbf{x}_{i,j}^\top\mathbf{w})(\mathbf{x}_{i,j}(\mathbf{a}\bar{\mathbf{b}}^\top - \mathbf{a}^*\mathbf{b}^{*\top}))\bar{\mathbf{b}} - \mathbf{w}^\top(\mathbf{a}\bar{\mathbf{b}}^\top - \mathbf{a}^*\mathbf{b}^{*\top})\bar{\mathbf{b}}\|_{\psi_1} \tag{52}$$

$$\le \|(\mathbf{x}_{i,j}^\top\mathbf{w})(\mathbf{x}_{i,j}(\mathbf{a}\bar{\mathbf{b}}^\top - \mathbf{a}^*\mathbf{b}^{*\top}))\bar{\mathbf{b}}\|_{\psi_1} + \|\mathbf{w}^\top(\mathbf{a}\bar{\mathbf{b}}^\top - \mathbf{a}^*\mathbf{b}^{*\top})\bar{\mathbf{b}}\|_{\psi_1} \tag{53}$$

$$\le \|(\mathbf{x}_{i,j}^\top\mathbf{w})(\mathbf{x}_{i,j}(\mathbf{a}\bar{\mathbf{b}}^\top - \mathbf{a}^*\mathbf{b}^{*\top}))\bar{\mathbf{b}}\|_{\psi_1} + \frac{|\mathbf{w}^\top(\mathbf{a}\bar{\mathbf{b}}^\top - \mathbf{a}^*\mathbf{b}^{*\top})\bar{\mathbf{b}}|}{\log 2} \tag{54}$$

$$\le c_2\cdot\|\mathbf{w}\|\cdot\|\mathbf{a}\bar{\mathbf{b}}^\top - \mathbf{a}^*\mathbf{b}^{*\top}\|_{op}\cdot\|\bar{\mathbf{b}}\| + \frac{|\mathbf{w}^\top(\mathbf{a}\bar{\mathbf{b}}^\top - \mathbf{a}^*\mathbf{b}^{*\top})\bar{\mathbf{b}}|}{\log 2} \tag{55}$$

$$\le c_2\cdot\|\mathbf{w}\|\cdot\|\mathbf{a}\bar{\mathbf{b}}^\top - \mathbf{a}^*\mathbf{b}^{*\top}\|_{op}\cdot\|\bar{\mathbf{b}}\| + \frac{\|\mathbf{w}\|\cdot\|(\mathbf{a}\bar{\mathbf{b}}^\top - \mathbf{a}^*\mathbf{b}^{*\top})\bar{\mathbf{b}}\|}{\log 2} \tag{56}$$

$$\le c_2\cdot\|\mathbf{w}\|\cdot\|\mathbf{a}\bar{\mathbf{b}}^\top - \mathbf{a}^*\mathbf{b}^{*\top}\|_{op}\cdot\|\bar{\mathbf{b}}\| + \frac{\|\mathbf{w}\|\cdot\|\mathbf{a}\bar{\mathbf{b}}^\top - \mathbf{a}^*\mathbf{b}^{*\top}\|_{op}\cdot\|\bar{\mathbf{b}}\|}{\log 2} \tag{57}$$

$$= c_4\cdot\|\mathbf{a}\bar{\mathbf{b}}^\top - \mathbf{a}^*\mathbf{b}^{*\top}\|_{op}\cdot\|\bar{\mathbf{b}}\| \tag{58}$$

where $c_4 = c_2 + \frac{1}{\log 2}$ is some absolute constant greater than 1. Equation (54) is due to the fact that for a constant $c \in \mathbb{R}$,

$$\|c\|_{\psi_1} = \inf_t \exp\left\{\frac{|c|}{t} \le 2\right\} = \frac{|c|}{\log 2}.$$

Equation (55) is derived similarly as (22)-(24).

The summands in (51) are mutually independent and have zero mean. Applying sub-exponential Bernstein inequality

(Theorem A.8) with $t = \epsilon_3 Nm \|\mathbf{a}\bar{\mathbf{b}}^\top - \mathbf{a}^*\mathbf{b}^{*\top}\|_{op} \cdot \|\bar{\mathbf{b}}\|$,

$$\mathbb{P}\left\{\left|\sum_{i=1}^{N}\sum_{j=1}^{m}[((\mathbf{x}_{i,j}^\top \mathbf{w})(\mathbf{x}_{i,j}(\mathbf{a}\bar{\mathbf{b}}^\top - \mathbf{a}^*\mathbf{b}^{*\top})) - \mathbf{w}^\top(\mathbf{a}\bar{\mathbf{b}}^\top - \mathbf{a}^*\mathbf{b}^{*\top}))\bar{\mathbf{b}}]\right| \geq \epsilon_3 Nm\|\mathbf{a}\bar{\mathbf{b}}^\top - \mathbf{a}^*\mathbf{b}^{*\top}\|_{op} \cdot \|\bar{\mathbf{b}}\|\right\} \quad (59)$$

$$\leq 2\exp\left(-c\min\left(\frac{\epsilon_3^2 Nm}{c_4^2}, \frac{\epsilon_3 Nm}{c_4}\right)\right) \quad (60)$$

$$= 2\exp\left(-c_5\epsilon_3^2 Nm\right) \quad (61)$$

for any fixed $\mathbf{w} \in \mathcal{N}$, $\epsilon_3 \in (0,1)$ and some absolute constant $c_5$.

Now we apply union bound over all $\mathbf{w} \in \mathcal{N}$ using Corollary 4.2.13 of (Vershynin, 2018). We can conclude that

$$\mathbb{P}\left\{\max_{\mathbf{w}\in\mathcal{N}}\left|\mathbf{w}^\top\nabla_{\mathbf{a}}l(\mathbf{a},\bar{\mathbf{b}}) - \mathbf{w}^\top\mathbb{E}[\nabla_{\mathbf{a}}l(\mathbf{a},\bar{\mathbf{b}})]\right| \geq 2\epsilon_3\|\mathbf{a}\bar{\mathbf{b}}^\top - \mathbf{a}^*\mathbf{b}^{*\top}\|_{op} \cdot \|\bar{\mathbf{b}}\|\right\} \quad (62)$$

$$\leq \sum_{\mathbf{w}\in\mathcal{N}}\mathbb{P}\left\{\left|\mathbf{w}^\top\nabla_{\mathbf{a}}l(\mathbf{a},\bar{\mathbf{b}}) - \mathbf{w}^\top\mathbb{E}[\nabla_{\mathbf{a}}l(\mathbf{a},\bar{\mathbf{b}})]\right| \geq 2\epsilon_3\|\mathbf{a}\bar{\mathbf{b}}^\top - \mathbf{a}^*\mathbf{b}^{*\top}\|_{op} \cdot \|\bar{\mathbf{b}}\|\right\} \quad (63)$$

$$\leq 2\exp\left(d\log(1 + \frac{2}{\epsilon_0}) - c_5\epsilon_3^2 Nm\right) \quad (64)$$

Combining (42), (50), and (62), with probability at least $1 - 2\exp\left(d\log(1 + \frac{2}{\epsilon_0}) - c_5\epsilon_3^2 Nm\right) - q^{-10}$,

$$\|\nabla_{\mathbf{a}}l(\mathbf{a},\bar{\mathbf{b}}) - \mathbb{E}[\nabla_{\mathbf{a}}l(\mathbf{a},\bar{\mathbf{b}})]\| \leq \frac{1}{1-\epsilon_0}\max_{\mathbf{w}\in\mathcal{N}}\left|\mathbf{w}^\top\nabla_{\mathbf{a}}l(\mathbf{a},\bar{\mathbf{b}}) - \mathbf{w}^\top\mathbb{E}[\nabla_{\mathbf{a}}l(\mathbf{a},\bar{\mathbf{b}})]\right| \quad (65)$$

$$\leq \frac{2\epsilon_3}{1-\epsilon_0}\|\mathbf{a}\bar{\mathbf{b}}^\top - \mathbf{a}^*\mathbf{b}^{*\top}\|_{op} \cdot \|\bar{\mathbf{b}}\| \quad (66)$$

$$= \frac{2\epsilon_3}{1-\epsilon_0}\|\mathbf{a}(\bar{\mathbf{b}} - \mathbf{g})^\top - (\mathbf{I}_d - \mathbf{a}\mathbf{a}^\top)\mathbf{a}^*\mathbf{b}^{*\top}\|_{op} \cdot \|\bar{\mathbf{b}}\| \quad (67)$$

$$\leq \frac{2\epsilon_3}{1-\epsilon_0}(\|\mathbf{a}^\top(\bar{\mathbf{b}} - \mathbf{g})\| + \|(\mathbf{I}_d - \mathbf{a}\mathbf{a}^\top)\mathbf{a}^*\mathbf{b}^{*\top}\|_{op})\|\bar{\mathbf{b}}\| \quad (68)$$

$$\leq \frac{2\epsilon_3}{1-\epsilon_0}(\|\bar{\mathbf{b}} - \mathbf{g}\| + \|(\mathbf{I}_d - \mathbf{a}\mathbf{a}^\top)\mathbf{a}^*\| \cdot \|\mathbf{b}^*\|)\|\bar{\mathbf{b}}\| \quad (69)$$

$$= \frac{2\epsilon_3}{1-\epsilon_0}(\|\bar{\mathbf{b}} - \mathbf{g}\| + \delta^t\|\mathbf{b}^*\|)\|\bar{\mathbf{b}} - \mathbf{g} + \mathbf{g}\| \quad (70)$$

$$\leq \frac{2\epsilon_3}{1-\epsilon_0}(\|\bar{\mathbf{b}} - \mathbf{g}\| + \delta^t\|\mathbf{b}^*\|)(\|\bar{\mathbf{b}} - \mathbf{g}\| + \|\mathbf{g}\|) \quad (71)$$

$$\leq \frac{2\epsilon_3}{1-\epsilon_0}(\|\bar{\mathbf{b}} - \mathbf{g}\| + \delta^t\|\mathbf{b}^*\|)(\|\bar{\mathbf{b}} - \mathbf{g}\| + \|\mathbf{b}^*\|) \quad (72)$$

$$= \frac{2\epsilon_3}{1-\epsilon_0}(\|\bar{\mathbf{b}} - \mathbf{g}\|^2 + \delta^t\|\bar{\mathbf{b}} - \mathbf{g}\|\|\mathbf{b}^*\| + \|\bar{\mathbf{b}} - \mathbf{g}\|\|\mathbf{b}^*\| + \delta^t\|\mathbf{b}^*\|^2) \quad (73)$$

$$\leq \frac{2\epsilon_3}{1-\epsilon_0}((\epsilon')^2(\delta^t)^2 + \epsilon'(\delta^t)^2 + \epsilon'\delta^t + \delta^t)\|\mathbf{b}^*\|^2 \quad (74)$$

$$\leq \frac{2\epsilon_3}{1-\epsilon_0}(\epsilon' + 1)^2\delta^t\|\mathbf{b}^*\|^2 \quad (75)$$

$$= 2\tilde{\epsilon}(\epsilon' + 1)^2\delta^t\|\mathbf{b}^*\|^2 \quad (76)$$

with $\tilde{\epsilon} = \frac{\epsilon_3}{1-\epsilon_0}$. (66) uses (62). (68) follows by triangle inequality. (70) follows by $\delta^t = \|(\mathbf{I}_d - \mathbf{a}\mathbf{a}^\top)\mathbf{a}^*\|$. (72) uses $\|\mathbf{g}\| = \|\mathbf{b}^*\mathbf{a}^{*\top}\mathbf{a}\| \leq \|\mathbf{b}^*\|$. (75) follows by $(\delta^t)^2 < \delta^t$ since $\delta^t \in (0,1)$.

If $Nm = \Omega(\frac{d\log(\frac{2}{\epsilon_0})}{\epsilon_3^2})$, then existing large constant $C$,

$$1 - 2\exp\left(d\log(1 + \frac{2}{\epsilon_0}) - c_5\epsilon_3^2 Nm\right) - q^{-10} > 1 - \exp(-Cd) - q^{-10} \tag{77}$$

$$> 1 - d^{-10} - q^{-10} \tag{78}$$

$$> 1 - 2q^{-10} \tag{79}$$

Thus with probability at least $1 - 2q^{-10}$, (76) holds. $\qquad\square$

**Lemma A.11** (Lemma 5.3). *Let* $\mathbf{a} = \mathbf{a}^t$. *Let* $\delta^t = \|(\mathbf{I}_d - \mathbf{a}^*\mathbf{a}^{*\top})\mathbf{a}\| = \|(\mathbf{I}_d - \mathbf{a}\mathbf{a}^\top)\mathbf{a}^*\|$ *denote the angle distance between* $\mathbf{a}^*$ *and* $\mathbf{a}$. *Assume that Assumption 5.1 holds and* $\delta^t \leq \delta^{t-1} \leq \cdots \leq \delta^0$. *Let* $m$ *be the number of samples for each updating step, let* $\epsilon' = \frac{\epsilon_2}{(1-\epsilon_0)(1-\epsilon_1)}, \tilde{\epsilon} = \frac{\epsilon_3}{1-\epsilon_0}$ *for* $\epsilon_0, \epsilon_1, \epsilon_2, \epsilon_3 \in (0,1)$, *if*

$$m = \Omega\left(\max\left\{\frac{\log(N)}{[\min(\epsilon_1, \epsilon_2)]^2}, \frac{d\log(\frac{2}{\epsilon_0})}{\epsilon_2^2}\right\}\right)$$

*and* $\epsilon', \tilde{\epsilon} < \frac{1-(\delta^0)^2}{16}$, *for any* $t$ *and* $\eta \leq \frac{1}{L_{max}^2}$, *then we have,*

$$\delta^{t+1} \leq \delta^t\sqrt{1 - \eta(1 - \delta^{02})\|\mathbf{b}^*\|^2} \tag{80}$$

*with probability at least* $1 - 2q^{-10}$ *for* $q = \max\left\{\frac{\log(N)}{[\min(\epsilon_1, \epsilon_2)]^2}, \frac{d\log(\frac{2}{\epsilon_0})}{\epsilon_2^2}\right\}$.

*Proof.* Recall that $\hat{\mathbf{a}}^+ = \mathbf{a} - \eta\nabla_{\mathbf{a}}l(\mathbf{a}, \bar{\mathbf{b}})$. We substract and add $\mathbb{E}[\nabla_{\mathbf{a}}l(\mathbf{a}, \bar{\mathbf{b}})]$, obtain

$$\hat{\mathbf{a}}^+ = \mathbf{a} - \eta\mathbb{E}[\nabla_{\mathbf{a}}l(\mathbf{a}, \bar{\mathbf{b}})] + \eta(\mathbb{E}[\nabla_{\mathbf{a}}l(\mathbf{a}, \bar{\mathbf{b}})] - \nabla_{\mathbf{a}}l(\mathbf{a}, \bar{\mathbf{b}})) \tag{81}$$

Multiply both sides by the projection operator $\mathbf{P} = \mathbf{I}_d - \mathbf{a}^*(\mathbf{a}^*)^\top$,

$$\mathbf{P}\hat{\mathbf{a}}^+ = \mathbf{P}\mathbf{a} - \eta\mathbf{P}\mathbb{E}[\nabla_{\mathbf{a}}l(\mathbf{a}, \bar{\mathbf{b}})] + \eta\mathbf{P}(\mathbb{E}[\nabla_{\mathbf{a}}l(\mathbf{a}, \bar{\mathbf{b}})] - \nabla_{\mathbf{a}}l(\mathbf{a}, \bar{\mathbf{b}})) \tag{82}$$

$$= \mathbf{P}\mathbf{a} - 2\eta\mathbf{P}(\mathbf{a}\bar{\mathbf{b}}^\top - \mathbf{a}^*\mathbf{b}^{*\top})\bar{\mathbf{b}} + \eta\mathbf{P}(\mathbb{E}[\nabla_{\mathbf{a}}l(\mathbf{a}, \bar{\mathbf{b}})] - \nabla_{\mathbf{a}}l(\mathbf{a}, \bar{\mathbf{b}})) \tag{83}$$

$$= \mathbf{P}\mathbf{a} - 2\eta\mathbf{P}\mathbf{a}\bar{\mathbf{b}}^\top\bar{\mathbf{b}} + \eta\mathbf{P}(\mathbb{E}[\nabla_{\mathbf{a}}l(\mathbf{a}, \bar{\mathbf{b}})] - \nabla_{\mathbf{a}}l(\mathbf{a}, \bar{\mathbf{b}})) \tag{84}$$

$$= \mathbf{P}\mathbf{a}(1 - 2\eta\bar{\mathbf{b}}^\top\bar{\mathbf{b}}) + \eta\mathbf{P}(\mathbb{E}[\nabla_{\mathbf{a}}l(\mathbf{a}, \bar{\mathbf{b}})] - \nabla_{\mathbf{a}}l(\mathbf{a}, \bar{\mathbf{b}})) \tag{85}$$

where (83) uses $\mathbb{E}[\nabla_{\mathbf{a}}l(\mathbf{a}, \bar{\mathbf{b}})] = 2(\mathbf{a}\bar{\mathbf{b}}^\top - \mathbf{a}^*\mathbf{b}^{*\top})\bar{\mathbf{b}}$, (84) follows by $\mathbf{P}\mathbf{a}^* = 0$. Thus, we get

$$\|\mathbf{P}\hat{\mathbf{a}}^+\| \leq \|\mathbf{P}\mathbf{a}\||1 - 2\eta\bar{\mathbf{b}}^\top\bar{\mathbf{b}}| + \eta\|(\mathbb{E}[\nabla_{\mathbf{a}}l(\mathbf{a}, \bar{\mathbf{b}})] - \nabla_{\mathbf{a}}l(\mathbf{a}, \bar{\mathbf{b}}))\| \tag{86}$$

Normalizing the left hand side, we obtain

$$\frac{\|\mathbf{P}\hat{\mathbf{a}}^+\|}{\|\hat{\mathbf{a}}^+\|} \leq \frac{\|\mathbf{P}\mathbf{a}\||1 - 2\eta\bar{\mathbf{b}}^\top\bar{\mathbf{b}}| + \eta\|(\mathbb{E}[\nabla_{\mathbf{a}}l(\mathbf{a}, \bar{\mathbf{b}})] - \nabla_{\mathbf{a}}l(\mathbf{a}, \bar{\mathbf{b}}))\|}{\|\hat{\mathbf{a}}^+\|} \tag{87}$$

$$\Rightarrow \delta^{t+1} \leq \frac{\delta^t|1 - 2\eta\bar{\mathbf{b}}^\top\bar{\mathbf{b}}| + \eta\|\mathbb{E}[\nabla_{\mathbf{a}}l(\mathbf{a}, \bar{\mathbf{b}})] - \nabla_{\mathbf{a}}l(\mathbf{a}, \bar{\mathbf{b}})\|}{\|\hat{\mathbf{a}}^+\|} \tag{88}$$

$$= \frac{E_1 + E_2}{\|\hat{\mathbf{a}}^+\|} \tag{89}$$

where (87) follows by $\delta^{t+1} = \frac{\|\mathbf{P}\hat{\mathbf{a}}^+\|}{\|\hat{\mathbf{a}}^+\|}$ and $\delta^t = \|\mathbf{P}\mathbf{a}\|$. We need to upper bound $E_1$ and $E_2$ accordingly. $E_2$ is upper bounded based on Lemma A.10. With probability at least $1 - 2q^{-10}$,

$$E_2 = \eta\|\mathbb{E}[\nabla_{\mathbf{a}}l(\mathbf{a}, \bar{\mathbf{b}})] - \nabla_{\mathbf{a}}l(\mathbf{a}, \bar{\mathbf{b}})\| \tag{90}$$

$$\leq 2\eta\tilde{\epsilon}(\epsilon' + 1)^2\delta^t\|\mathbf{b}^*\|^2 \tag{91}$$

To upper bound $E_1$, we need to lower bound $\|\bar{\mathbf{b}}\|^2$. We can first lower bound $\|\bar{\mathbf{b}}\|$ by:

$$\|\bar{\mathbf{b}}\| = \|\mathbf{g} - (\mathbf{g} - \bar{\mathbf{b}})\| \tag{92}$$

$$\geq \|\mathbf{g}\| - \|\mathbf{g} - \bar{\mathbf{b}}\| \tag{93}$$

$$= \sqrt{1 - (\delta^t)^2}\|\mathbf{b}^*\| - \|\mathbf{g} - \bar{\mathbf{b}}\| \tag{94}$$

$$\geq \sqrt{1 - (\delta^t)^2}\|\mathbf{b}^*\| - \epsilon'\delta^t\|\mathbf{b}^*\| \tag{95}$$

with probability at least $1 - q^{-10}$. (94) follows by $\mathbf{g}^\top = \mathbf{a}^\top\mathbf{a}^*\mathbf{b}^{*\top}$ and $\mathbf{a}^\top\mathbf{a}^* = \cos\theta(\mathbf{a}, \mathbf{a}^*)$, (95) follows by Lemma A.9. Assuming $\delta^t \leq \cdots \leq \delta^0$, we choose $\epsilon' < \frac{1-(\delta^0))^2}{16}$ to make $\sqrt{1 - (\delta^t)^2}\|\mathbf{b}^*\| - \epsilon'\delta^t\|\mathbf{b}^*\| \geq 0$. Hence $\|\bar{\mathbf{b}}\|^2$ is lower bounded by:

$$\|\bar{\mathbf{b}}\|^2 \geq (\sqrt{1 - (\delta^t)^2}\|\mathbf{b}^*\| - \epsilon'\delta^t\|\mathbf{b}^*\|)^2 \tag{96}$$

$$= (1 - (\delta^t)^2)\|\mathbf{b}^*\|^2 + (\epsilon')^2(\delta^t)^2\|\mathbf{b}^*\|^2 - 2\epsilon'\delta^t\sqrt{1 - (\delta^t)^2}\|\mathbf{b}^*\|^2 \tag{97}$$

$$\geq (1 - (\delta^t)^2)\|\mathbf{b}^*\|^2 + (\epsilon')^2(\delta^t)^2\|\mathbf{b}^*\|^2 - \epsilon'\|\mathbf{b}^*\|^2 \tag{98}$$

$$\geq (1 - (\delta^0)^2)\|\mathbf{b}^*\|^2 - \epsilon'\|\mathbf{b}^*\|^2 \tag{99}$$

with probability at least $1 - q^{-10}$. (98) follows by $xy \leq \frac{1}{2}$ for $x^2 + y^2 = 1$, (99) follows by assuming $\delta^t \leq \delta^{t-1} \leq \cdots \leq \delta^0$. $E_1$ is upper bounded below.

$$E_1 = \delta^t|1 - 2\eta\bar{\mathbf{b}}^\top\bar{\mathbf{b}}| \tag{100}$$

$$\leq \delta^t|1 - 2\eta((1 - (\delta^0)^2) - \epsilon')\|\mathbf{b}^*\|^2| \tag{101}$$

with probability at least $1 - q^{-10}$. Next we lower bound $\|\hat{\mathbf{a}}^+\|$.

$$\|\hat{\mathbf{a}}^+\|^2 = \|\mathbf{a} - \eta\nabla_{\mathbf{a}}l(\mathbf{a}, \bar{\mathbf{b}})\|^2 \tag{102}$$

$$= \mathbf{a}^\top\mathbf{a} + \eta^2\|\nabla_{\mathbf{a}}l(\mathbf{a}, \bar{\mathbf{b}})\|^2 - 2\eta\mathbf{a}^\top\nabla_{\mathbf{a}}l(\mathbf{a}, \bar{\mathbf{b}}) \tag{103}$$

$$\geq \mathbf{a}^\top\mathbf{a} - 2\eta\mathbf{a}^\top\nabla_{\mathbf{a}}l(\mathbf{a}, \bar{\mathbf{b}}) \tag{104}$$

$$= 1 - 2\eta\mathbf{a}^\top\nabla_{\mathbf{a}}l(\mathbf{a}, \bar{\mathbf{b}}) \tag{105}$$

$$= 1 - 2\eta\mathbf{a}^\top(\nabla_{\mathbf{a}}l(\mathbf{a}, \bar{\mathbf{b}}) - \mathbb{E}[\nabla_{\mathbf{a}}l(\mathbf{a}, \bar{\mathbf{b}})]) - 2\eta\mathbf{a}^\top\mathbb{E}[\nabla_{\mathbf{a}}l(\mathbf{a}, \bar{\mathbf{b}})] \tag{106}$$

where (104) follows by $\eta^2\|\nabla_{\mathbf{a}}l(\mathbf{a}, \bar{\mathbf{b}})\|^2 \geq 0$, and (105) follows by $\mathbf{a}^\top\mathbf{a} = 1$. The first subtrahend $2\eta\mathbf{a}^\top(\nabla_{\mathbf{a}}l(\mathbf{a}, \bar{\mathbf{b}}) - \mathbb{E}[\nabla_{\mathbf{a}}l(\mathbf{a}, \bar{\mathbf{b}})])$ is upper bounded such that

$$2\eta\mathbf{a}^\top(\nabla_{\mathbf{a}}l(\mathbf{a}, \bar{\mathbf{b}}) - \mathbb{E}[\nabla_{\mathbf{a}}l(\mathbf{a}, \bar{\mathbf{b}})]) \leq 2\eta\|\mathbf{a}\| \cdot \|(\nabla_{\mathbf{a}}l(\mathbf{a}, \bar{\mathbf{b}}) - \mathbb{E}[\nabla_{\mathbf{a}}l(\mathbf{a}, \bar{\mathbf{b}})])\| \tag{107}$$

$$= 2\eta\|(\nabla_{\mathbf{a}}l(\mathbf{a}, \bar{\mathbf{b}}) - \mathbb{E}[\nabla_{\mathbf{a}}l(\mathbf{a}, \bar{\mathbf{b}})])\| \tag{108}$$

$$\leq 4\eta\tilde{\epsilon}(\epsilon' + 1)^2\|\mathbf{b}^*\|^2 \tag{109}$$

with probability at least $1 - 2q^{-10}$. (109) uses Lemma A.9. The second subtrahend is upper bounded such that

$$2\eta\mathbf{a}^\top\mathbb{E}[\nabla_{\mathbf{a}}l(\mathbf{a}, \bar{\mathbf{b}})] = 4\eta\mathbf{a}^\top(\mathbf{a}\bar{\mathbf{b}}^\top - \mathbf{a}^*\mathbf{b}^{*\top})\bar{\mathbf{b}} \tag{110}$$

$$= 4\eta\mathbf{a}^\top(\mathbf{a}\bar{\mathbf{b}}^\top - \mathbf{a}^*\mathbf{b}^{*\top})\mathbf{g} - 4\eta\mathbf{a}^\top(\mathbf{a}\bar{\mathbf{b}}^\top - \mathbf{a}^*\mathbf{b}^{*\top})(\mathbf{g} - \bar{\mathbf{b}}) \tag{111}$$

where $\mathbf{a}^\top(\mathbf{a}\bar{\mathbf{b}}^\top - \mathbf{a}^*\mathbf{b}^{*\top})\mathbf{g} = -\mathbf{a}^\top((\mathbf{I}_d - \mathbf{a}\mathbf{a}^\top)\mathbf{a}^*\mathbf{b}^{*\top} + \mathbf{a}(\mathbf{g} - \bar{\mathbf{b}})^\top)\mathbf{g} = (\bar{\mathbf{b}} - \mathbf{g})^\top\mathbf{g}$. The second term is simplified via $\mathbf{a}^\top(\mathbf{a}\bar{\mathbf{b}}^\top - \mathbf{a}^*\mathbf{b}^{*\top})(\mathbf{g} - \bar{\mathbf{b}}) = \mathbf{a}^\top((\mathbf{I}_d - \mathbf{a}\mathbf{a}^\top)\mathbf{a}^*\mathbf{b}^{*\top} + \mathbf{a}(\mathbf{g} - \bar{\mathbf{b}})^\top)(\bar{\mathbf{b}} - \mathbf{g}) = -(\mathbf{g} - \bar{\mathbf{b}})^2$. Both simplifications use $\mathbf{a}^\top(\mathbf{I}_d - \mathbf{a}\mathbf{a}^\top) = 0$ and $\mathbf{a}^\top\mathbf{a} = 1$. (111) becomes

$$2\eta\mathbf{a}^\top\mathbb{E}[\nabla_{\mathbf{a}}l(\mathbf{a}, \bar{\mathbf{b}})] = 4\eta(\bar{\mathbf{b}} - \mathbf{g})^\top\mathbf{g} + 4\eta(\mathbf{g} - \bar{\mathbf{b}})^2 \tag{112}$$

$$\leq 4\eta\|\mathbf{g} - \bar{\mathbf{b}}\|\|\mathbf{b}^*\| + 4\eta\|\mathbf{g} - \bar{\mathbf{b}}\|^2 \tag{113}$$

$$\leq 4\eta\epsilon'\delta^t\|\mathbf{b}^*\|^2 + 4\eta(\epsilon')^2(\delta^t)^2\|\mathbf{b}^*\|^2 \tag{114}$$

$$\leq 4\eta((\epsilon')^2 + \epsilon')\|\mathbf{b}^*\|^2 \tag{115}$$

with probability at least $1 - q^{-10}$. (158) uses Lemma A.9. Combining (109) and (115), we obtain

$$\|\hat{\mathbf{a}}^+\|^2 \geq 1 - 4\eta\tilde{\epsilon}(\epsilon' + 1)^2\|\mathbf{b}^*\|^2 - 4\eta((\epsilon')^2 + \epsilon')\|\mathbf{b}^*\|^2 \tag{116}$$

with probability at least $1 - 2q^{-10}$. Combining (101), (91) and (116), we obtain

$$\delta^{t+1} \leq \frac{E_1 + E_2}{\|\hat{\mathbf{a}}^+\|} \leq \frac{\delta^t(1 - 2\eta((1 - (\delta^0)^2) - \epsilon')\|\mathbf{b}^*\|^2 + 2\eta\tilde{\epsilon}(\epsilon' + 1)^2\|\mathbf{b}^*\|^2)}{\sqrt{1 - 4\eta\tilde{\epsilon}(\epsilon' + 1)^2\|\mathbf{b}^*\|^2 - 4\eta((\epsilon')^2 + \epsilon')\|\mathbf{b}^*\|^2}} = \delta^t C \tag{117}$$

We can choose $\epsilon', \tilde{\epsilon} < \frac{1-(\delta^0)^2}{16}$ such that $(1 - (\delta^0)^2) > \max(4(\tilde{\epsilon}(\epsilon' + 1)^2 + (\epsilon')^2 + \epsilon'), 2\epsilon' + 2\tilde{\epsilon}(\epsilon' + 1)^2)$ holds. Then we obtain

$$C = \frac{1 - 2\eta((1 - (\delta^0)^2) - \epsilon')\|\mathbf{b}^*\|^2 + 2\eta\tilde{\epsilon}(\epsilon' + 1)^2\|\mathbf{b}^*\|^2}{\sqrt{1 - 4\eta\tilde{\epsilon}(\epsilon' + 1)^2\|\mathbf{b}^*\|^2 - 4\eta((\epsilon')^2 + \epsilon')\|\mathbf{b}^*\|^2}} \tag{118}$$

$$= \frac{1 - 2\eta(1 - (\delta^0)^2)\|\mathbf{b}^*\|^2 + 2\eta\epsilon'\|\mathbf{b}^*\|^2 + 2\eta\tilde{\epsilon}(\epsilon' + 1)^2\|\mathbf{b}^*\|^2}{\sqrt{1 - 4\eta(\tilde{\epsilon}(\epsilon' + 1)^2 + (\epsilon')^2 + \epsilon')\|\mathbf{b}^*\|^2}} \tag{119}$$

$$\leq \frac{1 - 2\eta(1 - (\delta^0)^2)\|\mathbf{b}^*\|^2 + \eta(2\epsilon' + 2\tilde{\epsilon}(\epsilon' + 1)^2)\|\mathbf{b}^*\|^2}{\sqrt{1 - 4\eta(\tilde{\epsilon}(\epsilon' + 1)^2 + (\epsilon')^2 + \epsilon')\|\mathbf{b}^*\|^2}} \tag{120}$$

$$\leq \frac{1 - \eta(1 - (\delta^0)^2)\|\mathbf{b}^*\|^2}{\sqrt{1 - \eta(1 - (\delta^0)^2)\|\mathbf{b}^*\|^2}} \tag{121}$$

$$= \sqrt{1 - \eta(1 - (\delta^0)^2)\|\mathbf{b}^*\|^2} \tag{122}$$

Assuming $\eta \leq \frac{1}{L_{max}^2} \leq \frac{1}{\|\mathbf{b}^*\|^2}$, $1 - \eta(1 - (\delta^0)^2)\|\mathbf{b}^*\|^2$ is strictly positive. Therefore we obtain, with probability at least $1 - 2q^{-10}$,

$$\delta^{t+1} \leq \delta^t\sqrt{1 - \eta(1 - (\delta^0)^2)\|\mathbf{b}^*\|^2}. \tag{123}$$

$\square$

## A.4. Proof of Theorem 5.4

*Proof.* In Lemma 5.3, we have shown the angle distance between $\mathbf{a}$ and $\mathbf{a}^*$ decreasing in $t$-th iteration such that with probability at least $1 - 2q^{-10}$ for $q = \max\{\log(N), d\}$, $\delta^{t+1} \leq \delta^t C$ for $c \in (0, 1), C = \sqrt{1 - c(1 - (\delta^0)^2)}$ with proper choice of step size $\eta$.

**Proving $\delta^1 \leq \delta^0 C$** . Now we are to prove that for the first iteration, $\delta^1 \leq \delta^0 C$ with certain probability.

By Lemma A.9, we get $\|\bar{\mathbf{b}} - \mathbf{g}\| \leq \epsilon'\delta^0\|\mathbf{b}^*\|$ with probability at least $1 - q^{-10}$.

By Lemma A.10, we get $\|\nabla_{\mathbf{a}}l(\mathbf{a}, \bar{\mathbf{b}}) - \mathbb{E}[\nabla_{\mathbf{a}}l(\mathbf{a}, \bar{\mathbf{b}})]\| \leq 2\tilde{\epsilon}((\epsilon')^2 + \epsilon')\delta^0\|\mathbf{b}^*\|^2$ with probability at least $1 - 2q^{-10}$.

We drop superscript of the iteration index for simplicity. Recall that $\hat{\mathbf{a}}^+ = \mathbf{a} - \eta\nabla_{\mathbf{a}}l(\mathbf{a}, \bar{\mathbf{b}})$. We substract and add $\mathbb{E}[\nabla_{\mathbf{a}}l(\mathbf{a}, \bar{\mathbf{b}})]$, obtain

$$\hat{\mathbf{a}}^+ = \mathbf{a} - \eta\mathbb{E}[\nabla_{\mathbf{a}}l(\mathbf{a}, \bar{\mathbf{b}})] + \eta(\mathbb{E}[\nabla_{\mathbf{a}}l(\mathbf{a}, \bar{\mathbf{b}})] - \nabla_{\mathbf{a}}l(\mathbf{a}, \bar{\mathbf{b}})) \tag{124}$$

Multiply both sides by the projection operator $\mathbf{P} = \mathbf{I}_d - \mathbf{a}^*(\mathbf{a}^*)^\top$,

$$\mathbf{P}\hat{\mathbf{a}}^+ = \mathbf{Pa} - \eta\mathbf{P}\mathbb{E}[\nabla_{\mathbf{a}}l(\mathbf{a}, \bar{\mathbf{b}})] + \eta\mathbf{P}(\mathbb{E}[\nabla_{\mathbf{a}}l(\mathbf{a}, \bar{\mathbf{b}})] - \nabla_{\mathbf{a}}l(\mathbf{a}, \bar{\mathbf{b}})) \tag{125}$$

$$= \mathbf{Pa} - 2\eta\mathbf{P}(\mathbf{a}\bar{\mathbf{b}}^\top - \mathbf{a}^*\mathbf{b}^{*\top})\bar{\mathbf{b}} + \eta\mathbf{P}(\mathbb{E}[\nabla_{\mathbf{a}}l(\mathbf{a}, \bar{\mathbf{b}})] - \nabla_{\mathbf{a}}l(\mathbf{a}, \bar{\mathbf{b}})) \tag{126}$$

$$= \mathbf{Pa} - 2\eta\mathbf{Pa}\bar{\mathbf{b}}^\top\bar{\mathbf{b}} + \eta\mathbf{P}(\mathbb{E}[\nabla_{\mathbf{a}}l(\mathbf{a}, \bar{\mathbf{b}})] - \nabla_{\mathbf{a}}l(\mathbf{a}, \bar{\mathbf{b}})) \tag{127}$$

$$= \mathbf{Pa}(1 - 2\eta\bar{\mathbf{b}}^\top\bar{\mathbf{b}}) + \eta\mathbf{P}(\mathbb{E}[\nabla_{\mathbf{a}}l(\mathbf{a}, \bar{\mathbf{b}})] - \nabla_{\mathbf{a}}l(\mathbf{a}, \bar{\mathbf{b}})) \tag{128}$$

where (126) uses $\mathbb{E}[\nabla_\mathbf{a} l(\mathbf{a}, \bar{\mathbf{b}})] = 2(\mathbf{a}\bar{\mathbf{b}}^\top - \mathbf{a}^* \mathbf{b}^{*^\top})\bar{\mathbf{b}}$, (127) follows by $\mathbf{P}\mathbf{a}^* = 0$. Thus, we get

$$\|\mathbf{P}\hat{\mathbf{a}}^+\| \leq \|\mathbf{P}\mathbf{a}\| |1 - 2\eta\bar{\mathbf{b}}^\top\bar{\mathbf{b}}| + \eta\|(\mathbb{E}[\nabla_\mathbf{a} l(\mathbf{a}, \bar{\mathbf{b}})] - \nabla_\mathbf{a} l(\mathbf{a}, \bar{\mathbf{b}}))\| \tag{129}$$

Normalizing the left hand side, we obtain

$$\frac{\|\mathbf{P}\hat{\mathbf{a}}^+\|}{\|\hat{\mathbf{a}}^+\|} \leq \frac{\|\mathbf{P}\mathbf{a}\| |1 - 2\eta\bar{\mathbf{b}}^\top\bar{\mathbf{b}}| + \eta\|(\mathbb{E}[\nabla_\mathbf{a} l(\mathbf{a}, \bar{\mathbf{b}})] - \nabla_\mathbf{a} l(\mathbf{a}, \bar{\mathbf{b}}))\|}{\|\hat{\mathbf{a}}^+\|} \tag{130}$$

$$\Rightarrow \delta^1 \leq \frac{\delta^0 |1 - 2\eta\bar{\mathbf{b}}^\top\bar{\mathbf{b}}| + \eta\|\mathbb{E}[\nabla_\mathbf{a} l(\mathbf{a}, \bar{\mathbf{b}})] - \nabla_\mathbf{a} l(\mathbf{a}, \bar{\mathbf{b}})\|}{\|\hat{\mathbf{a}}^+\|} \tag{131}$$

$$= \frac{E_1 + E_2}{\|\hat{\mathbf{a}}^+\|} \tag{132}$$

where (130) follows by $\delta^1 = \frac{\|\mathbf{P}\hat{\mathbf{a}}^+\|}{\|\hat{\mathbf{a}}^+\|}$ and $\delta^0 = \|\mathbf{P}\mathbf{a}\|$. We need to upper bound $E_1$ and $E_2$ accordingly. $E_2$ is upper bounded based on Lemma A.10. With probability at least $1 - 2q^{-10}$,

$$E_2 = \eta\|\mathbb{E}[\nabla_\mathbf{a} l(\mathbf{a}, \bar{\mathbf{b}})] - \nabla_\mathbf{a} l(\mathbf{a}, \bar{\mathbf{b}})\| \tag{133}$$

$$\leq 2\eta\tilde{\epsilon}(\epsilon' + 1)^2 \delta^0 \|\mathbf{b}^*\|^2 \tag{134}$$

To upper bound $E_1$, we need to lower bound $\|\bar{\mathbf{b}}\|^2$. We can first lower bound $\|\bar{\mathbf{b}}\|$ by:

$$\|\bar{\mathbf{b}}\| = \|\mathbf{g} - (\mathbf{g} - \bar{\mathbf{b}})\| \tag{135}$$

$$\geq \|\mathbf{g}\| - \|\mathbf{g} - \bar{\mathbf{b}}\| \tag{136}$$

$$= \sqrt{1 - (\delta^0)^2}\|\mathbf{b}^*\| - \|\mathbf{g} - \bar{\mathbf{b}}\| \tag{137}$$

$$\geq \sqrt{1 - (\delta^0)^2}\|\mathbf{b}^*\| - \epsilon'\delta^0\|\mathbf{b}^*\| \tag{138}$$

with probability at least $1 - q^{-10}$. (137) follows by $\mathbf{g}^\top = \mathbf{a}^\top \mathbf{a}^* \mathbf{b}^{*^\top}$ and $\mathbf{a}^\top \mathbf{a}^* = \cos\theta(\mathbf{a}, \mathbf{a}^*)$, (138) follows by Lemma A.9. Still we choose $\epsilon' < \frac{1 - (\delta^0)^2}{16}$ to make $\sqrt{1 - (\delta^0)^2}\|\mathbf{b}^*\| - \epsilon'\delta^0\|\mathbf{b}^*\| \geq 0$. Hence $\|\bar{\mathbf{b}}\|^2$ is lower bounded by:

$$\|\bar{\mathbf{b}}\|^2 \geq (\sqrt{1 - (\delta^0)^2}\|\mathbf{b}^*\| - \epsilon'\delta^0\|\mathbf{b}^*\|)^2 \tag{139}$$

$$= (1 - (\delta^0)^2)\|\mathbf{b}^*\|^2 + (\epsilon')^2(\delta^0)^2\|\mathbf{b}^*\|^2 - 2\epsilon'\delta^0\sqrt{1 - (\delta^0)^2}\|\mathbf{b}^*\|^2 \tag{140}$$

$$\geq (1 - (\delta^0)^2)\|\mathbf{b}^*\|^2 + (\epsilon')^2(\delta^0)^2\|\mathbf{b}^*\|^2 - \epsilon'\|\mathbf{b}^*\|^2 \tag{141}$$

$$\geq (1 - (\delta^0)^2)\|\mathbf{b}^*\|^2 - \epsilon'\|\mathbf{b}^*\|^2 \tag{142}$$

with probability at least $1 - q^{-10}$. (141) follows by $xy \leq \frac{1}{2}$ for $x^2 + y^2 = 1$. $E_1$ is upper bounded below.

$$E_1 = \delta^0 |1 - 2\eta\bar{\mathbf{b}}^\top\bar{\mathbf{b}}| \tag{143}$$

$$\leq \delta^0 |1 - 2\eta((1 - (\delta^0)^2) - \epsilon')\|\mathbf{b}^*\|^2| \tag{144}$$

with probability at least $1 - q^{-10}$. Next we lower bound $\|\hat{\mathbf{a}}^+\|$.

$$\|\hat{\mathbf{a}}^+\|^2 = \|\mathbf{a} - \eta\nabla_\mathbf{a} l(\mathbf{a}, \bar{\mathbf{b}})\|^2 \tag{145}$$

$$= \mathbf{a}^\top\mathbf{a} + \eta^2\|\nabla_\mathbf{a} l(\mathbf{a}, \bar{\mathbf{b}})\|^2 - 2\eta\mathbf{a}^\top\nabla_\mathbf{a} l(\mathbf{a}, \bar{\mathbf{b}}) \tag{146}$$

$$\geq \mathbf{a}^\top\mathbf{a} - 2\eta\mathbf{a}^\top\nabla_\mathbf{a} l(\mathbf{a}, \bar{\mathbf{b}}) \tag{147}$$

$$= 1 - 2\eta\mathbf{a}^\top\nabla_\mathbf{a} l(\mathbf{a}, \bar{\mathbf{b}}) \tag{148}$$

$$= 1 - 2\eta\mathbf{a}^\top(\nabla_\mathbf{a} l(\mathbf{a}, \bar{\mathbf{b}}) - \mathbb{E}[\nabla_\mathbf{a} l(\mathbf{a}, \bar{\mathbf{b}})]) - 2\eta\mathbf{a}^\top\mathbb{E}[\nabla_\mathbf{a} l(\mathbf{a}, \bar{\mathbf{b}})] \tag{149}$$

where (147) follows by $\eta^2\|\nabla_\mathbf{a} l(\mathbf{a}, \bar{\mathbf{b}})\|^2 \geq 0$, and (148) follows by $\mathbf{a}^\top\mathbf{a} = 1$. The first subtrahend $2\eta\mathbf{a}^\top(\nabla_\mathbf{a} l(\mathbf{a}, \bar{\mathbf{b}}) - \mathbb{E}[\nabla_\mathbf{a} l(\mathbf{a}, \bar{\mathbf{b}})])$ is upper bounded such that

$$2\eta\mathbf{a}^\top(\nabla_\mathbf{a} l(\mathbf{a}, \bar{\mathbf{b}}) - \mathbb{E}[\nabla_\mathbf{a} l(\mathbf{a}, \bar{\mathbf{b}})]) \leq 2\eta\|\mathbf{a}\| \cdot \|(\nabla_\mathbf{a} l(\mathbf{a}, \bar{\mathbf{b}}) - \mathbb{E}[\nabla_\mathbf{a} l(\mathbf{a}, \bar{\mathbf{b}})])\| \tag{150}$$

$$= 2\eta\|(\nabla_\mathbf{a} l(\mathbf{a}, \bar{\mathbf{b}}) - \mathbb{E}[\nabla_\mathbf{a} l(\mathbf{a}, \bar{\mathbf{b}})])\| \tag{151}$$

$$\leq 4\eta\tilde{\epsilon}(\epsilon' + 1)^2\|\mathbf{b}^*\|^2 \tag{152}$$

(152) uses Lemma A.9. The second subtrahend is upper bounded such that

$$2\eta \mathbf{a}^\top \mathbb{E}[\nabla_\mathbf{a} l(\mathbf{a}, \bar{\mathbf{b}})] = 4\eta \mathbf{a}^\top (\mathbf{a}\bar{\mathbf{b}}^\top - \mathbf{a}^* {\mathbf{b}^*}^\top)\bar{\mathbf{b}} \tag{153}$$

$$= 4\eta \mathbf{a}^\top (\mathbf{a}\bar{\mathbf{b}}^\top - \mathbf{a}^* {\mathbf{b}^*}^\top)\mathbf{g} - 4\eta \mathbf{a}^\top (\mathbf{a}\bar{\mathbf{b}}^\top - \mathbf{a}^* {\mathbf{b}^*}^\top)(\mathbf{g} - \bar{\mathbf{b}}) \tag{154}$$

where $\mathbf{a}^\top (\mathbf{a}\bar{\mathbf{b}}^\top - \mathbf{a}^* {\mathbf{b}^*}^\top)\mathbf{g} = -\mathbf{a}^\top((\mathbf{I}_d - \mathbf{a}\mathbf{a}^\top)\mathbf{a}^* {\mathbf{b}^*}^\top + \mathbf{a}(\mathbf{g} - \bar{\mathbf{b}})^\top)\mathbf{g} = (\bar{\mathbf{b}} - \mathbf{g})^\top \mathbf{g}$. The second term is simplified via $\mathbf{a}^\top (\mathbf{a}\bar{\mathbf{b}}^\top - \mathbf{a}^* {\mathbf{b}^*}^\top)(\mathbf{g} - \bar{\mathbf{b}}) = \mathbf{a}^\top((\mathbf{I}_d - \mathbf{a}\mathbf{a}^\top)\mathbf{a}^* {\mathbf{b}^*}^\top + \mathbf{a}(\mathbf{g} - \bar{\mathbf{b}})^\top)(\bar{\mathbf{b}} - \mathbf{g}) = -(\mathbf{g} - \bar{\mathbf{b}})^2$. Both simplifications use $\mathbf{a}^\top(\mathbf{I}_d - \mathbf{a}\mathbf{a}^\top) = 0$ and $\mathbf{a}^\top \mathbf{a} = 1$. (154) becomes

$$2\eta \mathbf{a}^\top \mathbb{E}[\nabla_\mathbf{a} l(\mathbf{a}, \bar{\mathbf{b}})] = 4\eta(\bar{\mathbf{b}} - \mathbf{g})^\top \mathbf{g} + 4\eta(\mathbf{g} - \bar{\mathbf{b}})^2 \tag{155}$$

$$\leq 4\eta \|\mathbf{g} - \bar{\mathbf{b}}\|\|\mathbf{b}^*\| + 4\eta\|\mathbf{g} - \bar{\mathbf{b}}\|^2 \tag{156}$$

$$\leq 4\eta \epsilon' \delta^0 \|\mathbf{b}^*\|^2 + 4\eta (\epsilon')^2 (\delta^t)^2 \|\mathbf{b}^*\|^2 \tag{157}$$

$$\leq 4\eta ((\epsilon')^2 + \epsilon') \|\mathbf{b}^*\|^2 \tag{158}$$

with probability at least $1 - q^{-10}$. (157) uses Lemma A.9. Combining (152) and (158), we obtain

$$\|\hat{\mathbf{a}}^+\|^2 \geq 1 - 4\eta\tilde{\epsilon}(\epsilon' + 1)^2 \|\mathbf{b}^*\|^2 - 4\eta((\epsilon')^2 + \epsilon')\|\mathbf{b}^*\|^2 \tag{159}$$

with probability at least $1 - 2q^{-10}$. Combining (144), (134) and (159), we obtain

$$\delta^1 \leq \frac{E_1 + E_2}{\|\hat{\mathbf{a}}^+\|} \leq \frac{\delta^0(1 - 2\eta((1 - (\delta^0)^2) - \epsilon')\|\mathbf{b}^*\|^2 + 2\eta\tilde{\epsilon}(\epsilon' + 1)^2\|\mathbf{b}^*\|^2)}{\sqrt{1 - 4\eta\tilde{\epsilon}(\epsilon' + 1)^2\|\mathbf{b}^*\|^2 - 4\eta((\epsilon')^2 + \epsilon')\|\mathbf{b}^*\|^2}} = \delta^0 C \tag{160}$$

Still we can choose $\epsilon', \tilde{\epsilon} < \frac{1 - (\delta^0)^2}{16}$ such that $(1 - (\delta^0)^2) > \max(4(\tilde{\epsilon}(\epsilon' + 1)^2 + (\epsilon')^2 + \epsilon'), 2\epsilon' + 2\tilde{\epsilon}(\epsilon' + 1)^2)$ holds. Then we obtain

$$C = \frac{1 - 2\eta((1 - (\delta^0)^2) - \epsilon')\|\mathbf{b}^*\|^2 + 2\eta\tilde{\epsilon}(\epsilon' + 1)^2\|\mathbf{b}^*\|^2}{\sqrt{1 - 4\eta\tilde{\epsilon}(\epsilon' + 1)^2\|\mathbf{b}^*\|^2 - 4\eta((\epsilon')^2 + \epsilon')\|\mathbf{b}^*\|^2}} \tag{161}$$

$$= \frac{1 - 2\eta(1 - (\delta^0)^2)\|\mathbf{b}^*\|^2 + 2\eta\epsilon'\|\mathbf{b}^*\|^2 + 2\eta\tilde{\epsilon}(\epsilon' + 1)^2\|\mathbf{b}^*\|^2}{\sqrt{1 - 4\eta(\tilde{\epsilon}(\epsilon' + 1)^2 + (\epsilon')^2 + \epsilon')\|\mathbf{b}^*\|^2}} \tag{162}$$

$$\leq \frac{1 - 2\eta(1 - (\delta^0)^2)\|\mathbf{b}^*\|^2 + \eta(2\epsilon' + 2\tilde{\epsilon}(\epsilon' + 1)^2)\|\mathbf{b}^*\|^2}{\sqrt{1 - 4\eta(\tilde{\epsilon}(\epsilon' + 1)^2 + (\epsilon')^2 + \epsilon')\|\mathbf{b}^*\|^2}} \tag{163}$$

$$\leq \frac{1 - \eta(1 - (\delta^0)^2)\|\mathbf{b}^*\|^2}{\sqrt{1 - \eta(1 - (\delta^0)^2)\|\mathbf{b}^*\|^2}} \tag{164}$$

$$= \sqrt{1 - \eta(1 - (\delta^0)^2)\|\mathbf{b}^*\|^2} \tag{165}$$

Assuming $\eta \leq \frac{1}{L_{max}^2} \leq \frac{1}{\|\mathbf{b}^*\|^2}$, $1 - \eta(1 - (\delta^0)^2)\|\mathbf{b}^*\|^2$ is strictly positive. Therefore we obtain, with probability at least $1 - 2q^{-10}$,

$$\delta^1 \leq \delta^0 \sqrt{1 - \eta(1 - (\delta^0)^2)\|\mathbf{b}^*\|^2}. \tag{166}$$

**Inductive Hypothesis.** Based on inductive hypothesis, by proving $\delta^1 \leq \delta^0 C$, the assumption that $\delta^t \leq \delta^{t-1} C \leq \cdots \leq \delta^1 C^{t-1}$, and proving $\delta^{t+1} \leq \delta^t C$, we conclude that $\delta^t \leq \delta^{t-1} C$ holds for all $t \in [T]$ iterations. We take a union bound over all $t \in [T]$ such that,

$$\mathbb{P}\left\{ \bigcap_{t=0}^{T-1} \delta^{t+1} \leq \delta^t \sqrt{1 - c(1 - (\delta^0)^2)} \right\} \geq 1 - 2Tq^{-10}. \tag{167}$$

**Solve for $T$.** In order to achieve $\epsilon$-recovery of $\mathbf{a}^*$, we need

$$\delta^0 (1 - c(1 - (\delta^0)^2))^{\frac{T}{2}} \leq \epsilon \tag{168}$$

$$(1 - c(1 - (\delta^0)^2))^{\frac{T}{2}} \leq \frac{\epsilon}{\delta^0} \tag{169}$$

$$\frac{T}{2} \log(1 - c(1 - (\delta^0)^2)) \leq \log(\frac{\epsilon}{\delta^0}) \tag{170}$$

$$\tag{171}$$

We proceed such that

$$T \geq \frac{2 \log(\frac{\epsilon}{\delta^0})}{\log(1 - c(1 - (\delta^0)^2))} \tag{172}$$

$$> \frac{2 \log(\frac{\epsilon}{\delta^0})}{-c(1 - (\delta^0)^2)} \tag{173}$$

$$= \frac{2}{c(1 - (\delta^0)^2)} \log(\frac{\delta^0}{\epsilon}) \tag{174}$$

where (173) follows by using $\log(1-x) < -x$ for $|x| < 1$. Thus, with probability at least $1 - 2Tq^{-10}$, $\delta^T = \sin\theta(\mathbf{a}^T, \mathbf{a}^*) \leq \epsilon$.

**Convergence to the target model.** We now aim to upper bound $\|\mathbf{a}^T(\mathbf{b}^{T+1})^\top - \mathbf{a}^*(\mathbf{b}^*)^\top\|$. Recall that $(\mathbf{g}^T)^\top = (\mathbf{a}^T)^\top \mathbf{a}^* \mathbf{b}^{*\top}$ and $\delta^T = \|(\mathbf{I}_d - \mathbf{a}^T(\mathbf{a}^T)^\top)\mathbf{a}^*\|$, we have

$$\|\mathbf{a}^T(\mathbf{b}^{T+1})^\top - \mathbf{a}^*\mathbf{b}^{*\top}\| = \|\mathbf{a}^T(\mathbf{b}^{T+1})^\top - \mathbf{a}^T(\mathbf{g}^T)^\top + \mathbf{a}^T(\mathbf{g}^T)^\top - \mathbf{a}^*\mathbf{b}^{*\top}\| \tag{175}$$

$$\leq \|\mathbf{a}^T(\mathbf{b}^{T+1})^\top - \mathbf{a}^T(\mathbf{g}^T)^\top\| + \|\mathbf{a}^T(\mathbf{g}^T)^\top - \mathbf{a}^*\mathbf{b}^{*\top}\| \tag{176}$$

$$= \|\mathbf{a}^T(\mathbf{b}^{T+1} - \mathbf{g}^T)^\top\| + \|(\mathbf{a}^T(\mathbf{a}^T)^\top - \mathbf{I}_d)\mathbf{a}^*\mathbf{b}^{*\top}\| \tag{177}$$

$$= \|\mathbf{a}^T\|\|\mathbf{b}^{T+1} - \mathbf{g}^T\| + \|(\mathbf{I}_d - \mathbf{a}^T(\mathbf{a}^T)^\top)\mathbf{a}^*\|\|\mathbf{b}^*\| \tag{178}$$

$$\leq \epsilon'\delta^T\|\mathbf{b}^*\| + \delta^T\|\mathbf{b}^*\| \tag{179}$$

$$= (1 + \epsilon')\epsilon\|\mathbf{b}^*\| \tag{180}$$

$$= (1 + \epsilon')\epsilon\|\mathbf{a}^*\mathbf{b}^{*\top}\| \tag{181}$$

where (179) is by Lemma A.9 and the fact that $\|\mathbf{a}^T\| = 1$, and (181) is due to the fact that $\|\mathbf{x}\mathbf{y}^\top\| = \|\mathbf{x}\|\|\mathbf{y}\|$ and $\|\mathbf{a}^*\| = 1$. □

### A.4.1. PROOF OF PROPOSITION 5.5

*Proof.* We start by fixing $\mathbf{a}^0$ and updating $\mathbf{b}_i$ to minimize the objective. Let $\mathbf{a} = \mathbf{a}^0$. We obtain

$$\mathbf{b}_i^\top = \frac{\mathbf{a}^\top \mathbf{X}_i^\top \mathbf{X}_i \mathbf{a}^* \mathbf{b}^{*\top}}{\mathbf{a}^\top \mathbf{X}_i^\top \mathbf{X}_i \mathbf{a}} \tag{182}$$

$$(\mathbf{b}^{FFA})^\top = \frac{1}{N} \sum_{i=1}^{N} \frac{\mathbf{a}^\top \mathbf{X}_i^\top \mathbf{X}_i \mathbf{a}^* \mathbf{b}^{*\top}}{\mathbf{a}^\top \mathbf{X}_i^\top \mathbf{X}_i \mathbf{a}} \tag{183}$$

let $\bar{\mathbf{b}} = \mathbf{b}^{FFA}$. We aim to compute the expected value of $\frac{1}{N} \sum_{i=1}^{N} \frac{1}{m} \|\mathbf{X}_i \mathbf{a}^* \mathbf{b}^{*\top} - \mathbf{X}_i \mathbf{a} \bar{\mathbf{b}}^\top\|^2$ where the expectation is over all the randomness in the $\mathbf{X}_i$. We define

$$s_i = \frac{\mathbf{a}^\top \mathbf{X}_i^\top \mathbf{X}_i \mathbf{a}^*}{\mathbf{a}^\top \mathbf{X}_i^\top \mathbf{X}_i \mathbf{a}} = \frac{(\mathbf{X}_i \mathbf{a})^\top (\mathbf{X}_i \mathbf{a}^*)}{\|\mathbf{X}_i \mathbf{a}\|^2} \tag{184}$$

so that $\bar{\mathbf{b}} = \frac{1}{N} \sum_{i=1}^{N} s_i \mathbf{b}^* = \bar{s}\mathbf{b}^*$. For each $i$, the norm becomes

$$\|\mathbf{X}_i \mathbf{a}^* \mathbf{b}^{*\top} - \mathbf{X}_i \mathbf{a} \bar{\mathbf{b}}^\top\|^2 = \|(\mathbf{X}_i \mathbf{a}^* - \bar{s}\mathbf{X}_i \mathbf{a})\mathbf{b}^{*\top}\|^2 \tag{185}$$

$$= \|\mathbf{X}_i \mathbf{a}^* - \bar{s}\mathbf{X}_i \mathbf{a}\|^2 \|\mathbf{b}^*\|^2 \tag{186}$$

using the fact that $\|\mathbf{u}\mathbf{v}^\top\|^2 = \|\mathbf{u}\|^2\|\mathbf{v}\|^2$ for vectors $\mathbf{u}$ and $\mathbf{v}$. Therefore, $\mathbb{E}[\frac{1}{N} \sum_{i=1}^{N} \frac{1}{m} \|\mathbf{X}_i \mathbf{a}^* \mathbf{b}^{*\top} - \mathbf{X}_i \mathbf{a} \bar{\mathbf{b}}^\top\|^2]$ is reduced to $\mathbb{E}[\frac{1}{N} \sum_{i=1}^{N} \frac{1}{m} \|\mathbf{X}_i \mathbf{a}^* - \mathbf{X}_i \mathbf{a}\|^2] \cdot \|\mathbf{b}^*\|^2$.

Since each entry of $\mathbf{X}_i$ is independently and identically distributed according to a standard Gaussian distribution, both $\mathbf{a}^*$ and $\mathbf{a}$ are unit vectors, the vectors $\mathbf{X}_i \mathbf{a}^*$ and $\mathbf{X}_i \mathbf{a}$ are $\mathcal{N}(0, \mathbf{I}_m)$. The cross-covariance is $\alpha \mathbf{I}_m$ where $\alpha = \mathbf{a}^\top \mathbf{a}^*$.

By linearity, we can show that $\frac{1}{N}\sum_{i=1}^{N}\frac{1}{m}\|\mathbf{X}_i\mathbf{a}^* - \bar{s}\mathbf{X}_i\mathbf{a}\|^2$ has the same expectation as $\frac{1}{m}\|\mathbf{X}_1\mathbf{a}^* - \bar{s}\mathbf{X}_1\mathbf{a}\|^2$ because all $(\mathbf{X}_i\mathbf{a}^*, \mathbf{X}_i\mathbf{a})$ are i.i.d. pairs. Let $z_1 = \frac{s_1}{N}$ and $z_2 = \frac{s_2+\cdots+s_N}{N}$, we have $\|\mathbf{X}_1\mathbf{a}^* - z_1\mathbf{X}_1\mathbf{a} - z_2\mathbf{X}_1\mathbf{a}\|^2$. Let $\mathbf{v} = \mathbf{X}_1\mathbf{a}^*$, $\mathbf{u}_1 = z_1\mathbf{X}_1\mathbf{a}$ and $\mathbf{u}_2 = z_2\mathbf{X}_1\mathbf{a}$. Thus,

$$\|\mathbf{X}_1\mathbf{a}^* - z_1\mathbf{X}_1\mathbf{a} - z_2\mathbf{X}_1\mathbf{a}\|^2 = \|\mathbf{v} - \mathbf{u}_1 - \mathbf{u}_2\|^2 \tag{187}$$

$$= \mathbf{v}^\top\mathbf{v} + \mathbf{u}_1^\top\mathbf{u}_1 + \mathbf{u}_2^\top\mathbf{u}_2 - 2\mathbf{v}^\top\mathbf{u}_1 - 2\mathbf{v}^\top\mathbf{u}_2 + 2\mathbf{u}_1^\top\mathbf{u}_2 \tag{188}$$

Now we compute the expectation term by term.

**Expected value of $\mathbf{v}^\top\mathbf{v}$**   We have $\mathbb{E}[\mathbf{v}^\top\mathbf{v}] = \mathbb{E}[\|\mathbf{X}_1\mathbf{a}^*\|^2] = m$.

**Expected value of $\mathbf{u}_1^\top\mathbf{u}_1$**   We have

$$\mathbf{u}_1^\top\mathbf{u}_1 = z_1^2\|\mathbf{X}_1\mathbf{a}\|^2 \tag{189}$$

$$= \frac{s_1^2}{N^2}\|\mathbf{X}_1\mathbf{a}\|^2 \tag{190}$$

$$= \frac{1}{N^2}\frac{((\mathbf{X}_1\mathbf{a})^\top(\mathbf{X}_1\mathbf{a}^*))^2}{\|\mathbf{X}_1\mathbf{a}\|^4}\|\mathbf{X}_1\mathbf{a}\|^2 \tag{191}$$

$$= \frac{1}{N^2}\frac{((\mathbf{X}_1\mathbf{a})^\top(\mathbf{X}_1\mathbf{a}^*))^2}{\|\mathbf{X}_1\mathbf{a}\|^2} \tag{192}$$

Note that $(\mathbf{X}_1\mathbf{a}^*, \mathbf{X}_1\mathbf{a})$ is a correlated Gaussian pair with correlation $\alpha = \mathbf{a}^\top\mathbf{a}^*$. Without loss of generality, we assume $\mathbf{a} = \mathbf{e}_1$ thus $\mathbf{a}^* = \alpha\mathbf{e}_1 + \sqrt{1-\alpha^2}\mathbf{e}_2$, where $\mathbf{e}_1$ and $\mathbf{e}_2$ are standard basis vectors in $\mathbb{R}^d$. So we can get $\mathbf{X}_1\mathbf{a} = \mathbf{X}_1\mathbf{e}_1 = \mathbf{x}_{1,1}$, where $\mathbf{x}_{1,1}$ denotes the first column of $\mathbf{X}_1$. Accordingly $\mathbf{X}_1\mathbf{a}^* = \alpha\mathbf{X}_1\mathbf{e}_1 + \sqrt{1-\alpha^2}\mathbf{X}_1\mathbf{e}_2 = \alpha\mathbf{x}_{1,1} + \sqrt{1-\alpha^2}\mathbf{x}_{1,2}$ where $\mathbf{x}_{1,2}$ denotes the second column of $\mathbf{X}_1$. Therefore (192) can be written as $\frac{1}{N^2}\frac{(\mathbf{x}_{1,1}^\top(\alpha\mathbf{x}_{1,1}+\beta\mathbf{x}_{1,2}))^2}{\|\mathbf{x}_{1,1}\|^2}$. Now we take expectation of it.

$$\mathbb{E}\left[\frac{1}{N^2}\frac{((\mathbf{X}_1\mathbf{a})^\top(\mathbf{X}_1\mathbf{a}^*))^2}{\|\mathbf{X}_1\mathbf{a}\|^2}\right] = \mathbb{E}\left[\frac{1}{N^2}\frac{(\mathbf{x}_{1,1}^\top(\alpha\mathbf{x}_{1,1}+\beta\mathbf{x}_{1,2}))^2}{\|\mathbf{x}_{1,1}\|^2}\right] = \frac{1}{N^2}\mathbb{E}\left[\frac{(\mathbf{x}_{1,1}^\top(\alpha\mathbf{x}_{1,1}+\beta\mathbf{x}_{1,2}))^2}{\|\mathbf{x}_{1,1}\|^2}\right] \tag{193}$$

Let $r_1 = \|\mathbf{x}_{1,1}\|^2$ and $r_2 = \mathbf{x}_{1,1}^\top\mathbf{x}_{1,2}$. We have

$$\mathbb{E}\left[\frac{(\mathbf{x}_{1,1}^\top(\alpha\mathbf{x}_{1,1}+\beta\mathbf{x}_{1,2}))^2}{\|\mathbf{x}_{1,1}\|^2}\right] = \mathbb{E}\left[\frac{(\alpha r_1 + \beta r_2)^2}{r_1}\right] \tag{194}$$

$$= \mathbb{E}\left[\frac{\alpha^2 r_1^2 + \beta^2 r_2^2 + 2\alpha\beta r_1 r_2}{r_1}\right] \tag{195}$$

$$= \mathbb{E}\left[\alpha^2 r_1\right] + \mathbb{E}\left[\frac{\beta^2 r_2^2}{r_1}\right] + \mathbb{E}\left[2\alpha\beta r_2\right] \tag{196}$$

where $\mathbb{E}\left[\alpha^2 r_1\right] = \alpha^2\mathbb{E}\left[\|\mathbf{x}_{1,1}\|^2\right] = \alpha^2 m$, and $\mathbb{E}\left[2\alpha\beta r_2\right] = 2\alpha\beta\mathbb{E}\left[r_2\right] = 2\alpha\beta\mathbb{E}\left[\mathbf{x}_{1,1}^\top\mathbf{x}_{1,2}\right] = 0$ because $\mathbf{x}_{1,1}$ and $\mathbf{x}_{1,2}$ are independent standard Gaussian vectors. Then we analyze $\mathbb{E}\left[\frac{\beta^2 r_2^2}{r_1}\right] = \beta^2\mathbb{E}\left[\frac{r_2^2}{r_1}\right]$. Condition on $\mathbf{x}_{1,1}$,

$$\mathbb{E}\left[r_2|\mathbf{x}_{1,1}\right] = \mathbb{E}\left[\mathbf{x}_{1,1}^\top\mathbf{x}_{1,2}|\mathbf{x}_{1,1}\right] = \mathbf{x}_{1,1}^\top\mathbb{E}\left[\mathbf{x}_{1,2}\right] = 0 \tag{197}$$

and $\mathrm{Var}(r_2|\mathbf{x}_{1,1}) = \|\mathbf{x}_{1,1}\|^2 = r_1$, thus

$$r_2|\mathbf{x}_{1,1} = \mathbf{x}_{1,1}^\top\mathbf{x}_{1,2}|\mathbf{x}_{1,1} \sim \mathcal{N}(0, r_1) \tag{198}$$

Then we obtain

$$\mathbb{E}\left[r_2^2|\mathbf{x}_{1,1}\right] = r_1 \tag{199}$$

Therefore $\mathbb{E}\left[\frac{r_2^2}{r_1}|\mathbf{x}_{1,1}\right] = \frac{\mathbb{E}\left[r_2^2|\mathbf{x}_{1,1}\right]}{r_1} = 1$. We take total expectation $\mathbb{E}\left[\frac{r_2^2}{r_1}\right] = \mathbb{E}\left[\mathbb{E}\left[\frac{r_2^2}{r_1}|\mathbf{x}_{1,1}\right]\right] = 1$. Summarizing,

$$\mathbb{E}\left[\frac{((\mathbf{X}_1\mathbf{a})^\top(\mathbf{X}_1\mathbf{a}^*))^2}{\|\mathbf{X}_1\mathbf{a}\|^2}\right] = \mathbb{E}\left[\frac{(\alpha r_1 + \beta r_2)^2}{r_1}\right] \tag{200}$$

$$= \mathbb{E}\left[\alpha^2 r_1\right] + \mathbb{E}\left[\frac{\beta^2 r_2^2}{r_1}\right] + \mathbb{E}\left[2\alpha\beta r_2\right] \tag{201}$$

$$= \alpha^2 m + \beta^2 \tag{202}$$

$$\mathbb{E}\left[\mathbf{u}_1^\top\mathbf{u}_1\right] = \frac{1}{N^2}\mathbb{E}\left[\frac{((\mathbf{X}_1\mathbf{a})^\top(\mathbf{X}_1\mathbf{a}^*))^2}{\|\mathbf{X}_1\mathbf{a}\|^2}\right] \tag{203}$$

$$= \frac{\alpha^2 m + (1-\alpha^2)}{N^2} \tag{204}$$

**Expected value of $\mathbf{u}_2^\top\mathbf{u}_2$** We have $\mathbf{u}_2^\top\mathbf{u}_2 = z_2^2\|\mathbf{X}_1\mathbf{a}\|^2$ where $z_2 = \frac{s_2 + \cdots + s_N}{N}$ is independent of pair $(\mathbf{X}_1\mathbf{a}^*, \mathbf{X}_1\mathbf{a})$. To compute $\mathbb{E}\left[z_2^2\|\mathbf{X}_1\mathbf{a}\|^2\right]$, first we condition on $z_2$ to obtain,

$$\mathbb{E}\left[z_2^2\|\mathbf{X}_1\mathbf{a}\|^2|z_2\right] = z_2^2\mathbb{E}\left[\|\mathbf{X}_1\mathbf{a}\|^2\right] = z_2^2 m \tag{205}$$

Then we take total expectation $\mathbb{E}\left[z_2^2\|\mathbf{X}_1\mathbf{a}\|^2\right] = \mathbb{E}\left[\mathbb{E}\left[z_2^2\|\mathbf{X}_1\mathbf{a}\|^2|z_2\right]\right] = \mathbb{E}\left[z_2^2 m\right] = m\mathbb{E}\left[z_2^2\right]$.

$$\mathbb{E}\left[z_2^2\right] = \mathbb{E}\left[\frac{(s_2 + \cdots + s_N)^2}{N^2}\right] \tag{206}$$

$$= \frac{1}{N^2}\mathbb{E}\left[\sum_{i=2}^{N} s_i^2 + \sum_{\substack{i=1,j=1 \\ i \neq j}}^{N} s_i s_j\right] \tag{207}$$

$$= \frac{1}{N^2}\left(\sum_{i=2}^{N}\mathbb{E}\left[s_i^2\right] + \sum_{\substack{i=1,j=1 \\ i \neq j}}^{N}\mathbb{E}\left[s_i s_j\right]\right) \tag{208}$$

Write $s_i = \frac{(\mathbf{X}_i\mathbf{a})^\top(\mathbf{X}_i\mathbf{a}^*)}{\|\mathbf{X}_i\mathbf{a}\|^2}$. Without loss of generality, we assume $\mathbf{a} = \mathbf{e}_1$ thus $\mathbf{a}^* = \alpha\mathbf{e}_1 + \sqrt{1-\alpha^2}\mathbf{e}_2$, where $\mathbf{e}_1$ and $\mathbf{e}_2$ are standard basis vectors in $\mathbb{R}^d$. Thus, we have $\mathbf{X}_i\mathbf{a} = \mathbf{X}_i\mathbf{e}_1 = \mathbf{x}_{i,1}$, where $\mathbf{x}_{i,1}$ represents the first column of $\mathbf{X}_i$. Similarly, $\mathbf{X}_i\mathbf{a}^* = \alpha\mathbf{X}_i\mathbf{e}_1 + \sqrt{1-\alpha^2}\mathbf{X}_i\mathbf{e}_2 = \alpha\mathbf{x}_{i,1} + \sqrt{1-\alpha^2}\mathbf{x}_{i,2}$, where $\mathbf{x}_{i,2}$ denotes the second column of $\mathbf{X}_i$.

Hence,

$$(\mathbf{X}_i\mathbf{a})^\top(\mathbf{X}_i\mathbf{a}^*) = \mathbf{x}_{i,1}^\top(\alpha\mathbf{x}_{i,1} + \sqrt{1-\alpha^2}\mathbf{x}_{i,2}) = \alpha\|\mathbf{x}_{i,1}\|^2 + \sqrt{1-\alpha^2}(\mathbf{x}_{i,1}^\top\mathbf{x}_{i,2}) \tag{209}$$

With $\|\mathbf{X}_i\mathbf{a}\|^2 = \|\mathbf{x}_{i,1}\|^2$, we have

$$s_i = \frac{\alpha\|\mathbf{x}_{i,1}\|^2 + \sqrt{1-\alpha^2}(\mathbf{x}_{i,1}^\top\mathbf{x}_{i,2})}{\|\mathbf{x}_{i,1}\|^2} = \alpha + \sqrt{1-\alpha^2}\frac{\mathbf{x}_{i,1}^\top\mathbf{x}_{i,2}}{\|\mathbf{x}_{i,1}\|^2} \tag{210}$$

Let $R = \frac{\mathbf{x}_{i,1}^\top\mathbf{x}_{i,2}}{\|\mathbf{x}_{i,1}\|^2}$. Then

$$\mathbb{E}\left[s_i^2\right] = \mathbb{E}\left[\left(\alpha + \sqrt{1-\alpha^2}R\right)^2\right] \tag{211}$$

$$= \mathbb{E}\left[\alpha^2 + (1-\alpha^2)R^2 + 2\alpha\sqrt{1-\alpha^2}R\right] \tag{212}$$

$$= \alpha^2 + (1-\alpha^2)\mathbb{E}\left[R^2\right] + 2\alpha\sqrt{1-\alpha^2}\mathbb{E}\left[R\right] \tag{213}$$

For $\mathbb{E}\left[R\right] = \mathbb{E}\left[\frac{\mathbf{x}_{i,1}^{\top}\mathbf{x}_{i,2}}{\|\mathbf{x}_{i,1}\|^2}\right]$, similarly as in (198), $\mathbf{x}_{i,1}^{\top}\mathbf{x}_{i,2}|\mathbf{x}_{i,1} \sim \mathcal{N}(0, \|\mathbf{x}_{i,1}\|^2)$, thus $\frac{\mathbf{x}_{i,1}^{\top}\mathbf{x}_{i,2}}{\|\mathbf{x}_{i,1}\|^2}|\mathbf{x}_{i,1} \sim \mathcal{N}(0, \frac{1}{\|\mathbf{x}_{i,1}\|^2})$, then

$$\mathbb{E}\left[R\right] = \mathbb{E}\left[\mathbb{E}\left[R|\mathbf{x}_{i,1}\right]\right] = 0 \tag{214}$$

For $\mathbb{E}\left[R^2\right]$, since $\mathbf{x}_{i,1}^{\top}\mathbf{x}_{i,2}|\mathbf{x}_{i,1} \sim \mathcal{N}(0, \|\mathbf{x}_{i,1}\|^2)$, so $\mathbb{E}\left[(\mathbf{x}_{i,1}^{\top}\mathbf{x}_{i,2})^2|\mathbf{x}_{i,1}\right] = \|\mathbf{x}_{i,1}\|^2$. Thus, with $R^2 = \frac{(\mathbf{x}_{i,1}^{\top}\mathbf{x}_{i,2})^2}{\|\mathbf{x}_{i,1}\|^4}$,

$$\mathbb{E}\left[R^2|\mathbf{x}_{i,1}\right] = \frac{\mathbb{E}\left[(\mathbf{x}_{i,1}^{\top}\mathbf{x}_{i,2})^2|\mathbf{x}_{i,1}\right]}{\|\mathbf{x}_{i,1}\|^4} = \frac{1}{\|\mathbf{x}_{i,1}\|^2} \tag{215}$$

$$\mathbb{E}\left[R^2\right] = \mathbb{E}\left[\frac{1}{\|\mathbf{x}_{i,1}\|^2}\right] \tag{216}$$

For a $m$-dimensional standard Gaussian vector, $\|\mathbf{x}_{i,1}\|^2$ follows a chi-squared distribution with $m$ degrees of freedom. Therefore, $\mathbb{E}\left[R^2\right] = \frac{1}{m-2}$. (213) becomes

$$\mathbb{E}\left[s_i^2\right] = \alpha^2 + (1-\alpha^2)\mathbb{E}\left[R^2\right] + 2\alpha\sqrt{1-\alpha^2}\mathbb{E}\left[R\right] \tag{217}$$

$$= \alpha^2 + (1-\alpha^2)\frac{1}{m-2} \tag{218}$$

Now we compute $\mathbb{E}\left[s_i s_j\right]$ for $i \neq j$. By independence of $s_i$ and $s_j$, $\mathbb{E}\left[s_i s_j\right] = \mathbb{E}\left[s_i\right] \cdot \mathbb{E}\left[s_j\right] = \mathbb{E}\left[s_i\right]^2$. Take expectation of (210),

$$\mathbb{E}\left[s_i\right] = \mathbb{E}\left[\alpha + \sqrt{1-\alpha^2}\frac{\mathbf{x}_{i,1}^{\top}\mathbf{x}_{i,2}}{\|\mathbf{x}_{i,1}\|^2}\right] \tag{219}$$

$$= \alpha + \sqrt{1-\alpha^2}\mathbb{E}\left[\frac{\mathbf{x}_{i,1}^{\top}\mathbf{x}_{i,2}}{\|\mathbf{x}_{i,1}\|^2}\right] \tag{220}$$

$$= \alpha + \sqrt{1-\alpha^2}\mathbb{E}\left[R\right] \tag{221}$$

$$= \alpha \tag{222}$$

Hence, $\mathbb{E}\left[s_i s_j\right] = \alpha^2$. Summarizing,

$$\mathbb{E}\left[\mathbf{u}_2^{\top}\mathbf{u}_2\right] = m\mathbb{E}\left[z_2^2\right] \tag{223}$$

$$= \frac{m}{N^2}\left((N-1)\mathbb{E}\left[s_i^2\right] + (N-1)(N-2)\mathbb{E}\left[s_i s_j\right]\right) \tag{224}$$

$$= \frac{m}{N^2}\left((N-1)(\alpha^2 + (1-\alpha^2)\frac{1}{m-2}) + (N-1)(N-2)\alpha^2\right) \tag{225}$$

$$= \frac{m}{N^2}\left[(N-1)^2\alpha^2 + (N-1)\frac{1-\alpha^2}{m-2}\right] \tag{226}$$

**Expected value of $\mathbf{v}^{\top}\mathbf{u}_1$**   We have $\mathbf{v}^{\top}\mathbf{u}_1 = z_1(\mathbf{X}_1\mathbf{a}^*)^{\top}(\mathbf{X}_1\mathbf{a}) = \frac{s_1}{N}(\mathbf{X}_1\mathbf{a}^*)^{\top}(\mathbf{X}_1\mathbf{a})$. We factor out $\frac{1}{N}$, $\mathbb{E}\left[\mathbf{v}^{\top}\mathbf{u}_1\right] = \frac{1}{N}\mathbb{E}\left[s_1(\mathbf{X}_1\mathbf{a}^*)^{\top}(\mathbf{X}_1\mathbf{a})\right]$. By (202),

$$\mathbb{E}\left[s_1(\mathbf{X}_1\mathbf{a}^*)^{\top}(\mathbf{X}_1\mathbf{a})\right] = \mathbb{E}\left[\frac{((\mathbf{X}_1\mathbf{a})^{\top}(\mathbf{X}_1\mathbf{a}^*))^2}{\|\mathbf{X}_1\mathbf{a}\|^2}\right] \tag{227}$$

$$= \alpha^2 m + (1-\alpha^2) \tag{228}$$

Then

$$\mathbb{E}\left[\mathbf{v}^{\top}\mathbf{u}_1\right] = \frac{\alpha^2 m + (1-\alpha^2)}{N} \tag{229}$$

**Expected value of $\mathbf{v}^\top \mathbf{u}_2$**  We have $\mathbf{v}^\top \mathbf{u}_2 = z_2(\mathbf{X}_1\mathbf{a}^*)^\top(\mathbf{X}_1\mathbf{a})$. Condition on $z_2$ which is independent of $(\mathbf{X}_1\mathbf{a}^*)^\top(\mathbf{X}_1\mathbf{a})$, we obtain

$$\mathbb{E}\left[z_2(\mathbf{X}_1\mathbf{a}^*)^\top(\mathbf{X}_1\mathbf{a})|z_2\right] = z_2\mathbb{E}\left[(\mathbf{X}_1\mathbf{a}^*)^\top(\mathbf{X}_1\mathbf{a})\right] \tag{230}$$

Still we assume $\mathbf{a} = \mathbf{e}_1$ thus $\mathbf{a}^* = \alpha\mathbf{e}_1 + \sqrt{1-\alpha^2}\mathbf{e}_2$, where $\mathbf{e}_1$ and $\mathbf{e}_2$ are standard basis vectors in $\mathbb{R}^d$. With $\mathbf{X}_1\mathbf{a} = \mathbf{X}_1\mathbf{e}_1 = \mathbf{x}_{1,1}$, where $\mathbf{x}_{1,1}$ denotes the first column of $\mathbf{X}_1$, and $\mathbf{X}_1\mathbf{a}^* = \alpha\mathbf{X}_1\mathbf{e}_1 + \sqrt{1-\alpha^2}\mathbf{X}_1\mathbf{e}_2 = \alpha\mathbf{x}_{1,1} + \sqrt{1-\alpha^2}\mathbf{x}_{1,2}$ where $\mathbf{x}_{1,2}$ denotes the second column of $\mathbf{X}_1$, using (209),

$$\mathbb{E}\left[(\mathbf{X}_1\mathbf{a}^*)^\top(\mathbf{X}_1\mathbf{a})\right] = \mathbb{E}\left[\alpha\|\mathbf{x}_{1,1}\|^2 + \sqrt{1-\alpha^2}(\mathbf{x}_{1,1}^\top\mathbf{x}_{1,2})\right] \tag{231}$$

$$= \alpha\mathbb{E}\left[\|\mathbf{x}_{1,1}\|^2\right] + z_2\sqrt{1-\alpha^2}\mathbb{E}\left[\mathbf{x}_{1,1}^\top\mathbf{x}_{1,2}\right] \tag{232}$$

$$= \alpha m \tag{233}$$

Thus $z_2\mathbb{E}\left[(\mathbf{X}_1\mathbf{a}^*)^\top(\mathbf{X}_1\mathbf{a})\right] = z_2\alpha m$, Then we take total expectation

$$\mathbb{E}\left[\mathbb{E}\left[z_2(\mathbf{X}_1\mathbf{a}^*)^\top(\mathbf{X}_1\mathbf{a})|z_2\right]\right] = \mathbb{E}\left[z_2\alpha m\right] \tag{234}$$

$$= \alpha m\mathbb{E}\left[z_2\right] \tag{235}$$

where $z_2 = \frac{s_2 + \cdots + s_N}{N}$. Therefore,

$$\alpha m\mathbb{E}\left[z_2\right] = \frac{\alpha m}{N}\sum_{i=2}^{N}\mathbb{E}\left[s_i\right] = \frac{m}{N}(N-1)\alpha^2 \tag{236}$$

where (236) follows by $\mathbb{E}\left[s_i\right] = \alpha$. Summarizing, we obtain $\mathbb{E}\left[\mathbf{v}^\top \mathbf{u}_2\right] = \frac{m}{N}(N-1)\alpha^2$.

**Expected value of $\mathbf{u}_1^\top \mathbf{u}_2$**  We have $\mathbf{u}_1^\top \mathbf{u}_2 = z_1 z_2\|\mathbf{X}_1\mathbf{a}\|^2$. By definition of $z_1$ and $z_2$, we obtain

$$z_1 z_2\|\mathbf{X}_1\mathbf{a}\|^2 = \frac{1}{N^2}((\mathbf{X}_1\mathbf{a}^*)^\top(\mathbf{X}_1\mathbf{a}))\sum_{i=2}^{N}s_i \tag{237}$$

Since $(\mathbf{X}_1\mathbf{a}^*)^\top(\mathbf{X}_1\mathbf{a})$ depends only on $\mathbf{X}_1$ , $\sum_{i=2}^{N}s_i$ is independent of $\mathbf{X}_1$, we obtain

$$\mathbb{E}\left[\frac{1}{N^2}((\mathbf{X}_1\mathbf{a}^*)^\top(\mathbf{X}_1\mathbf{a}))\sum_{i=2}^{N}s_i\right] = \frac{1}{N^2}\mathbb{E}\left[(\mathbf{X}_1\mathbf{a}^*)^\top(\mathbf{X}_1\mathbf{a})\right]\cdot\mathbb{E}\left[\sum_{i=2}^{N}s_i\right] \tag{238}$$

$$= \frac{1}{N^2}\mathbb{E}\left[(\mathbf{X}_1\mathbf{a}^*)^\top(\mathbf{X}_1\mathbf{a})\right]\cdot(N-1)\mathbb{E}\left[s_i\right] \tag{239}$$

$$= \frac{(N-1)m\alpha^2}{N^2} \tag{240}$$

where (240) follows by $\mathbb{E}\left[(\mathbf{X}_1\mathbf{a}^*)^\top(\mathbf{X}_1\mathbf{a})\right] = \alpha m$ and $\mathbb{E}\left[s_i\right] = \alpha$.

Combining (204), (226),(229),(236),(240) and (188),

$$\frac{1}{m}\|\mathbf{X}_1\mathbf{a}^* - \bar{s}\mathbf{X}_1\mathbf{a}\|^2 = \frac{1}{m}(\mathbf{v}^\top\mathbf{v} + \mathbf{u}_1^\top\mathbf{u}_1 + \mathbf{u}_2^\top\mathbf{u}_2 - 2\mathbf{v}^\top\mathbf{u}_1 - 2\mathbf{v}^\top\mathbf{u}_2 + 2\mathbf{u}_1^\top\mathbf{u}_2) \tag{241}$$

$$= (1-\alpha^2)\left[1 + \frac{N(4-m)-2}{N^2 m(m-2)}\right] \tag{242}$$

$$= (\delta^0)^2(1+\tilde{c}) \tag{243}$$

where $\delta^0$ is the angle distance between $\mathbf{a}$ and $\mathbf{a}^*$. The quantity $\tilde{c} = \frac{N(4-m)-2}{N^2 m(m-2)} = O(\frac{1}{Nm})$ as $N$ and $m$ approach infinity. Therefore,

$$\mathbb{E}\left[\frac{1}{N}\sum_{i=1}^{N}\frac{1}{m}\|\mathbf{X}_i\mathbf{a}^*\mathbf{b}^{*\top} - \mathbf{X}_i\mathbf{a}\bar{\mathbf{b}}^\top\|^2\right] = (1+\tilde{c})(\delta^0)^2\|\mathbf{b}^*\|^2 \tag{244}$$

$\square$

## A.5. Vector-vector case with heterogeneous clients

Consider a federated setting with $N$ clients, each with the following local linear model

$$f_i(\mathbf{X}_i) = \mathbf{X}_i \mathbf{a} \mathbf{b}^\top \tag{245}$$

where $\mathbf{a} \in \mathbb{R}^d$ is a unit vector and $\mathbf{b} \in \mathbb{R}^d$ are the LoRA weights corresponding to rank $r = 1$. In this setting, we model the local data of $i$-th client such that $\mathbf{Y}_i = \mathbf{X}_i \mathbf{a}^* \mathbf{b}_i^{*\top}$ for some ground truth LoRA weights $\mathbf{a}^* \in \mathbb{R}^d$, which is a unit vector, and local $\mathbf{b}_i^* \in \mathbb{R}^d$. We consider the following objective

$$\min_{\mathbf{a} \in \mathbb{R}^d, \mathbf{b} \in \mathbb{R}^d} \frac{1}{N} \sum_{i=1}^{N} l_i(\mathbf{a}, \mathbf{b}) \tag{246}$$

We consider the local population loss $l_i(\mathbf{a}, \mathbf{b}) = \|\mathbf{a}^* \mathbf{b}_i^{*\top} - \mathbf{a} \mathbf{b}^\top\|^2$.

We aim to learn a shared model $(\mathbf{a}, \mathbf{b})$ for all the clients. It is straightforward to observe that $(\mathbf{a}', \mathbf{b}')$ is a global minimizer of if and only if $\mathbf{a}' \mathbf{b}'^\top = \mathbf{a}^* \bar{\mathbf{b}}^*$, where $\bar{\mathbf{b}}^* = \frac{1}{N} \sum_{i=1}^{N} \mathbf{b}_i^*$. The solution is unique and satisfies $\mathbf{a}' = \mathbf{a}^*$ and $\mathbf{b}' = \bar{\mathbf{b}}^*$. With this global minimizer, we obtain the corresponding minimum global error of $\frac{1}{N} \sum_{i=1}^{N} \|\mathbf{a}^* (\mathbf{b}_i^* - \bar{\mathbf{b}}^*)^\top\|^2$.

We aim to show that the training procedure described in Algorithm 2 learns the global minimizer $(\mathbf{a}^*, \bar{\mathbf{b}}^*)$. First, we make typical assumption and definition.

**Assumption A.12.** There exists $L_{max} < \infty$ (known a priori), s.t. $\|\bar{\mathbf{b}}^*\| \leq L_{max}$.

**Definition A.13.** (Client variance) For $\gamma > 0$, we define $\gamma^2 := \frac{1}{N} \sum_{i=1}^{N} \|\mathbf{b}_i^* - \bar{\mathbf{b}}^*\|^2$, where $\bar{\mathbf{b}}^* = \frac{1}{N} \sum_{i=1}^{N} \mathbf{b}_i^*$.

**Theorem A.14.** (Convergence of RoLoRA for linear regressor in heterogeneous setting) Let $\delta^t = \|(\mathbf{I}_d - \mathbf{a}^* \mathbf{a}^{*\top}) \mathbf{a}^t\|$ be the angle distance between $\mathbf{a}^*$ and $\mathbf{a}^t$ of $t$-th iteration. Suppose we are in the setting described in Section A.5 and apply Algorithm 2 for optimization. Given a random initial $\mathbf{a}^0$, an initial angle distance $\delta_0 \in (0, 1)$, we set the step size $\eta \leq \frac{1}{2L_{max}^2}$ and the number of iterations $T \geq \frac{1}{c(1-(\delta^0)^2)} \log(\frac{\delta^0}{\epsilon})$ for $c \in (0, 1)$. Under these conditions, we achieve the following

$$\sin \theta(\mathbf{a}^T, \mathbf{a}^*) \leq \epsilon, \text{ and } \|\mathbf{a}^T (\mathbf{b}^{T+1})^\top - \mathbf{a}^* (\bar{\mathbf{b}}^*)^\top\| \leq \epsilon \|\mathbf{a}^* (\bar{\mathbf{b}}^*)^\top\|$$

*which we refer to as $\epsilon$-accurate recovery of the global minimizer.*

Theorem A.14 follows by recursively applying Lemma A.16 for $T$ iterations. We start by computing the update rule for $\mathbf{a}$ as in Lemma A.15. Using Lemma A.15, we analyze the convergence of $\mathbf{a}$ in Lemma A.16. We also show the global error that can be achieved by FFA-LoRA within this setting in Proposition A.17.

**Lemma A.15.** (Update for $\mathbf{a}$) In RoLoRA for linear regressor, the update for $\mathbf{a}$ and $b$ in each iteration is:

$$\mathbf{b}^{t+1} = \bar{\mathbf{b}} = \bar{\mathbf{b}}^* \mathbf{a}^{*\top} \mathbf{a}^t \tag{247}$$

$$\mathbf{a}^{t+1} = \hat{\mathbf{a}} = \frac{\mathbf{a}^t - 2\eta(\mathbf{a}^t \bar{\mathbf{b}}^\top \bar{\mathbf{b}} - \mathbf{a}^* \bar{\mathbf{b}}^{*\top} \bar{\mathbf{b}})}{\|\hat{\mathbf{a}}^+\|} \tag{248}$$

*where $\bar{\mathbf{b}}^* = \sum_{i=1}^{N} \mathbf{b}_i^*, \|\hat{\mathbf{a}}^+\| = \|\mathbf{a}^t - 2\eta(\mathbf{a}^t \bar{\mathbf{b}}^\top \bar{\mathbf{b}} - \mathbf{a}^* \bar{\mathbf{b}}^{*\top} \bar{\mathbf{b}})\|$.*

*Proof.* **Minimization for $b_i$.** At the start of each iteration, each client computes the analytic solution for $\mathbf{b}_i$ by fixing $\mathbf{a}$ and solving their local objective $\operatorname{argmin}_{\mathbf{b}_i} \|\mathbf{a}^* \mathbf{b}_i^{*\top} - \mathbf{a} \mathbf{b}_i^\top\|^2$, where $\mathbf{a}^*$ and $\mathbf{a}$ are both unit vectors. Setting $\mathbf{a} = \mathbf{a}^t$, we obtain $\mathbf{b}_i$ such that

$$\mathbf{b}_i = \frac{\mathbf{b}_i^* \mathbf{a}^{*\top} \mathbf{a}}{\mathbf{a}^\top \mathbf{a}} = \mathbf{b}_i^* \mathbf{a}^{*\top} \mathbf{a} \tag{249}$$

(249) follows since $\mathbf{a}^\top \mathbf{a} = 1$.

**Aggregation for $b_i$.** The server simply computes the average of $\{\mathbf{b}_i\}_{i=1}^{N}$ and gets

$$\bar{\mathbf{b}} = \sum_{i=1}^{N} \mathbf{b}_i = \sum_{i=1}^{N} \mathbf{b}_i^* \mathbf{a}^{*\top} \mathbf{a} = \bar{\mathbf{b}}^* \mathbf{a}^{*\top} \mathbf{a} \tag{250}$$

The server then sends $\bar{\mathbf{b}}$ to clients for synchronization.

**Gradient Descent for $\hat{\mathbf{a}}$.** In this step, each client fixes $\mathbf{b}_i$ to $\bar{\mathbf{b}}$ received from the server and update $\mathbf{a}$ using gradient descent. With the following gradient

$$\nabla_{\mathbf{a}} l_i(\mathbf{a}, \bar{\mathbf{b}}) = 2(\mathbf{a}\bar{\mathbf{b}}^\top\bar{\mathbf{b}} - \mathbf{a}^*{\mathbf{b}_i^*}^\top\bar{\mathbf{b}}) \tag{251}$$

Thus, with step size $\eta$, $\mathbf{a}$ is updated such as

$$\hat{\mathbf{a}}^+ = \mathbf{a} - \frac{\eta}{N}\sum_{i=1}^{N}\nabla_{\mathbf{a}}l_i(\mathbf{a}, \bar{\mathbf{b}})$$

$$= \mathbf{a} - 2\frac{\eta}{N}\sum_{i=1}^{N}(\mathbf{a}\bar{\mathbf{b}}^\top\bar{\mathbf{b}} - \mathbf{a}^*{\mathbf{b}_i^*}^\top\bar{\mathbf{b}})$$

$$= \mathbf{a} - 2\eta(\mathbf{a}\bar{\mathbf{b}}^\top\bar{\mathbf{b}} - \mathbf{a}^*{\bar{\mathbf{b}}^*}^\top\bar{\mathbf{b}}) \tag{252}$$

$$\hat{\mathbf{a}} = \frac{\mathbf{a} - 2\eta(\mathbf{a}\bar{\mathbf{b}}^\top\bar{\mathbf{b}} - \mathbf{a}^*{\bar{\mathbf{b}}^*}^\top\bar{\mathbf{b}})}{\|\hat{\mathbf{a}}^+\|} \tag{253}$$

$\square$

**Lemma A.16.** *Let $\delta_t = |\sin\theta(\mathbf{a}^*, \mathbf{a}^t)|$ be the angle distance between $\mathbf{a}^*$ and $\mathbf{a}^t$. Assume that Assumption A.12 holds and $\delta_t \leq \delta_{t-1} \leq \cdots \leq \delta_0$, if $\eta \leq \frac{1}{2L_{\max}^2}$, then*

$$|\sin\theta(\mathbf{a}^{t+1}, \mathbf{a}^*)| = \delta_{t+1} \leq \delta_t \cdot (1 - 2\eta(1 - (\delta^0)^2)\|\bar{\mathbf{b}}^*\|^2) \tag{254}$$

*Proof.* From Lemma A.15, $\mathbf{a}^{t+1}$ and $\mathbf{b}^{t+1}$ are computed as follows:

$$\mathbf{b}^{t+1} = \bar{\mathbf{b}} = \bar{\mathbf{b}}^*{\mathbf{a}^*}^\top\mathbf{a}^t \tag{255}$$

$$\mathbf{a}^{t+1} = \frac{\mathbf{a}^t - 2\eta(\mathbf{a}^t\bar{\mathbf{b}}^\top\bar{\mathbf{b}} - \mathbf{a}^*{\bar{\mathbf{b}}^*}^\top\bar{\mathbf{b}})}{\|\mathbf{a}^t - 2\eta(\mathbf{a}^t\bar{\mathbf{b}}^\top\bar{\mathbf{b}} - \mathbf{a}^*{\bar{\mathbf{b}}^*}^\top\bar{\mathbf{b}})\|} \tag{256}$$

Note that $\mathbf{a}^t$ and $\mathbf{a}^{t+1}$ are both unit vectors. Now, we multiply both sides of Equation (256) by the projection operator $\mathbf{P} = \mathbf{I}_d - \mathbf{a}^*(\mathbf{a}^*)^\top$, which is the projection to the direction orthogonal to $\mathbf{a}^*$. We obtain:

$$\mathbf{P}\mathbf{a}^{t+1} = \frac{\mathbf{P}\mathbf{a}^t - 2\eta\mathbf{P}\mathbf{a}^t\bar{\mathbf{b}}^\top\bar{\mathbf{b}} + \mathbf{P}\mathbf{a}^*{\bar{\mathbf{b}}^*}^\top\bar{\mathbf{b}}}{\|\mathbf{a}^t - 2\eta(\mathbf{a}^t\bar{\mathbf{b}}^\top\bar{\mathbf{b}} - \mathbf{a}^*{\bar{\mathbf{b}}^*}^\top\bar{\mathbf{b}})\|} \tag{257}$$

$$= \frac{\mathbf{P}\mathbf{a}^t - 2\eta\mathbf{P}\mathbf{a}^t\bar{\mathbf{b}}^\top\bar{\mathbf{b}}}{\|\mathbf{a}^t - 2\eta(\mathbf{a}^t\bar{\mathbf{b}}^\top\bar{\mathbf{b}} - \mathbf{a}^*{\bar{\mathbf{b}}^*}^\top\bar{\mathbf{b}})\|} \tag{258}$$

The third term of the numerator is canceled since $\mathbf{P}\mathbf{a}^* = (\mathbf{I}_d - \mathbf{a}^*(\mathbf{a}^*)^\top)\mathbf{a}^* = 0$. Thus,

$$\|\mathbf{P}\mathbf{a}^{t+1}\| \leq \frac{\|\mathbf{P}\mathbf{a}^t\|\,|1 - 2\eta\bar{\mathbf{b}}^\top\bar{\mathbf{b}}|}{\|\mathbf{a}^t - 2\eta(\mathbf{a}^t\bar{\mathbf{b}}^\top\bar{\mathbf{b}} - \mathbf{a}^*{\bar{\mathbf{b}}^*}^\top\bar{\mathbf{b}})\|} \tag{259}$$

Let $\delta_t = |\sin\theta(\mathbf{a}^*, \mathbf{a}^t)|$. Equation (257) becomes:

$$\delta_{t+1} \leq \delta^t \frac{|1 - 2\eta\bar{\mathbf{b}}^\top\bar{\mathbf{b}}|}{\|\mathbf{a}^t - 2\eta(\mathbf{a}^t\bar{\mathbf{b}}^\top\bar{\mathbf{b}} - \mathbf{a}^*{\bar{\mathbf{b}}^*}^\top\bar{\mathbf{b}})\|} \tag{260}$$

$$= \delta_t \frac{|1 - 2\eta\bar{\mathbf{b}}^\top\bar{\mathbf{b}}|}{\|\mathbf{a}^t(1 - 2\eta\bar{\mathbf{b}}^\top\bar{\mathbf{b}}) + 2\eta\mathbf{a}^*{\bar{\mathbf{b}}^*}^\top\bar{\mathbf{b}}\|} \tag{261}$$

$$= \delta_t C \tag{262}$$

Obviously $C \geq 0$. We drop the superscript $t$ when it is clear from context. Note that we have

$$C^2 = \frac{|1 - 2\eta\bar{\mathbf{b}}^\top\bar{\mathbf{b}}|^2}{\|\mathbf{a}(1 - 2\eta\bar{\mathbf{b}}^\top\bar{\mathbf{b}}) + 2\eta\mathbf{a}^*\bar{\mathbf{b}}^{*\top}\bar{\mathbf{b}}\|^2} \tag{263}$$

$$= \frac{|1 - 2\eta\bar{\mathbf{b}}^\top\bar{\mathbf{b}}|^2}{(1 - 2\eta\bar{\mathbf{b}}^\top\bar{\mathbf{b}})^2\mathbf{a}^\top\mathbf{a} + 4\eta^2(\bar{\mathbf{b}}^{*\top}\bar{\mathbf{b}})^2 + 4\eta(1 - 2\eta\bar{\mathbf{b}}^\top\bar{\mathbf{b}})\mathbf{a}^\top\mathbf{a}^*\bar{\mathbf{b}}^{*\top}\bar{\mathbf{b}}} \tag{264}$$

$$= \frac{|1 - 2\eta\bar{\mathbf{b}}^\top\bar{\mathbf{b}}|^2}{(1 - 2\eta\bar{\mathbf{b}}^\top\bar{\mathbf{b}})^2 + 4\eta^2(\bar{\mathbf{b}}^{*\top}\bar{\mathbf{b}})^2 + 4\eta(1 - 2\eta\bar{\mathbf{b}}^\top\bar{\mathbf{b}})\mathbf{a}^\top\mathbf{a}^*\bar{\mathbf{b}}^{*\top}\bar{\mathbf{b}}} \tag{265}$$

Recall that $\bar{\mathbf{b}} = \bar{\mathbf{b}}^*\mathbf{a}^{*\top}\mathbf{a} = \bar{\mathbf{b}}^*\cos\theta(\mathbf{a}^*, \mathbf{a})$, (265) becomes:

$$C^2 = \frac{|1 - 2\eta\bar{\mathbf{b}}^\top\bar{\mathbf{b}}|^2}{(1 - 2\eta\bar{\mathbf{b}}^\top\bar{\mathbf{b}})^2 + 4\eta^2(\bar{\mathbf{b}}^{*\top}\bar{\mathbf{b}})^2 + 4\eta(1 - 2\eta\bar{\mathbf{b}}^\top\bar{\mathbf{b}})\mathbf{a}^\top\mathbf{a}^*\bar{\mathbf{b}}^{*\top}\bar{\mathbf{b}}} \tag{266}$$

$$= \frac{|1 - 2\eta\bar{\mathbf{b}}^\top\bar{\mathbf{b}}|^2}{1 + 4\eta^2\bar{\mathbf{b}}^\top\bar{\mathbf{b}}(\bar{\mathbf{b}}^{*\top}\bar{\mathbf{b}}^* - \bar{\mathbf{b}}^\top\bar{\mathbf{b}})} \tag{267}$$

$$\leq (1 - 2\eta\bar{\mathbf{b}}^\top\bar{\mathbf{b}})^2 \tag{268}$$

$$= (1 - 2\eta\|\bar{\mathbf{b}}\|^2)^2 \tag{269}$$

where (268) holds because $\bar{\mathbf{b}}^{*\top}\bar{\mathbf{b}}^* - \bar{\mathbf{b}}^\top\bar{\mathbf{b}} = (1 - \cos^2\theta(\mathbf{a}^*, \mathbf{a}))\bar{\mathbf{b}}^{*\top}\bar{\mathbf{b}}^* \geq 0$. Equation (269) implies $C \leq 1 - 2\eta\|\bar{\mathbf{b}}\|^2$ if $2\eta\|\bar{\mathbf{b}}\|^2 \leq 1$, which can be ensured by choosing a proper step size $\eta \leq \frac{1}{2L_{max}^2} \leq \frac{1}{2\|\bar{\mathbf{b}}\|^2}$. Now by the assumption that $\delta_t \leq \delta_{t-1} \leq \cdots \leq \delta_0$,

$$C \leq 1 - 2\eta\|\bar{\mathbf{b}}\|^2 \tag{270}$$

$$= 1 - 2\eta\cos^2\theta(\mathbf{a}^*, \mathbf{a})\|\bar{\mathbf{b}}^*\|^2 \tag{271}$$

$$= 1 - 2\eta(1 - (\delta^t)^2)\|\bar{\mathbf{b}}^*\|^2 \tag{272}$$

$$\leq 1 - 2\eta(1 - (\delta^0)^2)\|\bar{\mathbf{b}}^*\|^2 \tag{273}$$

Summarizing, we obtain $\delta^{t+1} \leq \delta^t C \leq \delta^t(1 - 2\eta(1 - (\delta^0)^2)\|\bar{\mathbf{b}}^*\|^2)$. $\qquad\square$

**Proposition A.17.** *(FFA-LoRA lower bound) Suppose we are in the setting described in Section A.5. For any set of ground truth parameters $(\mathbf{a}^*, \{\mathbf{b}_i^*\}_{i=1}^N)$, initialization $\mathbf{a}^0$, initial angle distance $\delta_0 \in (0, 1)$, we apply Freezing-A scheme to obtain a shared global model $(\mathbf{a}^0, \mathbf{b}^{FFA})$, where $\mathbf{b}^{FFA} = \bar{\mathbf{b}}^*\mathbf{a}^{*\top}\mathbf{a}^0$. The global loss is*

$$\frac{1}{N}\sum_{i=1}^N l_i(\mathbf{a}^0, \mathbf{b}^{FFA}) = \gamma^2 + \|\bar{\mathbf{b}}^*\|^2\delta_0^2 \tag{274}$$

*Proof.* Through single step of minimization on $\mathbf{b}_i$ and corresponding aggregation, the minimum of the global objective is reached by FFA-LoRA. $\bar{\mathbf{b}}^{FFA}$ is obtained through:

$$\mathbf{b}_i = \frac{\mathbf{b}_i^*\mathbf{a}^{*\top}\mathbf{a}^0}{\mathbf{a}^{0\top}\mathbf{a}^0} = \mathbf{b}_i^*\mathbf{a}^{*\top}\mathbf{a}^0 \tag{275}$$

$$\mathbf{b}^{FFA} = \frac{1}{N}\sum_{i=1}^N \mathbf{b}_i = \bar{\mathbf{b}}^*\mathbf{a}^{*\top}\mathbf{a}^0 \tag{276}$$

Next we compute the global loss with a shared global model $(\mathbf{a}^0, \bar{\mathbf{b}}^{FFA})$. Note that we use Tr(.) to denote the trace of a

matrix.

$$\frac{1}{N}\sum_{i=1}^{N}l_i(\mathbf{a}^0, \mathbf{b}^{FFA}) \tag{277}$$

$$= \frac{1}{N}\sum_{i=1}^{N}\|\mathbf{a}^*(\mathbf{b}_i^*)^\top - \mathbf{a}^0(\mathbf{b}^{FFA})^\top\|^2 \tag{278}$$

$$= \frac{1}{N}\sum_{i=1}^{N}\|\mathbf{a}^*(\mathbf{b}_i^*)^\top - \mathbf{a}^*(\bar{\mathbf{b}}^*)^\top + \mathbf{a}^*(\bar{\mathbf{b}}^*)^\top - \mathbf{a}^0(\mathbf{b}^{FFA})^\top\|^2 \tag{279}$$

$$= \frac{1}{N}\sum_{i=1}^{N}(\|\mathbf{a}^*(\mathbf{b}_i^*)^\top - \mathbf{a}^*(\bar{\mathbf{b}}^*)^\top\|^2 + \|\mathbf{a}^*(\bar{\mathbf{b}}^*)^\top - \mathbf{a}^0(\mathbf{b}^{FFA})^\top\|^2$$
$$+ 2\mathrm{Tr}((\mathbf{a}^*(\mathbf{b}_i^*)^\top - \mathbf{a}^*(\bar{\mathbf{b}}^*)^\top)^\top(\mathbf{a}^*(\bar{\mathbf{b}}^*)^\top - \mathbf{a}^0(\mathbf{b}^{FFA})^\top)) \tag{280}$$

$$= \frac{1}{N}\sum_{i=1}^{N}(\|\mathbf{a}^*(\mathbf{b}_i^*)^\top - \mathbf{a}^*(\bar{\mathbf{b}}^*)^\top\|^2 + \|\mathbf{a}^*(\bar{\mathbf{b}}^*)^\top - \mathbf{a}^0(\mathbf{b}^{FFA})^\top\|^2)$$
$$+ 2\mathrm{Tr}((\mathbf{a}^*\frac{1}{N}\sum_{i=1}^{N}(\mathbf{b}_i^*)^\top - \mathbf{a}^*(\bar{\mathbf{b}}^*)^\top)^\top(\mathbf{a}^*(\bar{\mathbf{b}}^*)^\top - \mathbf{a}^0(\mathbf{b}^{FFA})^\top)) \tag{281}$$

$$= \frac{1}{N}\sum_{i=1}^{N}(\|\mathbf{a}(\mathbf{b}_i^* - \bar{\mathbf{b}}^*)^\top\|^2 + \|\mathbf{a}^*(\bar{\mathbf{b}}^*)^\top - \mathbf{a}^0{\mathbf{a}^0}^\top\mathbf{a}^*(\bar{\mathbf{b}}^*)^\top\|^2) \tag{282}$$

$$= \frac{1}{N}\sum_{i=1}^{N}\|\mathbf{b}_i^* - \bar{\mathbf{b}}^*\|^2 + \frac{1}{N}\sum_{i=1}^{N}\|(\mathbf{I}_d - \mathbf{a}^0{\mathbf{a}^0}^\top)\mathbf{a}^*(\bar{\mathbf{b}}^*)^\top\|^2 \tag{283}$$

$$= \frac{1}{N}\sum_{i=1}^{N}\|\mathbf{b}_i^* - \bar{\mathbf{b}}^*\|^2 + \frac{1}{N}\sum_{i=1}^{N}\|(\mathbf{I}_d - \mathbf{a}^0{\mathbf{a}^0}^\top)\mathbf{a}^*\|^2\|\bar{\mathbf{b}}^*\|^2 \tag{284}$$

$$= \gamma^2 + \|\bar{\mathbf{b}}^*\|^2\delta_0^2 \tag{285}$$

where (282) holds since the last term is 0, (283) and (284) hold since $\|\mathbf{u}\mathbf{v}^\top\| = \|\mathbf{u}\| \cdot \|\mathbf{v}^\top\|$ for vector $\mathbf{u}$ and $\mathbf{v}$, (285) holds because of Definition A.13. $\qquad\square$

**Proof of Theorem A.14**

*Proof.* In order to achieve $\epsilon$-recovery of $\mathbf{a}^*$, we need

$$\delta^0(1 - c(1 - (\delta^0)^2))^T \le \epsilon \tag{286}$$

$$(1 - c(1 - (\delta^0)^2))^T \le \frac{\epsilon}{\delta^0} \tag{287}$$

$$T\log\left(1 - c(1 - (\delta^0)^2)\right) \le \log(\frac{\epsilon}{\delta^0}) \tag{288}$$

$$\tag{289}$$

We proceed such that

$$T \ge \frac{\log(\frac{\epsilon}{\delta^0})}{\log\left(1 - c(1 - (\delta^0)^2)\right)} \tag{290}$$

$$> \frac{\log(\frac{\epsilon}{\delta^0})}{-c(1 - (\delta^0)^2)} \tag{291}$$

$$= \frac{1}{c(1 - (\delta^0)^2)}\log(\frac{\delta^0}{\epsilon}) \tag{292}$$

where (291) follows by using $\log(1-x) < -x$ for $|x| < 1$.

Now we show the convergence to the global minimizer. Recall that $\mathbf{b}^{T+1} = \bar{\mathbf{b}}^* \mathbf{a}^{*\top} \mathbf{a}^T$ and $\delta^T = \|(\mathbf{I}_d - \mathbf{a}^T(\mathbf{a}^T)^\top)\mathbf{a}^*\|$, we have

$$\|\mathbf{a}^T(\mathbf{b}^{T+1})^\top - \mathbf{a}^*\bar{\mathbf{b}}^{*\top}\| = \|\mathbf{a}^T(\mathbf{a}^T)^\top \mathbf{a}^*\bar{\mathbf{b}}^{*\top} - \mathbf{a}^*\bar{\mathbf{b}}^{*\top}\| \tag{293}$$

$$= \|(\mathbf{a}^T(\mathbf{a}^T)^\top - \mathbf{I}_d)\mathbf{a}^*\bar{\mathbf{b}}^{*\top}\| \tag{294}$$

$$= \|(\mathbf{I}_d - \mathbf{a}^T(\mathbf{a}^T)^\top)\mathbf{a}^*\| \cdot \|\bar{\mathbf{b}}^*\| \tag{295}$$

$$\leq \epsilon\|\bar{\mathbf{b}}^*\| \tag{296}$$

$$= \epsilon\|\mathbf{a}^*\bar{\mathbf{b}}^{*\top}\| \tag{297}$$

where (297) is due to the fact that $\|\mathbf{x}\mathbf{y}^\top\| = \|\mathbf{x}\|\|\mathbf{y}\|$ and $\|\mathbf{a}^*\| = 1$.

$\square$

Proposition A.17 shows that for any $\delta_0 \in (0,1)$, the global objective of FFA-LoRA is given by (285), comprising two terms: $\gamma^2$, reflecting the heterogeneity of $\{\mathbf{b}_i^*\}_{i=1}^N$, and $\|\bar{\mathbf{b}}^*\|^2\delta_0^2$, due to the angular distance between $\mathbf{a}^0$ and $\mathbf{a}^*$. By Theorem A.14, RoLoRA achieves $\epsilon$-accurate recovery of the global minimizer, with global loss upper bounded by $\gamma^2 + \|\bar{\mathbf{b}}^*\|^2\epsilon^2$, since RoLoRA reduces the angular distance loss from $\|\bar{\mathbf{b}}^*\|^2\delta_0^2$ to $\|\bar{\mathbf{b}}^*\|^2\epsilon^2$. We can make $\epsilon$ arbitrarily small by increasing the iterations.

# B. Experiments

## B.1. Hyper-parameters for GLUE task

| | SST-2 | QNLI | MNLI | QQP | RTE |
|---|---|---|---|---|---|
| Total comm. rounds | 500 | 500 | 500 | 500 | 200 |
| Batch Size | 64 | 32 | 32 | 32 | 32 |
| Local Epochs | 20 | 20 | 20 | 20 | 20 |

*Table 4.* Hyper-parameters configurations. Note the total communication rounds are for the setting with 3 clients. When increasing the number of clients, we decrease the total communication rounds accordingly to maintain a constant sample count used during fine-tuning

We show the hyper-parameter configurations for each dataset in Table 4.

## B.2. More Experimental Results

### B.2.1. EFFECT OF NUMBER OF CLIENTS

Table 5 shows the selected layer set attached with LoRA modules for Table 1. We present Table 1 with the results of FlexLoRA (Bai et al., 2024) added in Table 6.

| | Layer Attributes | SST-2 | QNLI | MNLI | QQP | RTE |
|---|---|---|---|---|---|---|
| $\mathcal{P}_2$ | Type | $W_v, W_q$ | $W_v, W_q$ | $W_v, W_q$ | $W_v, W_q$ | $W_v, W_q$ |
| | Index | $\{18,\ldots,23\}$ | $\{15,\ldots,23\}$ | $\{15,\ldots,23\}$ | $\{15,\ldots,23\}$ | $\{16,\ldots,23\}$ |

*Table 5.* The selected layer set attached with LoRA modules for Table 1

### B.2.2. EFFECT OF NUMBER OF LORA PARAMETERS

In Table 7, we include the details about layers attached with LoRA adapters for different budget of finetuning parameters, for each dataset.

### B.2.3. ALIGN THE COMMUNICATION COST

In Table 8, we include the details about layers attached with LoRA adapters.

| rank | Clients num | Method | SST-2 | QNLI | MNLI | QQP | RTE | Avg. |
|---|---|---|---|---|---|---|---|---|
| 4 | 3 | LoRA | $\mathbf{95.62}_{\pm 0.17}$ | $91.59_{\pm 0.21}$ | $\mathbf{86.20}_{\pm 0.05}$ | $86.13_{\pm 0.10}$ | $81.46_{\pm 1.22}$ | 88.20 |
| | | FFA-LoRA | $95.18_{\pm 0.09}$ | $91.35_{\pm 0.32}$ | $84.58_{\pm 0.21}$ | $85.50_{\pm 0.25}$ | $81.10_{\pm 0.33}$ | 87.48 |
| | | FlexLoRA | $94.91_{\pm 0.18}$ | $90.16_{\pm 0.49}$ | $85.16_{\pm 0.69}$ | $85.69_{\pm 0.17}$ | $79.3_{\pm 1.05}$ | 87.04 |
| | | RoLoRA | $95.49_{\pm 0.16}$ | $\mathbf{91.64}_{\pm 0.30}$ | $85.70_{\pm 0.04}$ | $\mathbf{86.14}_{\pm 0.06}$ | $\mathbf{82.43}_{\pm 0.84}$ | $\mathbf{88.28}$ |
| 4 | 20 | LoRA | $94.3_{\pm 0.27}$ | $86.67_{\pm 2.02}$ | $78.55_{\pm 7.31}$ | $83.1_{\pm 0.04}$ | $51.87_{\pm 3.24}$ | 78.90 |
| | | FFA-LoRA | $93.88_{\pm 0.06}$ | $89.11_{\pm 0.19}$ | $80.99_{\pm 1.74}$ | $83.92_{\pm 0.2}$ | $57.16_{\pm 1.46}$ | 80.01 |
| | | FlexLoRA | $90.97_{\pm 1.78}$ | $54.36_{\pm 0.36}$ | $53.30_{\pm 14.59}$ | $69.18_{\pm 10.39}$ | $53.19_{\pm 1.45}$ | 64.20 |
| | | RoLoRA | $\mathbf{94.88}_{\pm 0.18}$ | $\mathbf{90.35}_{\pm 0.37}$ | $\mathbf{85.28}_{\pm 1.04}$ | $\mathbf{85.83}_{\pm 0.1}$ | $\mathbf{78.82}_{\pm 1.7}$ | $\mathbf{87.03}$ |
| 4 | 50 | LoRA | $93.00_{\pm 0.35}$ | $78.13_{\pm 5.13}$ | $52.64_{\pm 15.07}$ | $77.60_{\pm 1.47}$ | $52.23_{\pm 1.1}$ | 70.72 |
| | | FFA-LoRA | $93.23_{\pm 0.12}$ | $85.05_{\pm 0.34}$ | $69.97_{\pm 5.57}$ | $78.44_{\pm 0.41}$ | $55.72_{\pm 1.99}$ | 76.48 |
| | | FlexLoRA | $54.08_{\pm 5.5}$ | $55.4_{\pm 2.03}$ | $39.14_{\pm 2.35}$ | $72.00_{\pm 7.64}$ | $52.71_{\pm 0.00}$ | 54.67 |
| | | RoLoRA | $\mathbf{94.80}_{\pm 0.17}$ | $\mathbf{90.00}_{\pm 0.63}$ | $\mathbf{82.98}_{\pm 3.36}$ | $\mathbf{85.71}_{\pm 0.18}$ | $\mathbf{75.57}_{\pm 2.88}$ | $\mathbf{85.81}$ |
| 8 | 50 | LoRA | $93.00_{\pm 0.23}$ | $79.87_{\pm 1.52}$ | $56.96_{\pm 2.02}$ | $77.45_{\pm 1.97}$ | $53.79_{\pm 6.57}$ | 64.03 |
| | | FFA-LoRA | $92.74_{\pm 0.13}$ | $83.69_{\pm 0.75}$ | $64.51_{\pm 1.92}$ | $79.71_{\pm 2.04}$ | $53.07_{\pm 1.3}$ | 72.46 |
| | | FlexLoRA | $50.92_{\pm 0.00}$ | $56.92_{\pm 1.04}$ | $37.43_{\pm 2.80}$ | $66.40_{\pm 4.74}$ | $52.59_{\pm 0.21}$ | 52.85 |
| | | RoLoRA | $\mathbf{94.53}_{\pm 0.17}$ | $\mathbf{90.1}_{\pm 0.45}$ | $\mathbf{85.17}_{\pm 0.41}$ | $\mathbf{85.25}_{\pm 0.13}$ | $\mathbf{76.3}_{\pm 4.9}$ | $\mathbf{86.27}$ |

*Table 6.* Results for four methods with RoBERTa-Large models with varying client numbers (3, 20, 50), maintaining a constant sample count during fine-tuning.

| | Layer Attributes | SST-2 | QNLI | MNLI | QQP | RTE |
|---|---|---|---|---|---|---|
| $\mathcal{P}_1$ | Type | $W_v$ | $W_v, W_q$ | $W_v, W_q$ | $W_v, W_q$ | $W_v, W_q$ |
| | Index | $\{21, \ldots, 23\}$ | $\{21, \ldots, 23\}$ | $\{21, \ldots, 23\}$ | $\{21, \ldots, 23\}$ | $\{21, \ldots, 23\}$ |
| $\mathcal{P}_2$ | Type | $W_v, W_q$ | $W_v, W_q$ | $W_v, W_q$ | $W_v, W_q$ | $W_v, W_q$ |
| | Index | $\{18, \ldots, 23\}$ | $\{15, \ldots, 23\}$ | $\{15, \ldots, 23\}$ | $\{15, \ldots, 23\}$ | $\{16, \ldots, 23\}$ |
| $\mathcal{P}_3$ | Type | $W_v, W_q$ | $W_v, W_q$ | $W_v, W_q$ | $W_v, W_q$ | $W_v, W_q$ |
| | Index | $\{0, \ldots, 23\}$ | $\{12, \ldots, 23\}$ | $\{12, \ldots, 23\}$ | $\{12, \ldots, 23\}$ | $\{12, \ldots, 23\}$ |

*Table 7.* The selected layer set attached with LoRA for the setup of Figure 4

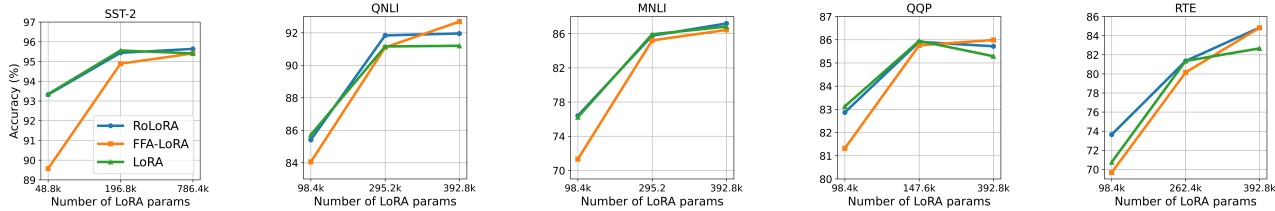

*Figure 6.* Results with RoBERTa-Large models on GLUE of different budget of finetuning parameters. It involves 3 clients using rank 8.

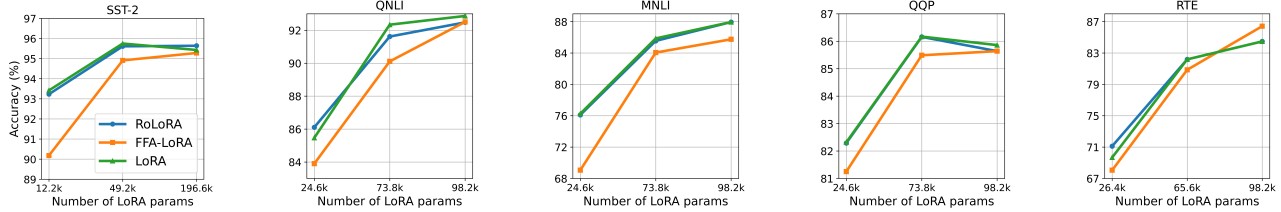

*Figure 7.* Results with RoBERTa-Large models on GLUE of different budget of finetuning parameters. It involves 3 clients using rank 2.

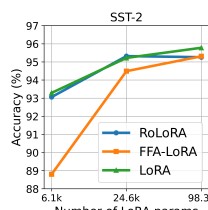 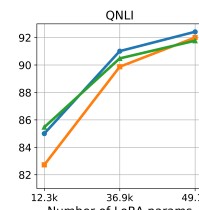 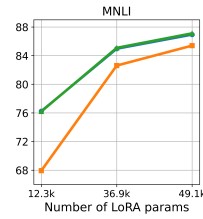 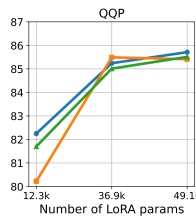 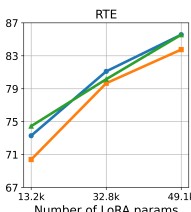

*Figure 8.* Results with RoBERTa-Large models on GLUE of different budget of finetuning parameters. It involves 3 clients using rank 1.

|  | Communication Cost | LoRA | FFA-LoRA | RoLoRA |
|---|---|---|---|---|
| 187.5 KB | Type | $W_v, W_q$ | $W_v, W_q$ | $W_v, W_q$ |
|  | Index | $\{21, \dots, 23\}$ | $\{18, \dots, 23\}$ | $\{18, \dots, 23\}$ |
| 250 KB | Type | $W_v, W_q$ | $W_v, W_q$ | $W_v, W_q$ |
|  | Index | $\{20, \dots, 23\}$ | $\{16, \dots, 23\}$ | $\{16, \dots, 23\}$ |

*Table 8.* The selected layer set attached with LoRA modules for the setup of Figure 5

### B.2.4. COMMONSENSE REASONING TASKS

We present the configurations for Table 2 in Table 9. We show the results under the same setup but using rank-2 LoRA modules in Table 10.

| Total comm. rounds | Batch size | Local Epochs | Layer type attached with LoRA | Layer index attached with LoRA |
|---|---|---|---|---|
| 10 | 1 | 30 | $W_k, W_v, W_q, W_o$ | $\{26, \dots, 31\}$ |

*Table 9.* Configurations for Commonsense Reasoning tasks.

### B.2.5. LANGUAGE GENERATION TASKS

**Model, Datasets and Metrics.** We evaluate the performance of three federated finetuning methods with the model Llama-2-7B (Touvron et al., 2023), on two datasets: CodeAlpaca (Chaudhary, 2023) for coding tasks, and Alpaca (Taori et al., 2023) for general instruction-following tasks. Using HumanEval (Chen et al., 2021) as the metric for CodeAlpaca, we assess the model's ability to generate accurate code solutions. For Alpaca, we employ MMLU (Massive Multitask Language Understanding) (Hendrycks et al., 2021) to evaluate the model's performance across diverse domains. This provides an assessment of Llama-2-7B's coding proficiency, and general language capabilities when finetuning in the federated setting. We show the setup in Table 11.

**Results** Table 12 presents the evaluation results of the Llama-2-7B model using three methods, across two tasks: HumanEval, and MMLU. The metrics reported include the average and standard deviation of performance over five seeds, with 50 clients involved. The results show that RoLoRA achieves the highest scores across most metrics, demonstrating slightly improved performance compared to LoRA and FFA-LoRA. The improvements are more evident in certain subcategories of the MMLU dataset.

| | BoolQ | PIQA | SIQA | HellaSwag | WinoGrande | ARC-e | ARC-c | OBQA |
|---|---|---|---|---|---|---|---|---|
| LoRA | $34.36_{\pm 16.90}$ | $42.87_{\pm 14.05}$ | $19.12_{\pm 4.22}$ | $26.21_{\pm 1.91}$ | $47.2_{\pm 0.64}$ | $10.31_{\pm 5.96}$ | $9.84_{\pm 6.13}$ | $12.33_{\pm 7.46}$ |
| FFA-LoRA | $44.04_{\pm 11.48}$ | $51.46_{\pm 9.81}$ | $25.38_{\pm 11.27}$ | $23.86_{\pm 2.67}$ | $46.93_{\pm 1.54}$ | $22.25_{\pm 7.92}$ | $20.65_{\pm 6.33}$ | $20.67_{\pm 5.33}$ |
| RoLoRA | $61.3_{\pm 0.99}$ | $60.81_{\pm 6.35}$ | $37.97_{\pm 5.39}$ | $29.62_{\pm 2.62}$ | $49.59_{\pm 1.2}$ | $37.05_{\pm 2.92}$ | $29.09_{\pm 3.33}$ | $28.93_{\pm 4.64}$ |

*Table 10.* Results with Llama-2-7B models on commonsense reasoning tasks. It involves 50 clients using rank 2.

| Total comm. rounds | Batch size | Local Epochs | Layer type attached with LoRA | Layer index attached with LoRA |
|---|---|---|---|---|
| 100 | 1 | 30 | $W_k, W_v, W_q, W_o$ | $\{24, \ldots, 31\}$ |

*Table 11.* Configurations for language generation tasks.

| | LoRA | FFA-LoRA | RoLoRA |
|---|---|---|---|
| HumanEval | $12.96_{\pm 0.37}$ | $13.29_{\pm 0.21}$ | $13.45_{\pm 0.28}$ |
| MMLU | $45.81_{\pm 0.03}$ | $45.80_{\pm 0.02}$ | $45.93_{\pm 0.01}$ |
| human | $43.26_{\pm 0.04}$ | $43.24_{\pm 0.01}$ | $43.46_{\pm 0.02}$ |
| other | $52.67_{\pm 0.06}$ | $52.72_{\pm 0.05}$ | $52.83_{\pm 0.04}$ |
| social | $51.73_{\pm 0.04}$ | $51.64_{\pm 0.05}$ | $51.81_{\pm 0.04}$ |
| stem | $37.10_{\pm 0.03}$ | $37.12_{\pm 0.01}$ | $37.05_{\pm 0.02}$ |

*Table 12.* Results with Llama-2-7B model on HumanEval, and MMLU. We report the average and std. over five seeds. The number of clients is 50. The metric used across all tasks is accuracy, where higher values indicate better performance.

