# OpenReview forum: "Robust Federated Finetuning of LLMs via Alternating Optimization of LoRA"
_ICML.cc/2025/Conference — Submitted to ICML 2025_

### Official Review · Reviewer_mjth · 2025-02-22

**Overall Recommendation:** 3

**Summary:**

The paper introduces RoLoRA, a federated fine-tuning framework for Large Language Models (LLMs) based on alternating optimization of LoRA adapters. RoLoRA optimizes both up-projection and down-projection matrices in LoRA adapters alternately, ensuring better convergence and adaptability in federated learning. Theoretical analysis demonstrates that RoLoRA achieves exponential convergence to the global optimum in a linear setting, outperforming methods that freeze down-projections. Empirical results on RoBERTa-Large and Llama-2-7B across GLUE, commonsense reasoning, and other tasks show that RoLoRA maintains robustness against increasing client numbers and reduced fine-tuning parameters while reducing communication overhead by half compared to standard LoRA fine-tuning.

**Claims And Evidence:**

The work states "... we explore the necessity of learning down-projection matrices and propose a federated fine-tuning framework with computational and communication advantages." in Lines 44 to 47 (second column). However, I am not entirely certain how RoLoRA is saving on either computation or communication.

**Essential References Not Discussed:**

FLoRA [1] and its baselines should be mentioned in detailed in this paper. According to my understanding, seems like FLoRA is advocating for what this work is finding a solution against. In FLoRA, the correct way to aggregate the A and B LoRA matrices are by first multiplying them for each client, while this works adds A and B matrices separately and then updates the model with the multiplication of those added A and B matrices. Seems like FedIT [2] (mentioned in FLoRA) is similar to RoLoRA as well.

I would appreciate clarification on how RoLoRA is different than FedIT and FLoRA. I would also recommend the authors to add them as baselines.

[1] FLoRA: Federated Fine-Tuning Large Language Models with Heterogeneous Low-Rank Adaptations (Wang et al., NeurIPS 2024)

[2] Towards Building the Federated GPT: Federated Instruction Tuning (Zhang et al., ICASSP 2024)

**Experimental Designs Or Analyses:**

1. In Figure 3, the variance of RoLoRA seems to be higher than the baselines, why is that so? And why is that not the case for larger models, as shown in table 2?

2. With Figure 3, seems like the curves are cut off before all the baselines have converged. Can the authors please produce plots for, say total round count = 100?

**Methods And Evaluation Criteria:**

From Algorithm 1, it seems that the authors are considering 1 round made up of two stages where the first stage trains A matrices and the second stage trains B matrices. In that case, saying that "Please note that in all tasks, we compare the performance of the three methods under the same number of communication rounds." in Lines 321-323 (second column) is questionable, since RoLoRA is not abided by the traditional definition of a federated round.

I would appreciate some clarification on communication cost per round of (a) server to all clients, and (b) one client to server. Seems like it should be exactly the same as original LoRA with Fed AVG setting. However, the lines 185 to 187 state "In each communication round, the number of trainable parameters in the model is effectively halved compared to FedAVG of LoRA."

**Other Comments Or Suggestions:**

Comment #1: Figure 1 can be improved with some more details. The boxes can be labeled with what they are (eg., Big blue box being W and smaller ones being A and B, the cloud icon being a server etc).

Comment #2: A minor comment would be to that the use of "Large" language model to describe RoBERTa-Large might be questionable. It is rather a medium-sized language model.

**Other Strengths And Weaknesses:**

Strengths: The paper is well-written and the work is well-motivated. The solution is elegant.

Weakness #1: The biggest weakness of this work would be the lack of comparison against and discussion of FLoRA [1]. I have discussed it in some more details under "Essential References Not Discussed" part of the review.

[1] FLoRA: Federated Fine-Tuning Large Language Models with Heterogeneous Low-Rank Adaptations (Wang et al., NeurIPS 2024)

Weakness #2: The related work section can be improved. Instead of just mentioning all the related works, I would appreciate it that the authors list down how RoLoRA differs from all those works.

**Questions For Authors:**

The theoretical analysis is insightful and backed by the empirical results, however, I would appreciate some comments about what differences or challenges non-linearity adds to RoLoRA.

Particularly, if we repeat the two-layer non-linear NN experiment on a one-layer linear NN, do we see a bigger accuracy improvement between LoRA/FFA-LoRA and RoLoRA?

**Relation To Broader Scientific Literature:**

The contributions of RoLoRA can advance the understanding and application of federated learning techniques, particularly in the context of fine-tuning large language models. By demonstrating the benefits of alternating optimization and the importance of learning both projection matrices, the authors provide insights and practical methods for improving model performance in federated settings, which can potential pave the way for privacy-preserving finetuning of LLMs.

However, the ideas seem pretty close to what FLoRA [1] has discussed, while the findings are opposite of FLoRA. (More on it in the next section of this review)

[1] FLoRA: Federated Fine-Tuning Large Language Models with Heterogeneous Low-Rank Adaptations (Wang et al., NeurIPS 2024)

**Theoretical Claims:**

1. Why is LoRA rank $r$ is fixed to 1 and not generalized like other dimensions are, for section 5 "Analysis"? I think keeping $r=1$ is skewing the results of theorem 5.4, especially related to sufficient number of samples $m$.

2. For Theorem 5.4, $q$ would grow with the model dimension $d$? (And potentially with $r$ as well?) If so, how do we justify sufficient number of samples $m$ rising with $d$?

---

> ### Author Rebuttal · Authors · 2025-04-01
>
> Thanks for the detailed review. We address your concerns below:
>
> **Federated Round Definition, Communication and Computation Efficiency.**
> Thanks for the helpful comments. In our paper, each communication round refers to one upload/download of either matrix A or B. RoLoRA updates and transmits only one matrix per round, halving both communication and computation compared to FedIT—see [table](https://anonymous.4open.science/r/random-3C54/R4-2.png).
>
> We apologize for any confusion from the use of "iteration" in Algorithm 1, which was for alignment with Algorithm 2 and theoretical clarity. All experiments were rigorosly benchmarked using communication rounds. We will revise the manuscript to clearly distinguish rounds from iterations and add a footnote in Algorithm 1 to clarify this.
>
> **Rank-1 Analysis and Sample Complexity.**
> Thanks for the insightful questions.
>
> Regarding sample complexity, Theorem 5.4 shows that $q$ scales with $d$ for rank-1, and generalizes to $O(dr)$ for higher ranks. This follows from the $\epsilon$-net argument (Eq. 34), where the covering number grows exponentially with dimension $dr$ [1, Sec. IV.A], leading to the updated $q$ (Line 755). The sample complexity can be justified that in models like $Y=Xab^\top$, $2d$ unknown parameters necessitate sample complexity proportional to $d$.
>
> The rank-1 case highlights FFA-LoRA's core limitation—its inability to align down-projection—which persists empirically in higher ranks (see Sec. 5.2), in language tasks. We further discuss the generalization with Reviewer JAAe under Rank-1 Limitation and Generalizability, and note the core issue of constrained expressivity remains unchanged in the higher-rank case.
>
> [1]Nayer, S., & Vaswani, N. (2022). Fast and sample-efficient federated low rank matrix recovery from column-wise linear and quadratic projections. IEEE Trans. Inf. Theory
>
>
> **Higher variance in Fig.3.** Thanks for the question. The higher variance in Fig. 3 before convergence is expected due to different initializations, which can lead to varying optimization trajectories. After convergence, RoLoRA shows low variance, consistent with its stable performance in Tables 1 and 2. FFA-LoRA's variance differs across Tables 1 and 2 mainly due to task difficulty—it's low on simpler tasks (e.g., GLUE, BoolQ) but higher on complex ones (e.g., PIQA, SIQA) due to sensitivity to initialization, aligning with its weaker performance in Table 1 and 2.
>
> **Convergence curve for 100 Communication rounds.** The [figure](https://anonymous.4open.science/r/random-3C54/convergence-100.png) shows the 100-round extension of Fig. 3, where RoLoRA consistently converges faster and achieves the highest accuracy.
>
> Fig. 3 aimed to compare convergence under a fixed sample budget, not full convergence. As Table 2 shows, this budget is sufficient for all methods with 3 clients, but only RoLoRA fully converges with 50 clients—highlighting its efficiency in low-resource settings.
>
> **Differences from FLoRA and FedIT.** Thanks for the questions. FedIT is equivalent to our FedAVG with LoRA baseline ("LoRA" in our paper), and we provide extensive comparisons. We'll explicitly cite and discuss FedIT in the revision.
>
> We already discuss FLoRA (Lines 31, 113, 162). FLoRA shares RoLoRA's motivation but differs in approach: FLoRA aggregates full matrix products of LoRA-A and LoRA-B, while RoLoRA freezes one matrix for efficient, exact updates (Eq. 3-4). See Sec. 3.2 for discussion.
>
> We've added [table](https://anonymous.4open.science/r/random-3C54/flora.png) comparing RoLoRA and FLoRA under IID setting. In the 3-client setting, we ran 500 rounds and scaled rounds down proportionally with more clients to keep the total sample budget fixed. RoLoRA consistently outperforms FLoRA across tasks and client counts. While FLoRA eventually converges (e.g., 83.3% on MNLI after 4000 rounds), it does so much more slowly, highlighting RoLoRA's faster convergence and better scalability.
>
> Please also see [table](https://anonymous.4open.science/r/random-3C54/cost-compare.png) comparing the communication cost and time cost. RoLoRA and FFA-LoRA have the lowest communication/time costs, while FLoRA is much more expensive.
>
> **Fig.1 Improvement and Use of Large in RoBERTa-Large.** Thanks for the suggestions. We've updated the [figure](https://anonymous.4open.science/r/random-3C54/overview.png) and will refer to RoBERTa-Large simply as a language model, without implying scale beyond its name.
>
> **Related Works.** We agree and will revise the related work section to clearly highlight RoLoRA's key differences from prior methods.
>
> **RoLoRA on Linear Model.**
> As suggested, we ran an experiment removing the ReLU from NN on MNIST—see [figure](https://anonymous.4open.science/r/random-3C54/non-linear.png). Across both linear and non-linear settings, all methods perform similarly, with RoLoRA showing modest improvement in the non-linear case, likely due to its better utilization of the added expressiveness from ReLU.

---

### Official Review · Reviewer_HoB2 · 2025-03-03

**Overall Recommendation:** 2

**Summary:**

The paper introduces RoLoRA, a federated fine-tuning framework that employs alternating optimization of LoRA adapters to enhance model expressiveness and robustness. By theoretically and empirically demonstrating the necessity of learning both down-projection and up-projection matrices, the authors show that RoLoRA outperforms existing methods (e.g., FedAVG of LoRA, FFA-LoRA) in terms of accuracy, communication efficiency, and robustness under varying client numbers and parameter budgets. The theoretical analysis on a linear model and extensive experiments on RoBERTa-Large, and Llama-2-7B validate the framework’s advantages.

**Claims And Evidence:**

Yes.

**Essential References Not Discussed:**

No.

**Experimental Designs Or Analyses:**

Yes.

**Methods And Evaluation Criteria:**

Yes.

**Other Comments Or Suggestions:**

No.

**Other Strengths And Weaknesses:**

Strengths:
1. The alternating optimization strategy for LoRA adapters is innovative, addressing the limitations of prior methods that either aggregate matrices inaccurately or freeze critical parameters.
2. The convergence analysis on a simplified linear model, while idealized, offers meaningful insights into the necessity of training both projection matrices. The exponential convergence guarantee strengthens the method’s credibility.

Weaknesses:
My primary concern is that the algorithm analysis is conducted only on a few special cases, such as the signal vector of the LoRA module, a linear regression objective, the homogeneous setting, and the Freezing-A scheme. While these analyses provide valuable insights, there remains a significant gap between these cases and the broader federated LLM fine-tuning framework proposed in this paper. A more comprehensive theoretical analysis covering the full scope of the proposed method would strengthen the work.

In Lemma 5.3, the assumption that \delta^(t)\le  \delta^(t-1)\le...\le \delta^0 appears overly restrictive and may be difficult to satisfy in practice. Additionally, the lemma's description does not specify which algorithm is used to obtain the stated bound. Moreover, since the error bound applies only to a, should b be frozen in this scenario? Clarifying these aspects would enhance the rigor and applicability of the theoretical analysis.

**Questions For Authors:**

Please refer to the weaknesses part.

**Relation To Broader Scientific Literature:**

This paper proposes a federated learning (FL) algorithm specifically designed for LoRA fine-tuning of LLMs. Therefore, existing federated LLM fine-tuning algorithms and broader federated learning literature on LLMs are relevant to this work.

**Theoretical Claims:**

Yes.

---

> ### Author Rebuttal · Authors · 2025-03-31
>
> Thanks for the thoughtful review. We appreciate the recognition of RoLoRA's contributions and the strengths of our alternating optimization strategy, theoretical insights, and empirical results. We address the main concerns below:
>
> **Limited Theoretical Scope.** We appreciate the reviewer's thoughtful feedback regarding the scope of our theoretical analysis. First, we would like to clarify that our convergence analysis, including comparisons between RoLoRA and FFA-LoRA, also covers the heterogeneous setting (Appendix A.5 at Line 1430). The analysis mirrors the homogeneous case: RoLoRA reduces the angle to the ground truth $\mathbf a$, ensuring convergence, while FFA-LoRA's loss remains tied to its initial angle. This has been discussed at Line 281 in maintext.
>
> While our theoretical analysis focuses on the rank-1 LoRA, linear regression setting, this case remains foundational and highly non-trivial. By directly comparing the solutions of RoLoRA and FFA-LoRA under this simplified yet fundamental scenario, we rigorously demonstrate the inherent limitations of FFA-LoRA in representing low-rank updates. This provides critical insights into the expressiveness of FFA-LoRA even in its most basic form, establishing a baseline for understanding its behavior. Our empirical validation on neural network and MNIST shows the theorem can be extended to higher-rank and non-linear setting. Critically, the results on realistic LLM tasks bridges theory and practice, showing that the phenomena observed in our theoretical framework persist in non-linear, non-convex settings. This alignment mirrors methodologies in prior works [1, 2], which adopt simplified setups to distill core principles before validating them in real-world contexts.
>
> A full theoretical comparison of federated LLM algorithms across all settings is currently infeasible. For neural networks, while convergence analyses exist, they rely on loss landscape assumptions (e.g., smoothness, PL-conditions) that preclude direct comparison of optima between algorithms. Instead, our work focuses on a tractable case where direct comparisons are feasible, allowing us to uncover provable limitations that also hold in real-world scenarios.
>
> [1]Collins, Liam, et al. "Exploiting shared representations for personalized federated learning." (ICML2021).
>
> [2]Collins, Liam, et al. "Fedavg with fine tuning: Local updates lead to representation learning." (NeurIPS2022)
>
>
> **Assumptions on decreasing angles.**
> Thank you for pointing this out. To clarify, although we assume a decreasing sequence of angles in Lemma 5.3, this result is used as an **intermediate** step in the proof of Theorem 5.4. In particular, we demonstrate in the proof of Theorem 5.4 at Line 1027 that the angle decreases in the first iteration. Building on this, we apply an inductive hypothesis to show that the decreasing trend holds for all subsequent iterations (see Line 1039). In summary, the decreasing angle sequence is not assumed in the proof of the main result (Theorem 5.4), but is instead derived as part of the argument.
>
> **Algorithm used for theoretical analysis.** Lemma 5.3 is tied to the update rule in Alg. 2. We will revise the lemma's description to make this connection more explicit.
>
> **Error bound on a.** Thank you for the insightful question. The error bounds in Lemma 5.3 and Theorem 5.4 are derived based on Alg. 2, which updates both $\mathbf{a}$ and $\mathbf{b}$. The reason the bound applies specifically to the angle distance of $\mathbf{a}$ is that $\mathbf{b}$ is fully optimized at each client, while $\mathbf{a}$ is updated via gradient descent. This ensures that $\mathbf{b}$ is always optimal with respect to the current $\mathbf{a}$, allowing us to focus the analysis on the convergence behavior of $\mathbf{a}$. We designed RoLoRA for the linear regressor using alternating minimization (for $\mathbf{b}$) and gradient descent (for $\mathbf{a}$) to decouple their updates and eliminate the potential impact of insufficient local updates of $\mathbf{b}$ on the convergence of $\mathbf{a}$. We will clarify this motivation and its implications in the revised manuscript.
>
> We hope these clarifications address the reviewer's concerns. Please feel free to reach out if the reviewer has any further concerns, and we would be glad to discuss them with the reviewer.

---

### Official Review · Reviewer_JAAe · 2025-03-13

**Overall Recommendation:** 4

**Summary:**

This work explores a better approach for training LoRAs in federated learning. Given a LoRA $B A$, the naive approach is to aggregate the updates of both weights simultaneously as in $\mathbb{E}_i[B_i] \mathbb{E}_i[A_i]$ for clients $i$ (which is not equal to $\mathbb{E}_i[B_i A_i]$). In this paper, the authors study the following method: train $B$ and aggregate, train $A$ and aggregate, and repeat (or do it in the opposite order). For linear regressors, $A$ is trained for one step and normalized after aggregation. The authors demonstrate that this algorithm performs better experimentally—even when taking the communication budget into account—and theoretically as well.  They provide a high-probability analysis of this algorithm (with respect to normal initialization), and show that the LoRA $BA$ converges to the optimal LoRA $B^* A^*$ to an arbitrary error that depends on the number of iterations. On the other hand, other competitive approaches, such as LoRA-FFA, can only converge within an error proportional to $\lVert B^*\rVert$.

## update after rebuttal

I thank the authors for clarifying most of my concerns. I stand by my recommendation for acceptance.

**Claims And Evidence:**

The authors claim that their method performs better than the baselines and converges to arbitrary error as the number of iterations increases. They also claim that FFA-LoRA is less robust than vanilla FedAvg. Furthermore, they claim that their method is robust across various tasks, number of clients, and number of parameters while still demonstrating communication efficiency.
Indeed, the experiments and the analysis are comprehensive and support these claims well in my opinion.

The authors mention in the introduction that their alternating algorithm is inspired by prior works in multitask linear representation learning, but its application in the context of LoRAs is novel.
I initially thought that this could have been explored before in the literature, but the novelty seems to be true based on a quick literature review (see essential references section below), at least within the context of LoRAs and not taking into account the prior works mentioned by the authors.

**Essential References Not Discussed:**

Despite thinking that a similar method might have been proposed before, I did not really find a similar algorithm after doing a quick literature search.
Perhaps the closest work that hasn’t been cited by the authors is [1].
Still, they only alternate the minimization within the same round and do not alternate the aggregations themselves.
Another work that is not directly relevant but interestingly shares some similarity in their algorithm is [2] (if you take the gradient wrt aggregated local parameters and normalize the global update, it is similar).
Based on the above, I believe that the authors have not really missed any essential references, but I have put these references for their interest.

[1] *Federated Matrix Factorization: Algorithm Design and Application to Data Clustering*. Wang & Chang. 2020.

[2] *Partially Personalized Federated Learning: Breaking the Curse of Data Heterogeneity*. Mishchenko et al. 2023.

**Experimental Designs Or Analyses:**

The experimental design is sound. The experiments are federated learning setups of well-known language tasks (evenly splitting dataset among clients). This is good for demonstrating the shortcomings of the baselines that RoLoRA aims to fix. The analysis is done under a setup that is realistic to some extent (normal initialization of LoRA and bounded optimum $\lVert B^* \rVert$).

However, the analysis is done for rank-1 LoRAs and the optimal output is linear wrt some optimal LoRA.
It would be interesting to know whether the analysis can be extended to the general case for rank > 1 and whether the assumption of bounded optimum stil suffices as it is when fine-tuning a LoRA wrt an arbitrary loss.

Also, the authors assume (in the proof) that the angles $\delta_t$ is a decreasing sequence, but is this always a valid assumption? It might not hold in the stochastic setting. This assumption should be stated clearly along with Assumption 5.1 / A.12.

**Methods And Evaluation Criteria:**

The baselines (FFA-LoRA and FedAvg) make sense. I'm not aware of any baselines that might be more competitive, let alone have convergence guarantees. The evaluation is done on various language tasks and is comprehensive enough to demonstrate the effectiveness of RoLoRA.

**Other Comments Or Suggestions:**

I believe the expression for $\tilde{b}$ from line 5 in Algorithm 2 should be written clearly somewhere, even though it might be simple to derive, as it is used directly in the proofs in the appendix, e.g., equation (14). It is not even shown in Table 3.

The proof for Proposition 5.5 is a bit too long. I believe there could be a way to make it more compact, but I cannot offer any concrete suggestions other than reusing steps in the proof and putting them under a lemma or something.

**Other Strengths And Weaknesses:**

The authors offer extensive experimental results and analysis. Most papers offer one part and make the other part unsatisfactory or leave it altogether. This is a strenght of this paper.
The authors also explain the inspiration of their method, which is good transparency.
The method is simple yet effective and does not sacrifice communication budget.
Many algorithms proposed in the literature are unnecessarily complex with marginal improvements.
It is not the case here.
The authors also did not stop at demonstrating the superiority of RoLoRA experimentally, but offered an extensive analysis of their method on the simple problem, and further show that freezing down-projections is provably less robust.

In terms of weaknesses, I would say that the proof is sometimes repetitive and the notation are not the best. Still, these might be stylistic choices that are ultimately tangible to the merit of this work.
Other weaknesses can be found in the Experimental Design & Analyses section above.

**Questions For Authors:**

- Some steps in the proof might not be directly generalizable to rank>1. For example, how would you generalize (17)? Because, In general, $\lVert A^{-1} \rVert $ is not necessarily smaller than $\lVert A \rVert^{-1} $.

- What’s the difference between Lemma A.11 and the first part of the proof of Theorem 5.4 in Sec A.4?

- How does equation (87) follow?

- Performance in Table 12 are pretty close for all methods, including the normal LoRA method. Why do you think that is the case?

**Relation To Broader Scientific Literature:**

The proposed method is a straightforward adjustment to the way LoRAs are trained in FL.
This is relevant to important applications in practice, including next-word prediction.
An efficient method that performs significantly better than the baselines is highly appreciated.
In addition, the analysis is extensive and can benefit future researchers working on alternating optimization of LoRAs, whether in FL or not.

**Theoretical Claims:**

From a first reading, the proof seems to be correct in general, but I'm not sure about the exact constants, etc.
The steps in the proof are explained equation by equation, which is great for verifying the correctness.

---

> ### Author Rebuttal · Authors · 2025-04-01
>
> We thank the reviewer for their thoughtful and detailed assessment of our paper. We are encouraged by the overall positive evaluation and would like to clarify several points.
>
> **Rank-1 Limitation and Generalizability to higher ranks.**
> Thanks for pointing this out. Although our analysis is conducted under the rank-1 setting, it remains highly nontrivial. In particular, we provide a direct and rigorous comparison between the solutions of RoLoRA and FFA-LoRA, clearly demonstrating the limitations of FFA-LoRA in this fundamental case. This already yields valuable theoretical insights into the core differences between the two methods. Prior works on similar algorithms and settings (See Sec 4.1 and 4.2 in [1], and Appendix A.3.1 and A.3.2 in [2]) have successfully analyzed both rank-1 and higher-rank cases using comparable techniques. This suggests that the rank-1 and higher-rank analyses share underlying structures and intuition. We believe our proof can be extended to higher ranks with additional technical work, but doing so would not change our main conclusion regarding FFA-LoRA's limited expressiveness. We outline key steps toward this extension and leave it for future work.
>
>
> *Orthonormalization in Algorithm.* For the rank-1 case, it is only required to normalize the updated $a$ to unit length (Line 12 of Algorithm 2). To maintain orthonormality for the higher-rank case, we need to include a QR step $A^+=AR$
> where $A$ is updated from through GD (cf. Line 11 of Algorithm 2), ${A^+}^\top A^+ =  I_r$, and $R$ is upper-triangular.
>
> *Error metric.* For the rank-r case, we define the subspace distance of two $d\times r$ matrices (with orthonormal columns) as the following:$ {SD}( A, A^*) = ||( I_ d- A A^\top ) A^*||_ F$
> This is a direct generalization of the rank-1 case. Geometrically speaking, $ {\rm SD}( A, A^*) = \sqrt{\sum_{r'=1}^r \sin^2(\theta_i)}$, where $\theta_1,\cdots, \theta_r$ are the principal angles between the column spaces of $ A$ and $ A^*$.
>
> *Generalizing Eq. (17).* For rank-$r$, we similarly bound $|\bar{{B}} - {G}|_{op}$ by ${\rm SD}({A}, {A}^*)$, where ${G} = B(A^*)^\top {A}$. The main technical step is controlling $|({A}^\top {X}_i^\top {X}_i {A})^{-1}|{op}$, handled via Bernstein's inequality and an $\epsilon$-net (see [3], Sec. IV-E).
>
> *Condition-Number Dependecies.* In the rank-1 case, the convergence and sample complexity depend on the norm of the ground-truth signal vector $b^*$, as shown in Eq. (13,44,123). For higher-rank settings, this dependency generalizes to the operator norm (i.e., the largest singular value). However, the operator norm alone does not fully characterize the problem complexity. Instead, the condition number, which is the ratio of largest and smallest singualr value, is critical. The condition number is reduced to 1 when rank=1.
>
> [1]Jain et al., "Low-rank matrix completion using alternating minimization," STOC 2013
>
> [2]Thekumparampil et al., "Statistically and computationally efficient linear meta-representation learning," NeurIPS 2021
>
> [3]Nayer, S., & Vaswani, N. (2022). Fast and sample-efficient federated low rank matrix recovery from column-wise linear and quadratic projections. IEEE Trans. Inf. Theory
>
> **Assumptions on decreasing angles.**
> Thank you for pointing this out. While Lemma 5.3 assumes a decreasing angle sequence, it's only an intermediate step for Theorem 5.4. In Theorem 5.4's proof (Line 1027), we show the angle decreases in the first step and use induction to extend this to all iterations (Line 1139).
> Stochastic setting: Our analysis is deterministic; extending to the stochastic case is left for future work. We'll clarify this in the revision.
>
> **Difference between Lemma A.11 and the first part of the proof of Theorem 5.4.** Thank you for the careful reading. Eq. (123) and Eq. (166) are essentially the same—Eq. (123) assumes a decreasing angle, while Eq. (166) proves this for the first iteration. Setting $t=0$ in Eq. (123) gives Eq. (166). We restated the proof for clarity and completeness but will revise it to be more concise.
>
> **Eq. (87).** Thank you for pointing this out. Eq. (87) follows by normalizing Eq. (86). The reference to Eq. (87) on Line 929 was a typo—it should be Eq. (88). We'll correct this in the revision.
>
> **Performance in Table 12.** Thank you for the observation. LLaMA-2-7B is already strong on MMLU due to its pretraining. Since MMLU focuses on factual recall, not task-specific adaptation, PEFT methods like LoRA, RoLoRA, or FFA-LoRA offer limited gains. This is also observed at Sec 5.3 in [4]. In contrast, we show clear improvements on task-specific adaptation tasks such as GLUE and commonsense reasoning tasks.
>
> [4]Guo, Pengxin, et al. "Selective Aggregation for Low-Rank Adaptation in Federated Learning." (ICLR 2025).
>
> **Related works.** Thanks for the helpful suggestions and for taking the time to check for related work. We will cite them in the related work section.

---

### Official Review · Reviewer_87LM · 2025-03-18

**Overall Recommendation:** 3

**Summary:**

This paper introduces RoLoRA, a federated fine-tuning framework that employs alternating optimization for LoRA-based adaptation. RoLoRA addresses the expressiveness limitations of FFA-LoRA (Sun et al. '24)in low-parameter settings while preserving communication efficiency. The authors provide theoretical proof of RoLoRA’s convergence in a single-layer linear regression model with rank-1 LoRA and highlight the drawbacks of FFA-LoRA, which freezes the LoRA module’s down-projection matrix (A), restricting model expressiveness. Empirical results demonstrate that RoLoRA consistently outperforms baselines and achieves significantly faster convergence.

**Claims And Evidence:**

The paper makes two key claims:


**1.FFA-LoRA has reduced expressiveness due to frozen down-projection matrices.** The authors argue that FFA-LoRA’s inability to update the down-projection matrix (A) limits model expressiveness, making optimization harder. Their theoretical analysis, using a simplified linear regression model, suggests that freezing A prevents the model from fully converging to an optimal solution. Equation (10) and Proposition 5.5 illustrate that, under this restriction, the global objective remains dependent on the initialization of A, leading to suboptimal performance. However, while this analysis demonstrates that freezing A introduces significant limitations, it does not rigorously prove that updating only the up-projection matrix (B) is fundamentally insufficient for optimization in all scenarios.

**2. RoLoRA improves robustness while maintaining efficiency.** RoLoRA introduces alternating updates to the A and B matrices, addressing the expressiveness limitations of FFA-LoRA while preserving its computational and communication efficiency. Unlike FFA-LoRA, which is sensitive to initialization and fine-tuning parameter budgets, RoLoRA achieves more stable optimization and better generalization across different settings. Empirical results across linear models, toy neural networks, and large language models (RoBERTa-Large, Llama-2-7B) demonstrate that RoLoRA consistently achieves higher accuracy and faster convergence than both FFA-LoRA and other baselines.

**Essential References Not Discussed:**

The paper should discuss Koo et al. (2024), “Towards Robust and Efficient Federated Low-Rank Adaptation with Heterogeneous Clients”, which employs a similar alternating LoRA approach even in non-IID federated settings.

**Experimental Designs Or Analyses:**

While the empirical results demonstrate RoLoRA’s effectiveness, additional experiments could further strengthen its claims and generalizability:

 - Robustness Across Different Rank Sizes: Testing RoLoRA with varying rank sizes would help assess its stability and effectiveness in low-parameter settings, particularly where parameter efficiency is critical.

 - Impact of Learning the Down-Projection: A more detailed empirical analysis of the role of the down-projection matrix (A) would clarify how much expressiveness is lost when A is frozen and whether learning it is always beneficial.

 - Evaluation Under Differential Privacy Constraints: Since privacy is a key motivation for federated learning, assessing RoLoRA’s performance under differential privacy (DP) would provide insight into its robustness in privacy-preserving scenarios.

 - Experiments in Heterogeneous Client Settings: Evaluating RoLoRA under non-IID data distributions and client heterogeneity would better reflect real-world FL conditions, where clients have different data distributions and computational constraints.

**Methods And Evaluation Criteria:**

The alternating update strategy in RoLoRA is well-motivated, and the paper evaluates it across various tasks and models. However, the evaluation lacks key comparisons:

 - Missing Baselines: The study does not include important FL+LoRA methods, such as FlexLoRA (Bai et al., 2024), which is crucial for a direct performance and efficiency comparison.

 - Limited to IID Data: The experiments assume IID data distributions, without evaluating non-IID settings, which are common in real-world federated learning. Since LoRA-based fine-tuning can be particularly sensitive to data heterogeneity, testing RoLoRA under non-IID conditions would better assess its robustness.

 - Lack of Differential Privacy Analysis: The study does not analyze differential privacy (DP), despite privacy being one of the key promises of FL. Since inexact model updates can be more problematic in DP-constrained environments, assessing RoLoRA’s performance under privacy-preserving conditions would strengthen its claims.

**Reference:**

Bai, Jiamu, et al. “Federated fine-tuning of large language models under heterogeneous tasks and client resources.” arXiv preprint arXiv:2402.11505 (2024).

**Other Comments Or Suggestions:**

L023: theoretical analysis -> a theoretical analysis
L106: We adopts -> We adopt

Please reorganize the proof for better readability.

**Other Strengths And Weaknesses:**

Additional Concerns:

**No Experiments on in-exact update settings:**
RoLoRA is presented to address the issue of in-exact model updates in a more robust and efficient way. However there is no evaluation under differential privacy or highly heterogeneous client settings.

**Weak link between theoretical analysis and claims:**
Theoretical results do not conclusively prove the necessity of down-projection learning or weakness of FFA-LoRA in fewer parameter settings. but merely provides intuition on constraints of freezing-A scheme.

**Questions For Authors:**

- Can you provide a stronger theoretical justification or empirical validation that freezing A is always suboptimal, especially in low-parameter regimes?


 - Have you tested RoLoRA in non-IID settings, where inexact model updates are more problematic?

 - Have you considered evaluating RoLoRA with DP mechanisms (e.g., DP-SGD, DP-FedAvg)?

 - Can you provide additional experiments to confirm its stability in extreme low-rank settings, where parameter efficiency is crucial?

 - Can you provide ablation studies comparing performance when A is learned versus when it is frozen?

 - How does RoLoRA compare to FlexLoRA in terms of communication efficiency, convergence, and model accuracy?  (I do understand the memory complexity issue, but you could perform a small-scale experiment)

**Relation To Broader Scientific Literature:**

The idea is simple yet the analysis is non-trivial. However, the contribution remains limited without empirical studies in more realistic FL settings, such as differential privacy and non-IID data.

**Theoretical Claims:**

The authors provide a convergence analysis for RoLoRA in a single-layer linear regression model with rank-1 LoRA and demonstrate that FFA-LoRA’s objective function is influenced by the initialization of the down-projection matrix (A). They also analyze a heterogeneous federated setting, but the results primarily focus on angle convergence rather than proving global optimality.

However, the theoretical claims have several limitations:

 - Limited Scope of Convergence Analysis: The convergence proof is restricted to a simplified linear regression model with rank-1 LoRA, which does not directly extend to more complex neural networks or real-world federated learning (FL) settings. The analysis does not account for data heterogeneity, model heterogeneity, or non-convex objectives, which are common in FL.

 - Expressiveness Limitation Not Fully Proven: While the paper argues that FFA-LoRA is less expressive due to freezing the down-projection (A), it does not rigorously prove that learning A is strictly necessary for effective fine-tuning in all fewer-parameter settings. The results suggest a limitation but do not quantify its impact beyond the linear setting.

 - RoLoRA’s Superiority Over FFA-LoRA Not Mathematically Established: The theoretical results support the authors’ intuition that alternating updates improve optimization, but they do not formally establish that RoLoRA consistently outperforms FFA-LoRA in general federated learning scenarios. A stronger theoretical claim would require proving that FFA-LoRA cannot reach the same optima under certain conditions.

While the theoretical insights provide useful intuition, they do not constitute a rigorous proof that RoLoRA is strictly superior to FFA-LoRA in all cases. Extending the analysis to non-convex settings, heterogeneous data distributions, and multi-layer models would strengthen the claims, although I do understand the challenges.

---

> ### Author Rebuttal · Authors · 2025-04-01
>
> Thank you for the detailed and constructive review. We address the key concerns below:
>
> **FFA-LoRA's Reduced Expressiveness and RoLoRA's Theoretical Superiority.**
> Thank you for the constructive comments. We show in Proposition 5.5 (proof in Appendix A.4.1) that FFA-LoRA is suboptimal in our linear setting. This result holds for any unit vector $\mathbf{a}$ and corresponding $\mathbf{b}$ obtained by fully minimizing the local loss, as in Line 5 and 7 of Algorithm 2. The same expected loss bound applies to RoLoRA. As discussed in Line 271-280, substituting RoLoRA's reduced angle $\epsilon$ at Line 259 into Eq. (11) shows its expected loss can be made arbitrarily small, unlike FFA-LoRA, whose loss is limited by the initial angle.
>
> **Scope of Convergence Analysis**
> Our convergence analysis also extends to the heterogeneous setting (Appendix A.5), following the same logic as the homogeneous case. RoLoRA reduces the angle between $\mathbf{a}$ and $\mathbf{a}^*$, ensuring convergence to the global optimum, while FFA-LoRA's loss remains limited by its initial angle, as discussed at Line 281.
>
> Though based on a simplified linear model, the proof is non-trivial. The linear setting allows direct comparison due to its unique global minimum, unlike neural networks, where only convergence to local minima can be shown and final losses across methods are not directly comparable.
>
> **Non-IID robustness.**
> Thank you for the constructive comments. We provide a theoretical analysis under a heterogeneous linear setting (Appendix A.5) and evaluate RoLoRA's robustness to non-IID data using a two-layer neural network (Fig. 2). Additionally, we ran experiments on a language model under non-IID conditions—see the **[table](https://anonymous.4open.science/r/random-3C54/noniid.png)**. RoLoRA consistently outperforms LoRA, FFA-LoRA, and FlexLoRA on MNLI and QQP across varying data heterogeneity and client counts.
>
> **Re-Organization of Proof.**
> Thank you for the thoughtful feedback. We already provide a high-level overview of the main proof at Line 271. To improve clarity, we will revise the appendix to include a more detailed outline and highlight key intuitions behind the technical steps.
>
> **Differential Privacy.** Thanks for the suggestion. We itegrated NbAFL [1] and the results are in the **[table](https://anonymous.4open.science/r/random-3C54/dp.png)**. We use $\epsilon = 10, \delta = 1e-6$. In this setting, RoLoRA outperform others across MNLI and QQP tasks.
>
> [1]Wei, Kang, et al. "Federated learning with differential privacy: Algorithms and performance analysis." IEEE transactions on information forensics and security 15 (2020)
>
> **Extreme low-rank stability.**
> Thank you for raising this point. Our experiments already include rank 1, 2, 4, and 8 settings (see Fig. 4, 6-8 in Appendix B.2.2 at Line 1743), demonstrating RoLoRA's robustness under tight parameter budgets. Table 1 and 6 show results with increasing clients at rank 4 and 8. We've also added experiments with rank 2 and different numbers of clients—see the **[table and figure](https://anonymous.4open.science/r/random-3C54/rank2.png)**—where RoLoRA still shows strong convergence and competitive accuracy.
>
> **Ablation study of the role of A.** Thank you for the insightful suggestion. To address this, we conducted an experiment comparing performance of FFA-LoRA, RoLoRA, and different mixing strategies under the setting with 50 clients. In these strategies, for example, 20%RoLoRA+80%FFA-LoRA means we finetune with RoLoRA (where A is learned) for the first 20% of communication rounds, followed by FFA-LoRA (where A is frozen) for the remaining 80%. The results are shown in the **[figure](https://anonymous.4open.science/r/random-3C54/ablation-A.png)**. We observe that finetuning with RoLoRA generally leads to faster convergence and higher final accuracy, highlighting the benefits of learning A, especially in early training.
>
>
>
> **Comparison with FlexLoRA.** Thank you for your comment. Our paper already compares RoLoRA and FlexLoRA across ranks and client counts (Table 6, Line 1705). We've added convergence results in a 50-client setting **[figure](https://anonymous.4open.science/r/random-3C54/convergence-100.png)** and a **[table](https://anonymous.4open.science/r/random-3C54/cost-compare.png)** comparing communication and time costs. Results show RoLoRA's clear advantages in large-scale, low-resource settings.
>
> **Related works.** Thanks for taking the time to check for related work. We will discuss Koo et al. (2024) in the related work section.

---

> > ### Comment · Reviewer_87LM · 2025-04-05
> >
> > Thanks for the response, which have resolved my major concerns. I have raised my score from 2 to 3.

---

> > > ### Author Response · Authors · 2025-04-07
> > >
> > > Thanks for the acknowledge of our work and for raising score. Thanks again for your time and effort in reviewing our paper.

---

### Decision · Program_Chairs · 2025-05-01

**Decision:**

Reject

**Comment:**

The paper introduces RoLoRA, a federated fine-tuning framework for LLMs that alternately optimizes both up-projection and down-projection matrices in LoRA adapters. This approach addresses expressiveness limitations in prior methods like FFA-LoRA, achieving better accuracy, robustness, and communication efficiency. Theoretical analysis demonstrates exponential convergence in a simplified linear model, and experiments on RoBERTa-Large and Llama-2-7B validate its superiority across tasks like GLUE and commonsense reasoning. However, broader theoretical analysis is needed to strengthen its impact.